# Kullback-Leibler Maillard Sampling for Multi-armed Bandits with Bounded Rewards

**Hao Qin**
University of Arizona
hqin@arizona.edu

**Kwang-Sung Jun**
University of Arizona
kjun@cs.arizona.edu

**Chicheng Zhang**
University of Arizona
chichengz@cs.arizona.edu

## Abstract

We study $K$-armed bandit problems where the reward distributions of the arms are all supported on the $[0, 1]$ interval. Maillard sampling [30], an attractive alternative to Thompson sampling, has recently been shown to achieve competitive regret guarantees in the sub-Gaussian reward setting [11] while maintaining closed-form action probabilities, which is useful for offline policy evaluation. In this work, we analyze the Kullback-Leibler Maillard Sampling (KL-MS) algorithm, a natural extension of Maillard sampling and a special case of Minimum Empirical Divergence (MED) [19] for achieving a KL-style finite-time gap-dependent regret bound. We show that KL-MS enjoys the asymptotic optimality when the rewards are Bernoulli and has an adaptive worst-case regret bound of the form $O(\sqrt{\mu^*(1 - \mu^*)KT \ln K} + K \ln T)$, where $\mu^*$ is the expected reward of the optimal arm, and $T$ is the time horizon length; this is the first time such adaptivity is reported in the literature for an algorithm with asymptotic optimality guarantees.

## 1 Introduction

The multi-armed bandit (abbrev. MAB) problem [41, 27, 29], a stateless version of the reinforcement learning problem, has received much attention by the research community, due to its relevance in may applications such as online advertising, recommendation, and clinical trials. In a multi-armed bandit problem, a learning agent has access to a set of $K$ arms (also known as actions), where for each $i \in [K] := \{1, \ldots, K\}$, arm $i$ is associated with a distribution $\nu_i$ with mean $\mu_i$; at each time step $t$, the agent adaptively chooses an arm $I_t \in [K]$ by sampling from a probability distribution $p_t \in \Delta^{K-1}$ and receives reward $y_t \sim \nu_{I_t}$, based on the information the agent has so far. The goal of the agent is to minimize its pseudo-regret over $T$ time steps: $\mathrm{Reg}(T) = T\mu^* - \mathbb{E} \sum_{t=1}^{T} y_t$, where $\mu^* = \max_i \mu_i$ is the optimal expected reward.

In this paper, we study the multi-armed bandit setting where reward distributions of all arms are supported on $[0, 1]$. [1] An important special case is Bernoulli bandits, where for each arm $i$, $\nu_i = \mathrm{Bernoulli}(\mu_i)$ for some $\mu_i \in [0, 1]$. It has practical relevance in settings such as computational advertising, where the reward feedback is oftentimes binary (click vs. not-click, buy vs. not-buy).

Broadly speaking, there are two popular families of provably regret-efficient algorithms for bounded-reward bandit problems: deterministic exploration algorithms (such as KL-UCB [17, 13, 31]) and randomized exploration algorithms (such as Thompson sampling (TS) [41]). Randomized exploration algorithms such as TS have been very popular, perhaps due to its excellent empirical performance and the ability to cope with delayed rewards better than deterministic counterparts [15]. In addition, the logged data collected from randomized exploration, of the form $(I_t, p_{t,I_t}, y_t)_{t=1}^{T}$, where $p_{t,I_t}$ is the probability with which arm $I_t$ was chosen, are useful for offline evaluation purposes by

---

[1] All of our results can be extended to distributions supported in $[L, U]$ for any known $L \leq U$ by shifting and scaling the rewards to lie in $[0, 1]$.

37th Conference on Neural Information Processing Systems (NeurIPS 2023).

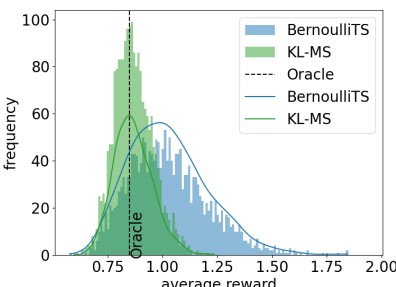

Figure 1: Histogram of the average rewards computed from the offline evaluation where the logged data is collected from Bernoulli TS and KL-MS (Algorithm 1) in a Bernoulli bandit environment with the mean reward (0.8, 0.9) with time horizon $T = 10,000$. For Bernoulli TS's log, we approximate the action probability by Monte Carlo Sampling with 1000 samples for each step. Here we estimate the expected reward of the uniform policy which has expected average reward of 0.85 (black dashed line). Across 2000 trials, the logged data of KL-MS induces an MSE of 0.00796; however, for half of the trials, the IPW estimator induced by Bernoulli TS's log returns invalid values due to the action probability estimates being zero. Even excluding those invalid values, the Bernoulli TS's logged data induces an MSE of 0.02015. See Appendix H for additional experiments.

employing the inverse propensity weighting (IPW) estimator [22] or the doubly robust estimator [38]. However, calculating the arm sampling probability distribution $p_t$ for Thompson sampling is nontrivial. Specifically, there is no known closed-form [2], and generic numerical integration methods and Monte-Carlo approximations suffer from instability issues: the time complexity for obtaining a numerical precision of $\epsilon$ is $\Omega(\text{poly}(1/\epsilon))$ [36]. This is too slow to be useful especially for web-scale deployments; e.g., Google AdWords receives ∼237M clicks per day. Furthermore, the computed probability will be used after taking the inversion, which means that even moderate amount of errors are intolerable. Indeed, Figure 1 shows that the offline evaluation with Thompson sampling as the behavioral policy will be largely biased and inaccurate due to the errors from the Monte Carlo approximation.

Recently, many studies have introduced alternative randomized algorithms that allow an efficient computation of $p_t$ [19, 30, 14, 43]. Of these, Maillard sampling (MS) [30, 11], a Gaussian adaptation of the Minimum Empirical Divergence (MED) algorithm [19] originally designed for finite-support reward distributions, provides a simple algorithm for the sub-Gaussian bandit setting that computes $p_t$ in a closed form:

$$p_{t,a} \propto \exp\left(-N_{t-1,a}\frac{\hat{\Delta}_{t-1,a}^2}{2\sigma^2}\right) \tag{1}$$

where at time step $t$, $N_{t,a}$ is the number of pulling arm $a$. We define the estimator of $\mu_a$ as $\hat{\mu}_{t,a} := \frac{\sum_{s=1}^t \mathbf{1}\{I_t=a\}y_t}{N_{t,a}}$ and the best performed mean value as $\hat{\mu}_{t,\max} := \max_{a \in [K]} \mu_{t,a}$. $\hat{\Delta}_{t-1,a} = \max_{a'} \hat{\mu}_{t-1,a'} - \hat{\mu}_{t-1,a}$ is the empirical suboptimality gap of arm $a$, and $\sigma$ is the subgaussian parameter of the reward distribution of all arms. For sub-Gaussian reward distributions, MS enjoys the asymptotic optimality under the special case of Gaussian rewards and a near-minimax optimality [11], making it an attractive alternative to Thompson sampling. Also, MS satisfies the sub-UCB criterion (see Section 2 for a precise definition) to help establish sharp finite-time instance-dependent regret guarantees. Can we adapt MS to the bounded reward setting and achieve the asymptotic, minimax optimality and sub-UCB criterion while computing the sampling probability in a closed-form? In this paper, we make significant progress on this question.

**Our contributions.** We focus on a Bernoulli adaptation of MS that we call Kullback-Leibler Maillard Sampling (abbrev. KL-MS) and perform a finite-time analysis of it in the bounded-reward bandit problem. KL-MS uses a sampling probability similar to MS but tailored to the $[0, 1]$-bounded reward setting:

$$p_{t,a} \propto \exp\left(-N_{t-1,a}\mathsf{kl}\left(\hat{\mu}_{t-1,a}, \hat{\mu}_{t-1,\max}\right)\right),$$

where $\mathsf{kl}(\mu, \mu') := \mu \ln \frac{\mu}{\mu'} + (1-\mu) \ln \frac{1-\mu}{1-\mu'}$ is the binary Kullback-Leibler (KL) divergence. We can also view KL-MS as an instantiation of MED [19] for Bernoulli rewards; See Section 3 for a

---

[2]Suppose the arms' mean reward posterior distributions' PMFs and PDFs are $(p_1, \ldots, p_K)$ and $(F_1, \ldots, F_K)$ respectively; for example, they are Beta distributions with different parameters, i.e. $p_i(x) = x^{a_i-1}(1 - x)^{b_i-1}I(x \in [0, 1])$ for some $a_i, b_i$. To the best of our knowledge, the action probabilities have the following integral expression: $\mathbb{P}(I_t = a) = \int_{\mathbb{R}} p_a(x) \prod_{i \neq a} F_i(x)\,\mathrm{d}x$ and cannot be further simplified.

| Algorithm& Analysis | Finite-Time Regret Minimax Ratio | Sub-UCB | Closed-form Probability | Reference |
|---|---|---|---|---|
| TS | $\sqrt{\ln K}$ | yes | no | See the caption |
| ExpTS | $\sqrt{\ln K}$ | yes | no | Jin et al. [24] |
| ExpTS$^+$ | 1 | $-^{\star\star}$ | no | Jin et al. [24] |
| kl-UCB | $\sqrt{\ln T}$ | yes | N/A | Cappé et al. [13] |
| kl-UCB++ | 1 | $-^{\star\star}$ | N/A | Ménard and Garivier [33] |
| kl-UCB-switch | 1 | $-^{\star\star}$ | N/A | Garivier et al. [18] |
| MED | $-$ | $-$ | no$^{\star}$ | Honda and Takemura [19] |
| DMED | $-$ | $-$ | N/A | Honda and Takemura [20] |
| IMED | $-$ | $-$ | N/A | Honda and Takemura [21] |
| KL-MS | $\sqrt{\ln K}$ | yes | yes | this paper |

Table 1: Comparison of regret bounds for bounded reward distributions; for space constraints we only include those that achieves the asymptotic optimality for the special case of Bernoulli distributions (this excludes, e.g., Maillard Sampling [30, 11], Tsallis-INF [43] and UCB-V [8]). '−'indicates that the corresponding analysis is not reported. 'N/A'indicates that the algorithm does have closed-form, but it is deterministic. '$\star$'indicates that its computational complexity for calculating the action probability is $\ln(1/\text{precision})$. '$\star\star$'indicates that we conjecture that the algorithm is not sub-UCB. The results on TS are reported by Agrawal and Goyal [3, 4], Korda et al. [26].

detailed comparison. KL-MS performs an efficient exploration for bounded rewards since one can use $\text{kl}(a,b) \geq 2(a-b)^2$ to verify that the probability being assigned to each empirical non-best arm by KL-MS is never larger than that of MS with $\sigma^2 = 1/4$, the best sub-Gaussian parameter for the bounded rewards in $[0,1]$. We show that KL-MS achieves a sharp finite-time regret guarantee (Theorem 1) that can be simultaneously converted to:

- an asymptotic regret upper bound (Theorem 4), which is asymptotically optimal when specialized to the Bernoulli bandit setting;
- a $\sqrt{T}$-style regret guarantee of $O(\sqrt{\mu^*(1-\mu^*)KT\ln K} + K\ln(T))$ (Theorem 3) where $\mu^*$ is the mean reward of the best arm. This bound has two salient features. First, in the worst case, it is at most a $\sqrt{\ln K}$ factor suboptimal than the minimax optimal regret of $\Theta(\sqrt{KT})$ [5, 10]. Second, its $\tilde{O}(\sqrt{\mu^*(1-\mu^*)})$ coefficient adapts to the variance of the optimal arm reward; this is the first time such adaptivity is reported in the literature for an algorithm with asymptotical optimality guarantees. [3]
- a sub-UCB regret guarantee, which many existing minimax optimal algorithms [33, 18] have not been proven to satisfy.

We also conduct experiments that show that thanks to its closed-form action probabilities, KL-MS generates much more reliable logged data than Bernoulli TS with Monte Carlo estimation of action probabilities; this is reflected in their offline evaluation performance using the IPW estimator; see Figure 1 and Appendix H for more details.

## 2 Preliminaries

Let $N_{t,a}$ be the number of times arm $a$ has been pulled until time step $t$ (inclusively). Denote the suboptimality gap of arm $a$ by $\Delta_a := \mu^* - \mu_a$, where $\mu^* = \max_{i \in [K]} \mu_i$ is the optimal expected reward. Denote the empirical suboptimality gap of arm $a$ by $\hat{\Delta}_{t,a} := \hat{\mu}_{t,\max} - \hat{\mu}_{t,a}$; here, $\hat{\mu}_{t,a}$ is the empirical estimation to $\mu_a$ up to time step $t$, i.e., $\hat{\mu}_{t,a} := \frac{1}{N_{t,a}} \sum_{s=1}^{t} y_s \mathbf{1}\{I_s = a\}$, and $\hat{\mu}_{t,\max} = \max_{a \in [K]} \hat{\mu}_{t,a}$ is the best empirical reward at time step $t$. For arm $a$, define $\tau_a(s) := \min\{t \geq 1 : N_{t,a} = s\}$ at the time step when arm $a$ is pulled for the $s$-th time, which is a stopping time; we also use $\hat{\mu}_{(s),a} := \hat{\mu}_{\tau_a(s),a}$ to denote empirical mean of the first $s$ reward values received from pulling arm $a$.

---

[3]As side results, we show in Appendix F that with some modifications of the analysis, existing algorithms [7, 13, 33] also achieve regret of the form $\tilde{O}(\sqrt{\mu^*(1-\mu^*)\text{poly}(K)T})$ for $[0,1]$-bounded reward MABs.

We define the Kullback-Leibler divergence between two distributions $\nu$ and $\rho$ as $\mathsf{KL}(\nu, \rho) = \mathbb{E}_{X \sim \nu}\left[\ln \frac{d\nu}{d\rho}(X)\right]$ if $\nu$ is absolutely continuous w.r.t. $\rho$, and $= +\infty$ otherwise. Recall that we define the binary Kullback-Leibler divergence between two numbers $\mu, \mu'$ in $[0, 1]$ as $\mathsf{kl}(\mu, \mu') := \mu \ln \frac{\mu}{\mu'} + (1 - \mu) \ln \frac{1-\mu}{1-\mu'}$, which is also the KL divergence between two Bernoulli distributions with mean parameters $\mu$ and $\mu'$ respectively. We define $\dot{\mu} = \mu(1 - \mu)$, which is the variance of Bernoulli($\mu$) but otherwise an upper bound on any distribution supported on $[0, 1]$ with mean $\mu$; see Lemma 16 for a formal justification.

In the regret analysis, we will oftentimes use the following notation for comparison up to constant factors: define $f \lesssim g$ (resp. $f \gtrsim g$) to denote that $f \leq Cg$ (resp. $f \geq Cg$) for some numerical constant $C > 0$. We define $a \vee b$ and $a \wedge b$ as $\max(a, b)$ and $\min(a, b)$, respectively. For an event $E$, we use $E^c$ to denote its complement.

Below, we define some useful criteria for measuring the performance of bandit algorithms, specialized to the $[0, 1]$ bounded reward setting.

**Asymptotic optimality in the Bernoulli reward setting** An algorithm is asymptotically optimal in the Bernoulli reward setting [27, 12] if for any Bernoulli bandit instance $(\nu_a = \text{Bernoulli}(\mu_a))_{a \in [K]}$, $\limsup_{T \to \infty} \frac{\text{Reg}(T)}{\ln T} = \sum_{a: \Delta_a > 0} \frac{\Delta_a}{\mathsf{kl}(\mu_a, \mu^*)}$.

**Minimax ratio** The minimax optimal regret of the $[0, 1]$ bounded reward bandit problem is $\Theta\left(\sqrt{KT}\right)$ [5, 10]. Given a $K$-armed bandit problem with time horizon $T$, an algorithm has a minimax ratio of $f(T, K)$ if its has a worst-case regret bound of $O(\sqrt{KT}f(T, K))$.

**Sub-UCB** Sub-UCB is originally defined in the context of sub-Gaussian bandits [29]: given a bandit problem with $K$ arms whose reward distributions are all sub-Gaussian, an algorithm is said to be sub-UCB if there exists some positive constants $C_1$ and $C_2$, such that for all $\sigma^2$-sub-Gaussian bandit instances, $\text{Reg}(T) \leq C_1 \sum_{a: \Delta_a > 0} \Delta_a + C_2 \sum_{a: \Delta_a > 0} \frac{\sigma^2}{\Delta_a} \ln T$. Specialized to our setting, as any distribution supported on $[0, 1]$ is also $\frac{1}{4}$-sub-Gaussian, and all suboptimal arm gaps $\Delta_a \in (0, 1]$ are such that $\Delta_a < \frac{1}{\Delta_a}$, the above sub-UCB criterion simplifies to: there exists some positive constant $C$, such that for all $[0, 1]$-bounded reward bandit instances, $\text{Reg}(T) \leq C \sum_{a: \Delta_a > 0} \frac{\ln T}{\Delta_a}$.

## 3 Related Work

**Bandits with bounded rewards.** Early works of Lai et al. [27], Burnetas and Katehakis [12] show that in the bounded reward setting, for any consistent stochastic bandit algorithm, the regret is lower bounded by $(1 + o(1)) \sum_{a: \Delta_a > 0} \frac{\Delta_a \ln T}{\mathsf{KL}_{\inf}(\nu_a, \mu^*)}$ and $\mathsf{KL}_{\inf}(\nu_a, \mu^*)$ is defined as

$$\mathsf{KL}_{\inf}(\nu, \mu^*) := \inf \left\{ \mathsf{KL}(\nu, \rho) : \mathbb{E}_{X \sim \rho}[X] > \mu^*, \text{supp}(\rho) \subset [0, 1] \right\}, \tag{2}$$

where the random variable follows a distribution $\rho$ bounded in $[0, 1]$. Therefore, any algorithm whose regret upper bound matches the lower bound is said to achieve asymptotic optimality. Cappé et al. [13] propose the KL-UCB algorithm and provide a finite time regret analysis, which is further refined by Lattimore and Szepesvári [29, Chapter 10]. Another line of work establishes asymptotic and finite-time regret guarantees for Thompson sampling algorithms and its variants [2, 4, 25, 24], which, when specialized to the Bernoulli bandit setting, can be combined with Beta priors for the Bernoulli parameters to design efficient algorithms.

A number of studies even go beyond the Bernoulli-KL-type regret bound and adapt to the variance of each arm in the bounded reward setting. UCB-V [7] achieves a regret bound that adapts to the variance. Efficient-UCBV [35] achieves a variance-adaptive regret bound and also an optimal minimax regret bound $O(\sqrt{KT})$, but it is not sub-UCB. Honda and Takemura [19] propose the MED algorithm that is asymptotically optimal for bounded rewards, but it only works for rewards that with finite supports. Honda and Takemura [21] propose the Indexed MED (IMED) algorithm that can handle a more challenging case where the reward distributions are supported in $(-\infty, 1]$.

As with worst-case regret bounds, first, it is well-known that for Bernoulli bandits as well as bandits with $[0, 1]$ bounded rewards, the minimax optimal regrets are of order $\Theta(\sqrt{KT})$ [10, 5]. Of the algorithms that enjoy asymptotic optimality under the Bernoulli reward setting described above, KL-UCB [13] has a worst-case regret bound of $O(\sqrt{KT \ln T})$, which is refined by the KL-UCB++

algorithm [33] that has a worst-case regret bound of $O(\sqrt{KT})$. We also show in Appendix F.1 and F.2 that with some modifications of existing analysis, KL-UCB and KL-UCB++ enjoy a regret bound of $O(\sqrt{\mu^*(1-\mu^*)KT\ln T})$ and $O(\sqrt{\mu^*(1-\mu^*)K^3 T\ln T})$ respectively. Although the regret is worse in the order of $K$, it adapts to $\mu^*$ and will have a better regret when $\mu^*$ is small (say, $\mu^* \leq 1/K^2$). KL-UCB++[34] and KL-UCB-Switch[18] achieves $O(\sqrt{KT})$ regret in the finite-time regime and asymptotic optimality, while the sub-UCB criterion has not been satisfied. However, Lattimore [28, §3] shows that MOSS [6] suffers a sub-optimal regret worse than UCB-like algorithms because of not satisfying sub-UCB criteria, and we suspect that KL-UCB-switch experience the same issue as MOSS. For Thompson Sampling style algorithms, Agrawal and Goyal [3] shows that the original Thompson Sampling algorithm has a worst-case regret of $O(\sqrt{KT\ln K})$, and the ExpTS+ algorithm [24] has a worst-case regret of $O(\sqrt{KT})$.

**Randomized exploration for bandits.** Many randomized exploration methods have been proposed for multi-armed bandits. Perhaps the most well-known is Thompson sampling [41], which is shown to achieve Bayesian and frequentist-style regret bounds in a broad range of settings [39, 2, 25, 26, 23, 24]. A drawback of Thompson sampling, as mentioned above, is that the action probabilities cannot be obtained easily and robustly. To cope with this, a line of works design randomized exploration algorithms with action probabilities in closed forms. For sub-Gaussian bandits, Cesa-Bianchi et al. [14] propose a variant of the Boltzmann exploration rule (that is, the action probabilities are proportional to exponential to empirical rewards, scaled by some positive numbers), and show that it has $O\left(\frac{K\ln^2 T}{\Delta}\right)$ instance-dependent and $O\left(\sqrt{KT}\ln K\right)$ worst-case regret bounds respectively, where $\Delta = \min_{a:\Delta_a > 0} \Delta_a$ is the minimum suboptimalty gap. Maillard sampling (MS; Eq. (1)) is an algorithm proposed by the thesis of Maillard [30] where the author reports that MS achieves the asymptotic optimality and has a finite-time regret of order $\sum_{a:\Delta_a > 0}\left(\frac{\ln T}{\Delta_a} + \frac{1}{\Delta_a^3}\right)$ from which a worst-case regret bound of $O(\sqrt{K}T^{3/4})$ can be derived. MED [19], albeit achieves asymptotic optimality for a broad family of bandits with finitely supported reward distributions, also has a high finite-time regret bound of at least $\sum_{a:\Delta_a > 0}\left(\frac{\ln T}{\Delta_a} + \frac{1}{\Delta_a^{2|\mathrm{supp}(\nu_1)|-1}}\right)$. [4] Recently, Bian and Jun [11] report a refined analysis of Maillard [30]'s sampling rule, showing that it has a finite time regret of order $\sum_{a:\Delta_a > 0}\frac{\ln(T\Delta_a^2)}{\Delta_a} + O\left(\sum_{a:\Delta_a > 0}\frac{1}{\Delta_a}\ln(\frac{1}{\Delta_a})\right)$, and additionally enjoys a $O\left(\sqrt{KT\ln T}\right)$ worst-case regret, and by inflating the exploration slightly (called MS$^+$), the bound can be improved and enjoy the minimax regret of $O\left(\sqrt{KT\ln K}\right)$, which matches the best-known regret bound among those that satisfy sub-UCB criterion, except for AdaUCB. In fact, it is easy to adapt our proof technique in this paper to show that MS, without any further modification, achieves a $O\left(\sqrt{KT\ln K}\right)$ worst-case regret.

Randomized exploration has also been studied from a nonstochastic bandit perspective [10, 5], where randomization serves both as a tool for exploration and a way to hedge bets against the nonstationarity of the arm rewards. Many recent efforts focus on designing randomized exploration bandit algorithms that achieve "best of both worlds" adaptive guarantees, i.e., achieving logarithmic regret for stochastic environments while achieving $\sqrt{T}$ regret for adversarial environments [e.g. 43, 42].

**Binarization trick.** It is a folklore result that bandits with $[0, 1]$ bounded reward distributions can be reduced to Bernoulli bandits via a simple binarization trick: at each time step $t$, the learner sees reward $r_t \in [0, 1]$, draws $\tilde{r}_t \sim \mathrm{Bernoulli}(r_t)$ and feeds it to a Bernoulli bandit algorithm. However, this reduction does not result in asymptotic optimality for the general bounded reward setting, where the asymptotic optimal regret is of the form $(1 + o(1))\sum_{a:\Delta_a > 0}\frac{\Delta_a \ln T}{\mathsf{KL}_{\inf}(\nu_a, \mu^*)}$ with $\mathsf{KL}_{\inf}(\nu_a, \mu^*)$ defined in the Eq (2). If we combine the binarization trick and the MED algorithm in the bounded reward setting, the size of the support set is viewed as 2, the finite-time regret bound is at best as $O(K^{1/4}T^{3/4})$ (ignoring logarithmic factors), which is much higher than $O(\sqrt{KT})$.

---

[4] A close examination of [19]'s Lemma 9 (specifically, equation (20)) shows that for each suboptimal arm $a$, the authors bound $\mathbb{E}[N_{T,a}]$ by a term at least $\sum_{t=1}^T K(t+1)^{|\mathrm{supp}(\nu_1)|} \cdot \exp\left(-tC(\mu_1, \mu_1 - \varepsilon)\right)$, where $C(\mu, \mu') := \frac{(\mu - \mu')^2}{2\mu'(1+\mu)}$ and $\varepsilon \leq \Delta_a$; this is $\Omega\left(\frac{1}{\Delta_a^{2|\mathrm{supp}(\nu_1)|}}\right)$ when $\mu_1$ is bounded away from 0 and 1.

---

**Algorithm 1** KL Maillard Sampling (KL-MS)

---

1: **Input:** $K \geq 2$
2: **for** $t = 1, 2, \cdots, T$ **do**
3:    **if** $t \leq K$ **then**
4:       Pull the arm $I_t = t$ and observe reward $y_t \sim \nu_i$.
5:    **else**
6:       For every $a \in [K]$, compute

$$p_{t,a} = \frac{1}{M_t} \exp\left(-N_{t-1,a} \cdot \mathsf{kl}(\hat{\mu}_{t-1,a}, \hat{\mu}_{t-1,\max})\right) \tag{3}$$

      where $M_t = \sum_{a=1}^{K} \exp(-N_{t-1,a}\mathsf{kl}(\hat{\mu}_{t-1,a}, \hat{\mu}_{t-1,\max}))$ is the normalizer.
7:       Pull the arm $I_t \sim p_t$.
8:       Observe reward $y_t \sim \nu_{I_t}$.
9:    **end if**
10: **end for**

---

**Bandit algorithms with worst-case regrets that depend on the optimal reward.** Recent linear logistic bandit works have shown worst-case regret bounds that depend on the variance of the best arm [32, 1]. When the arms are standard basis vectors, logistic bandits are equivalent to Bernoulli bandits, and the bounds of Abeille et al. [1] become $\tilde{O}\left(K\sqrt{\dot{\mu}^* T} + \frac{K^2}{\dot{\mu}_{\min}} \wedge (K^2 + A)\right)$ where $\dot{\mu}_{\min} = \min_{i \in [K]} \dot{\mu}_i$ and $A$ is an instance dependent quantity that can be as large as $T$. This bound, compared to ours, has an extra factor of $\sqrt{K}$ in the leading term and the lower order term has an extra factor of $K$. Even worse, it has the term $\dot{\mu}_{\min}^{-1}$ in the lower order term, which can be arbitrarily large. The bound in Mason et al. [32] becomes $\tilde{O}\left(\sqrt{\dot{\mu}^* KT} + \dot{\mu}_{\min}^{-1} K^2\right)$, which matches our bound in the leading term up to logarithmic factors yet still have extra factors of $K$ and $\dot{\mu}_{\min}^{-1}$ in the lower order term.

## 4 Main Result

**The KL Maillard Sampling Algorithm.** We propose an algorithm called KL Maillard sampling (KL-MS) for bounded reward distributions (Algorithm 1). For the first $K$ times steps, the algorithm pulls each arm once (steps 3 to 4); this ensures that starting from time step $K + 1$, the estimates of the reward distribution of all arms are well-defined. From time step $t = K + 1$ on, the learner computes the empirical mean $\hat{\mu}_{t-1,a}$ of all arms $a$. For each arm $a$, the learner computes the binary KL divergence between $\hat{\mu}_{t-1,a}$ and $\hat{\mu}_{t-1,\max}$, $\mathsf{kl}(\hat{\mu}_{t-1,a}, \hat{\mu}_{t-1,\max})$, as a measure of empirical suboptimality of that arm. The sampling probability of arm $a$, denoted by $p_{t,a}$, is proportional to the exponential of negative product between $N_{t-1,a}$ and $\mathsf{kl}(\hat{\mu}_{t-1,a}, \hat{\mu}_{t-1,\max})$ (Eq. (3) of step 6). This policy naturally trades off between exploration and exploitation: arm $a$ is sampled with higher probability, if either it has not been pulled many times ($N_{t-1,a}$ is small) or it appears to be close to optimal empirically ($\mathsf{kl}(\hat{\mu}_{t-1,a}, \hat{\mu}_{t-1,\max})$) is small). The algorithm samples an arm $I_t$ from $p_t$, and observe a reward $y_t$ of the arm chosen.

We remark that if the reward distributions $\nu_i$'s are Bernoulli, KL-MS is equivalent to the MED algorithm [19] since in this case, all reward distributions have a binary support of $\{0, 1\}$. However, KL-MS is different from MED in general: MED computes the empirical distributions of arm rewards $\hat{F}_{t-1,a}$, and chooses action according to probabilities $p_{t,a} \propto \exp(-N_{t-1,a}D_{t-1,a})$; here, $D_{t-1,a} := \mathsf{KL}(\hat{F}_{t-1,a}, \hat{\mu}_{t-1,\max})$ (recall its definition in Section 3) is the "minimum empirical divergence" between arm $a$ and the highest empirical mean reward, which is different from the binary KL divergence of the mean rewards used in KL-MS.

### 4.1 Main Regret Theorem

Our main result of this paper is the following theorem on the regret guarantee of KL-MS (Algorithm 1). Without loss of generality, throughout the rest of the paper, we assume $\mu_1 \geq \mu_2 \geq \cdots \geq \mu_K$.

**Theorem 1.** *For any $K$-arm bandit problem with reward distribution supported on $[0,1]$, KL-MS has regret bounded as follows. For any $\Delta \geq 0$ and $c \in (0, \frac{1}{4}]$:*

$$\mathrm{Reg}(T) \leq T\Delta + \sum_{a:\Delta_a > \Delta} \frac{\Delta_a \ln(T\mathsf{kl}(\mu_a + c\Delta_a, \mu_1 - c\Delta_a) \vee e^2)}{\mathsf{kl}(\mu_a + c\Delta_a, \mu_1 - c\Delta_a)}$$

$$+ 560 \sum_{a:\Delta_a > \Delta} \left( \frac{\dot{\mu}_1 + \Delta_a}{c^2 \Delta_a} \right) \ln \left( \left( \left( \frac{\dot{\mu}_1 + \Delta_a}{c^2 \Delta_a^2} \wedge \frac{c^2 T \Delta_a^2}{\dot{\mu}_1 + \Delta_a} \right) \vee e^2 \right) \right) \quad (4)$$

The regret bound of Theorem 1 is composed of three terms. The first term is $T\Delta$, which controls the contribution of regret from all $\Delta$-near-optimal arms. The second term is asymptotically $(1 + o(1)) \sum_{a:\Delta_a > 0} \frac{\Delta_a}{\mathsf{kl}(\mu_a, \mu_1)} \ln(T)$ with an appropriate choice of $c$, which is a term that grows in $T$ in a logarithmic rate. The third term is simultaneously upper bounded by two expressions. One is $\sum_{a:\Delta_a > 0} \left( \frac{\dot{\mu}_1 + \Delta_a}{c^2 \Delta_a} \right) \ln \left( \frac{c^2 T \Delta_a^2}{\dot{\mu}_1 + \Delta_a} \vee e^2 \right)$, which is of order $\ln T$ and helps establish a tight worst-case regret bound (Theorem 3); the other is $\sum_{a:\Delta_a > 0} \left( \frac{\dot{\mu}_1 + \Delta_a}{c^2 \Delta_a} \right) \ln \left( \left( \frac{\dot{\mu}_1 + \Delta_a}{c^2 \Delta_a^2} \right) \vee e^2 \right)$, which does not grow in $T$ and helps establish a tight asymptotic upper bound on the regret (Theorem 4).

To the best of our knowledge, existing regret analysis on Bernoulli bandits or bandits with bounded support have regret bounds of the form

$$\mathrm{Reg}(T) \leq T\Delta + \sum_{a:\Delta_a > \Delta} \frac{\Delta_a \ln(T)}{\mathsf{kl}(\mu_a + c\Delta_a, \mu_1 - c\Delta_a)} + O\left( \sum_{a:\Delta_a > \Delta} \frac{1}{c^2 \Delta_a} \right),$$

for some $c > 0$, where the third term is much larger than its counterpart given by Theorem 1 when $\Delta_a$ and $\dot{\mu}_1$ are small. As we will see shortly, as a consequence of its tighter bounds, our regret theorem yields a superior worst-case regret guarantee over previous works.

**Theorem 2** (Sub-UCB). *KL-MS's regret is bounded by $\mathrm{Reg}(T) \lesssim \sum_{a:\Delta_a > 0} \frac{\ln T}{\Delta_a}$. Therefore, KL-MS is sub-UCB.*

Sub-UCB criterion is important for measuring a bandit algorithm's finite-time instance-dependent performance. Indeed, Lattimore [28, §3] points out that MOSS [6] does not satisfy sub-UCB and that it leads to a strictly suboptimal regret in a specific instance compared to the standard UCB algorithm [9]. A close inspection of the finite-time regret bounds of existing asymptotically optimal and minimax optimal algorithms for the $[0,1]$-reward setting, such as KL-UCB++ [33] and KL-UCB-switch [18], reveals that they are not sub-UCB. Thus, we speculate that they would also have a suboptimal performance in the aforementioned instance.

In light of Theorem 1, our first corollary is that KL Maillard sampling achieves the following adaptive worst-case regret guarantee.

**Theorem 3** (Adaptive worst-case regret). *For any $K$-arm bandit problem with reward distribution supported on $[0,1]$, KL-MS has regret bounded as: $\mathrm{Reg}(T) \lesssim \sqrt{\dot{\mu}_1 K T \ln K} + K \ln T$.*

An immediate corollary is that KL Maillard sampling has a regret of order $O(\sqrt{KT \ln K})$, which is a factor of $O(\sqrt{\ln K})$ within the minimax optimal regret $\Theta(\sqrt{KT})$ [33, 5]. This also matches the worst-case regret bound $O(\sqrt{VKT \ln(K)})$ of Jin et al. [24] where $V = \frac{1}{4}$ is the worst-case variance for Bernoulli bandits using a Thompson sampling-style algorithm. Another main feature of this regret bound is its adaptivity to $\dot{\mu}_1$, the variance of the reward of the optimal arm for the Bernoulli bandit setting, or its upper bound in the general bounded reward setting (see Lemma 16). Specifically, if $\mu_1$ is close to 0 or 1, $\dot{\mu}_1$ is very small, which results in the regret being much smaller than $O(\sqrt{KT \ln K})$.

Note that UCB-V [7] and KL-UCB/KL-UCB++, while not reported, enjoy a worst-case regret bound of $O(\sqrt{\dot{\mu}_1 KT \ln T})$, which is worse than our bound in its logarithmic factor; see Appendix F.3 and F.1 for the proofs. Among these, UCB-V does not achieve the asymptotic optimality for the Bernoulli case. While logistic linear bandits [1, 32] can be applied to Bernoulli $K$-armed bandits and achieve similar worst-case regret bounds involving $\dot{\mu}_1$, their lower order term can be much worse as discussed in Section 3.

Our second corollary is that KL Maillard sampling achieves a tight asymptotic regret guarantee for the special case of Bernoulli rewards:

**Theorem 4.** *(Asymptotic Optimality) For any $K$-arm bandit problem with reward distribution supported on $[0, 1]$, KL-MS satisfies the following asymptotic regret upper bound:*

$$\limsup_{T \to \infty} \frac{\text{Reg}(T)}{\ln(T)} = \sum_{a \in [K]: \Delta_a > 0} \frac{\Delta_a}{\text{kl}(\mu_a, \mu_1)} \tag{5}$$

Specialized to the Bernoulli bandit setting, in light of the asymptotic lower bounds [27, 12], the above asymptotic regret upper bound implies that KL-MS is asymptotically optimal.

While the regret guarantee of KL-MS is not asymptotically optimal for the general $[0, 1]$ bounded reward setting, it nevertheless is a better regret guarantee than naively viewing this problem as a sub-Gaussian bandit problem and applying sub-Gaussian bandit algorithms on it. To see this, note that any reward distribution supported on $[0, 1]$ is $\frac{1}{4}$-sub-Gaussian; therefore, standard sub-Gaussian bandit algorithms will yield an asymptotic regret $(1 + o(1)) \sum_{a \in [K]: \Delta_a > 0} \frac{\ln T}{2\Delta_a}$. This is always no better than the asymptotic regret provided by Eq. (5), in view of Pinsker's inequality that $\text{kl}(\mu_a, \mu_1) \geq 2\Delta_a^2$.

## 5 Proof Sketch of Theorem 1

We provide an outline of our proof of Theorem 1, with full proof details deferred to Appendix C. Our approach is akin to the recent analysis of the sub-Gaussian Maillard Sampling algorithm in Bian and Jun [11] with several refinements tailored to the bounded reward setting and achieving $\sqrt{\ln K}$ minimax ratio. First, for any time horizon length $T$, $\text{Reg}(T)$ can be bounded by:

$$\text{Reg}(T) = \sum_{a \in [K]: \Delta_a > 0} \Delta_a \mathbb{E}\left[N_{T,a}\right] \leq \Delta T + \sum_{a \in [K]: \Delta_a > \Delta} \Delta_a \mathbb{E}\left[N_{T,a}\right], \tag{6}$$

i.e., the total regret can be decomposed to a $T\Delta$ term and the sum of regret $\Delta_a \mathbb{E}\left[N_{T,a}\right]$ from pulling $\Delta$-suboptimal arms $a$. Therefore, in subsequent analysis, we focus on bounding $\mathbb{E}\left[N_{T,a}\right]$. To this end, we show the following lemma.

**Lemma 5.** *For any suboptimal arm $a$, let $\varepsilon_1, \varepsilon_2 > 0$ be such that $\varepsilon_1 + \varepsilon_2 < \Delta_a$. Then its expected number of pulls is bounded as:*

$$\mathbb{E}\left[N_{T,a}\right] \leq 1 + \frac{\ln\left(T\text{kl}(\mu_a + \varepsilon_1, \mu_1 - \varepsilon_2) \vee e^2\right)}{\text{kl}(\mu_a + \varepsilon_1, \mu_1 - \varepsilon_2)} + \frac{1}{\text{kl}(\mu_a + \varepsilon_1, \mu_1 - \varepsilon_2)} + \frac{1}{\text{kl}(\mu_a + \varepsilon_1, \mu_a)}$$

$$+ 6H \ln\left(\left(\frac{T}{H} \wedge H\right) \vee e^2\right) + \frac{4}{\text{kl}(\mu_1 - \varepsilon_2, \mu_1)}, \tag{7}$$

*where $H := \frac{1}{(1 - \mu_1 + \varepsilon_2)(\mu_1 - \varepsilon_2)h^2(\mu_1, \varepsilon_2)} \lesssim \frac{2\dot{\mu}_1 + \varepsilon_2}{\varepsilon_2^2}$ and $h(\mu_1, \varepsilon_2) := \ln\left(\frac{(1 - \mu_1 + \varepsilon_2)\mu_1}{(1 - \mu_1)(\mu_1 - \varepsilon_2)}\right)$.*

Theorem 1 follows immediately from Lemma 5. See section C for details; we show a sketch here.

*Proof sketch of Theorem 1.* Fix any $c \in (0, \frac{1}{4}]$. Let $\varepsilon_1 = \varepsilon_2 = c\Delta_a$; by the choice of $c$, $\varepsilon_1 + \varepsilon_2 < \Delta_a$. From Lemma 5, $\mathbb{E}\left[N_{T,a}\right]$ is bounded by Eq. (7). Plugging in the values of $\varepsilon_1 = \varepsilon_2$, and using Lemma 26 that lower bounds the binary KL divergence, along with Lemma 22 that gives $H \lesssim \frac{2\dot{\mu}_1 + \varepsilon_2}{\varepsilon_2^2}$, and algebra, all terms except the second term on the right hand side of Eq. (7) are bounded by

$$\left(\frac{34}{c^2} + \frac{4}{(1 - 2c)^2}\right)\left(\frac{\dot{\mu}_1 + \Delta_a}{c^2\Delta_a^2}\right) \ln\left(\left(\frac{\dot{\mu}_1 + \Delta_a}{c^2\Delta_a^2} \wedge \frac{c^2 T\Delta_a^2}{\dot{\mu}_1 + \Delta_a}\right) \vee e^2\right).$$

As a result, KL-MS satisfies that, for any arm $a$, for any $c \in (0, \frac{1}{4}]$:

$$\mathbb{E}\left[N_{T,a}\right] \leq \frac{\ln(T\text{kl}(\mu_a + c\Delta_a, \mu_1 - c\Delta_a) \vee e^2)}{\text{kl}(\mu_a + c\Delta_a, \mu_1 - c\Delta_a)}$$

$$+ \left(\frac{34}{c^2} + \frac{4}{(1 - 2c)^2}\right)\left(\frac{\dot{\mu}_1 + \Delta_a}{c^2\Delta_a^2}\right) \ln\left(\left(\frac{\dot{\mu}_1 + \Delta_a}{c^2\Delta_a^2} \wedge \frac{c^2 T\Delta_a^2}{\dot{\mu}_1 + \Delta_a}\right) \vee e^2\right).$$

Theorem 1 follows by plugging the above bound to Eq. (6) for arms $a$ s.t. $\Delta_a > \Delta$ with $c = \frac{1}{4}$. $\quad\square$

## 5.1 Proof sketch of Lemma 5

We sketch the proof of Lemma 5 in this subsection. For full details of the proof, please refer to Appendix C.2. We first set up some useful notations that will be used throughout the proof. Let $u := \lceil \frac{\ln(T\mathsf{kl}(\mu_a+\varepsilon_1,\mu_1-\varepsilon_2)\vee e^2)}{\mathsf{kl}(\mu_a+\varepsilon_1,\mu_1-\varepsilon_2)} \rceil$. We define the following events

$$A_t := \{I_t = a\}, \quad B_t := \{N_{t,a} < u\}, \quad C_t := \{\hat{\mu}_{t,\max} \geq \mu_1 - \varepsilon_2\}, \quad D_t := \{\hat{\mu}_{t,a} \leq \mu_a + \varepsilon_1\},$$

By algebra, one has the following elementary upper bound on $\mathbb{E}[N_{T,a}]$: $\mathbb{E}[N_{T,a}] \leq u + \mathbb{E}[\sum_{t=K+1}^{T} \mathbf{1}\{A_t, B_{t-1}^c\}]$. Intuitively, the $u$ term serves to control the length of a "burn-in" phase when the number of pulls to arm $a$ is at most $u$. It now remains to control the second term, the number of pulls to arm $a$ after it is large enough, i.e., $N_{t-1,a} \geq u$. We decompose it to $F1$, $F2$, and $F3$, resulting in the following inequality:

$$\mathbb{E}[N_{T,a}] \leq u + \underbrace{\mathbb{E}\left[\sum_{t=K+1}^{T} \mathbf{1}\{A_t, B_{t-1}^c, C_{t-1}, D_{t-1}\}\right]}_{=:F1}$$

$$+ \underbrace{\mathbb{E}\left[\sum_{t=K+1}^{T} \mathbf{1}\{A_t, B_{t-1}^c, C_{t-1}, D_{t-1}^c,\}\right]}_{=:F2} + \underbrace{\mathbb{E}\left[\sum_{t=K+1}^{T} \mathbf{1}\{A_t, B_{t-1}^c, C_{t-1}^c\}\right]}_{=:F3}$$

Here:

- $F1$ corresponds to the "steady state" when the empirical means of arm $a$ and the optimal arm are both estimated accurately, i.e., $\hat{\mu}_{t-1,\max} \geq \mu_1 - \varepsilon_2$ and $\hat{\mu}_{t-1,a} \leq \mu_a + \varepsilon_1$. It can be straightforwardly bounded by $\frac{1}{\mathsf{kl}(\mu_a+\varepsilon_1,\mu_1-\varepsilon_2)}$, as we show in Lemma 10 (section D.1).

- $F2$ corresponds to the case when the empirical mean of arm $a$ is abnormally high, i.e., $\hat{\mu}_{t-1,a} > \mu_a + \varepsilon_1$. It can be straightforwardly bounded by $\frac{1}{\mathsf{kl}(\mu_a+\varepsilon_1,\mu_a)}$, as we show in Lemma 11 (section D.2).

- $F3$ corresponds to the case when the empirical mean of the optimal arm is abnormally low, i.e., $\hat{\mu}_{t-1,\max} \leq \mu_1 - \varepsilon_2$; it is the most challenging term and we discuss our techniques in bounding it in detail below (section D.3).

We provide an outline of our analysis of $F3$ in Appendix D.3.1 and sketch its main ideas and technical challenges here.

We follow the derivation from Bian and Jun [11] by first using a probability transferring argument (Lemma 23) to bound the expected counts of pulling suboptimal arm $a$ by the expectation of indicators of pulling the optimal arm with a multiplicative factor and then change the counting from global time step $t$ to local count of pulling the optimal arm. Then, $F3$ is bounded by,

$$\sum_{k=1}^{\infty} \underbrace{\mathbb{E}\left[\mathbf{1}\{\hat{\mu}_{(k),1} \leq \mu_1 - \varepsilon_2\} \exp(k \cdot \mathsf{kl}(\hat{\mu}_{(k),1}, \mu_1 - \varepsilon_2))\right]}_{M_k}.$$

Intuitively, each $M_k$ should be controlled: when $\exp(k \cdot \mathsf{kl}(\hat{\mu}_{(k),1}, \mu_1 - \varepsilon_2))$ is large, $\hat{\mu}_{(k),1}$ must significantly negatively deviate from $\mu_1 - \varepsilon_2$, which happens with low probability by Chernoff bound (Lemma 25). Using a double integration argument, we can bound each $M_k$ by

$$M_k \leq \left(\frac{2H}{k} + 1\right) \exp(-k\mathsf{kl}(\mu_1 - \varepsilon_2, \mu_1)).$$

Summing over all $k$, we can bound $F3_1$ by $O\left(H \ln(H \vee e^2) + \frac{1}{\mathsf{kl}(\mu_1-\varepsilon_2,\mu_1)}\right)$. Combining the bounds on $F1$ and $F2$, we can show a bound on $\mathbb{E}[N_{T,a}]$ similar to Eq. (7) without the "$\frac{T}{H}\wedge$" term in the logarithmic factor. This yields a regret bound of KL-MS, in the form of Eq. (4) without the "$\wedge \frac{c^2 T \Delta_a^2}{\hat{\mu}_1 + \Delta_a}$" term in the logarithmic factor. Such a regret bound can be readily used to show KL-MS's

Bernoulli asympototic optimality and sub-UCB property. An adaptive worst-case regret bound of $\sqrt{\dot{\mu}_1 KT \ln(T)}$ also follows immediately.

To show that MS has a tighter adaptive worst-case regret bound of $\sqrt{\dot{\mu}_1 KT \ln(K)}$, we adopt a technique in [33, 24]. First, we observe that the looseness of the above bound on $F3$ comes from small $k$ (denoted as $F3_1 := \sum_{k \leq H} M_k$), as the summation of $M_k$ for large $k$ (denoted as $F3_2 := \sum_{k>H} M_k$) is well-controlled. The key challenge in a better control of $F3_1$ comes from the difficulty in bounding the tail probability of $\hat{\mu}_{(k),1}$ for $k < H$ beyond Chernoff bound. To cope with this, we observe that a modified version of $F3_1$ that contains an extra favorable indicator of $\mathsf{kl}(\hat{\mu}_{(k),1}, \mu_1) \leq \frac{2\ln(T/k)}{k}$, denoted as:

$$\sum_{k \leq H} \mathbb{E}\left[\mathbf{1}\left\{\hat{\mu}_{(k),1} \leq \mu_1 - \varepsilon_2, \mathsf{kl}(\hat{\mu}_{(k),1}, \mu_1) \leq \frac{2\ln(T/k)}{k}\right\} \exp(k \cdot \mathsf{kl}(\hat{\mu}_{(k),1}, \mu_1 - \varepsilon_2))\right]$$

can be well-controlled. Utilizing this introduces another term in the regret analysis, $T \cdot \mathbb{P}(\mathcal{E}^C)$, where $\mathcal{E} = \left\{\forall k \in [1, H], \mathsf{kl}(\hat{\mu}_{(k),1}, \mu_1) \leq \frac{2\ln(T/k)}{k}\right\}$, which we bound by $O(H)$ via a time-uniform version of Chernoff bound. Putting everything together, we prove a bound of $F3$ of $O\left(H \ln\left(\left(\frac{T}{H} \wedge H\right) \vee e^2\right) + \frac{1}{\mathsf{kl}(\mu_1 - \varepsilon_2, \mu_1)}\right)$, which yields our final regret bound of KL-MS in Theorem 1 and the refined minimax ratio.

**Remark 6.** *Although our technique is inspired by [24, 33], we carefully set the case splitting threshold for $N_{t-1,1}$ (to obtain $F3_1$ and $F3_2$) to be $H = O(\frac{\dot{\mu}_1 + \epsilon_2}{\epsilon_2^2})$, which is significantly different from prior works $(\tilde{O}(\frac{1}{\epsilon_2^2}))$.*

**Remark 7.** *One can port our proof strategy back to sub-Gaussian MS and show that it achieves a minimax ratio of $\sqrt{\ln K}$ as opposed to $\sqrt{\ln T}$ reported in Bian and Jun [11]; a sketch of the proof is in Appendix G. Recall that Bian and Jun [11] proposed another algorithm $MS^+$ that achieved the minimax ratio of $\sqrt{\ln K}$ at the price of extra exploration. Our result makes $MS^+$ obsolete; MS should be preferred over $MS^+$ at all times.*

## 6 Conclusion

We have proposed KL-MS, a KL version of Maillard sampling for stochastic multi-armed bandits in the $[0, 1]$-bounded reward setting, with a closed-form probability computation, which is highly amenable to off-policy evaluation. Our algorithm requires constant time complexity with respect to the target numerical precision in computing the action probabilities, and our regret analysis shows that KL-MS achieves the best regret bound among those in the literature that allows computing the action probabilities with $O(\text{polylog}(1/\text{precision}))$ time complexity, for example, Tsallis-INF [43], EXP3++ [40], in the stochastic setting.

Our study opens up numerous open problems. One immediate open problem is to generalize KL-MS to handle exponential family reward distributions. Another exciting direction is to design randomized and off-policy-amenable algorithms that achieve the asymptotic optimality for bounded rewards (i.e., as good as IMED [21]).

One possible avenue is to extend MED [19] and remove the restriction that the reward distribution must have bounded support. Furthermore, it would be interesting to extend MS to structured bandits and find connections to the Decision-Estimation Coefficient [16], which have recently been reported to characterize the optimal minimax regret rate for structured bandits. Finally, we believe MS is practical by incorporating the booster hyperparameter introduced in Bian and Jun [11]. Extensive empirical evaluations on real-world problems would be an interesting future research direction.

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
