**Acknowledgments.** We thank Kyoungseok Jang for helpful discussions on refinements of binary Pinsker's inequality (Lemma 26). Hao Qin and Chicheng Zhang gratefully acknowledge funding support from University of Arizona FY23 Eighteenth Mile TRIF Funding.

## A    Proof of Worst-case Regret Bounds (Theorem 3) and Sub-UCB Property (Theorem 2)

Before proving Theorem 3, we first state and prove a useful lemma that gives us an upper bound to the regret, which is useful for subsequent minimax ratio analysis and asymptotic analysis. This regret bound consists of two components, which correspond to arms with suboptimality gaps at most or greater than a predetermined threshold $\Delta$ respectively . The former is bounded by $T\Delta$, while the latter is upper bounded by a finer $\tilde{O}(\sum_{a:\Delta_a > \Delta} \frac{\dot{\mu}_1}{\Delta_a} + K)$ term.

**Lemma 8.** *For KL-MS, its regret is bounded by: for any $\Delta \geq 0$,*

$$\mathrm{Reg}(T) \leq T\Delta + O\left( \sum_{a:\Delta_a > \Delta} \left( \frac{\dot{\mu}_1 + \Delta_a}{\Delta_a} \right) \ln\left( \frac{T\Delta_a^2}{\dot{\mu}_1 + \Delta_a} \vee e^2 \right) \right)$$

*Proof.* Applying Theorem 1 with $c = \frac{1}{4}$, we have:

$$\mathrm{Reg}(T)$$

$$\leq T\Delta + \sum_{a:\Delta_a > \Delta} \frac{\Delta_a \ln(T\mathsf{kl}(\mu_a + c\Delta_a, \mu_1 - c\Delta_a) \vee e^2)}{\mathsf{kl}(\mu_a + c\Delta_a, \mu_1 - c\Delta_a)}$$

$$+ 560\left( \sum_{a:\Delta_a > \Delta} \left( \frac{\dot{\mu}_1 + \Delta_a}{c^4 \Delta_a} \right) \ln\left( \left( \left( \frac{\dot{\mu}_1 + \Delta_a}{\Delta_a^2} \wedge \frac{T\Delta_a^2}{\dot{\mu}_1 + \Delta_a} \right) \vee e^2 \right) \right) \right) \qquad \text{(Theorem 1)}$$

$$\leq T\Delta + O\left( \sum_{a:\Delta_a > \Delta} \left( \frac{\dot{\mu}_1 + \Delta_a}{\Delta_a} \right) \ln\left( \frac{T\Delta_a^2}{\dot{\mu}_1 + \Delta_a} \vee e^2 \right) \right), \qquad \text{(Lemma 27 and Lemma 26)}$$

here, the second inequality is because we choose $\frac{T\Delta_a^2}{\dot{\mu}_1 + \Delta_a}$ as the upper bound in the lower order term then we use Lemma 26 to lower bound $\mathsf{kl}(\mu_a + c\Delta_a, \mu_1 - c\Delta_a) \gtrsim \frac{\Delta_a^2}{\dot{\mu}_a + \Delta_a}$ and Lemma 27 that $x \mapsto \frac{\ln(Tx \vee e^2)}{x}$ is monotonically decreasing when $x \geq 0$. Also by 1-Lipshitzness of $z \mapsto z(1 - z)$, we have $(\mu_1 - c\Delta_a)(1 - (\mu_1 - c\Delta_a)) \leq \dot{\mu}_1 + c\Delta_a$ and all terms except $T\Delta$ will be merged into the $O(\cdot)$ term. $\qquad \square$

*Proof of Theorem 3.* Let $\Delta = \sqrt{\frac{\dot{\mu}_1 K \ln K}{T}}$, from Lemma 8 we have

$$\mathrm{Reg}(T) \leq T\Delta + O\left( \sum_{a:\Delta_a > \Delta} \left( \frac{\dot{\mu}_1 + \Delta_a}{\Delta_a} \right) \ln\left( \frac{T\Delta_a^2}{\dot{\mu}_1 + \Delta_a} \vee e^2 \right) \right)$$

$$\leq T\Delta + O\left( \sum_{a:\Delta_a > \Delta} \frac{\dot{\mu}_1}{\Delta_a} \ln\left( \frac{T\Delta_a^2}{\dot{\mu}_1} \vee e^2 \right) \right) + O\left( \sum_{a:\Delta_a > \Delta} \ln\left( (T\Delta_a) \vee e^2 \right) \right)$$

$$\leq T\Delta + O\left( \frac{K\dot{\mu}_1}{\Delta} \ln\left( \frac{T\Delta^2}{\dot{\mu}_1} \vee e^2 \right) \right) + O\left( K \ln(T) \right) \qquad \text{(Lemma 27)}$$

$$\leq O\left( \sqrt{\dot{\mu}_1 K T \ln K} \right) + O\left( K \ln(T) \right),$$

where in the second inequality, we split fraction $\frac{\dot{\mu}_1+\Delta_a}{\Delta_a}$ into $\frac{\dot{\mu}_1}{\Delta_a}$ and $1$, then bound each term separately. The second-to-last inequality is due to the monotonicity of $x \mapsto \frac{\ln\,(bx^2\vee e^2)}{x}$ proven in Lemma 27; The last inequality is by algebra. $\qquad\square$

*Proof of Theorem 2.* This is an immediate consequence of Lemma 8 with $\Delta = 0$, along with the observations that $\frac{\dot{\mu}_1+\Delta_a}{\Delta_a} \le \frac{2}{\Delta_a}$, and $\frac{T\Delta_a^2}{\dot{\mu}_1+\Delta_a} \le T$. $\qquad\square$

# B  Proof of Asymptotic Optimality (Theorem 4)

We establish asymptotic optimality of KL-MS by analyzing the ratio between the expected regret to $\ln T$ and letting $T \to \infty$.

*Proof.* Starting from Theorem 1 and letting $\Delta = 0$ and $c = \frac{1}{\ln\ln T}$:

$$\limsup_{T\to\infty} \frac{\mathrm{Reg}(T)}{\ln(T)}$$

$$\le \lim_{T\to\infty} \sum_{a\in[K]:\Delta_a>0} \frac{\Delta_a \ln(T\mathsf{kl}(\mu_a + c\Delta_a, \mu_1 - c\Delta_a) \vee e^2)}{\ln T\mathsf{kl}(\mu_a + c\Delta_a, \mu_1 - c\Delta_a)}$$

$$+ \lim_{T\to\infty} 560 \left( \sum_{a\in[K]:\Delta_a>0} \left( \frac{\dot{\mu}_1 + \Delta_a}{c^4 \ln T\Delta_a} \right) \ln\left( \left( \frac{\dot{\mu}_1 + \Delta_a}{\Delta_a^2} \right) \vee e^2 \right) \right) \qquad \text{(Theorem 1)}$$

$$\le \lim_{T\to\infty} \sum_{a\in[K]:\Delta_a>0} \frac{\Delta_a \ln(T\mathsf{kl}(\mu_a + c\Delta_a, \mu_1 - c\Delta_a) \vee e^2)}{\mathsf{kl}(\mu_a + c\Delta_a, \mu_1 - c\Delta_a) \ln T}$$

$$= \lim_{T\to\infty} \sum_{a\in[K]:\Delta_a>0} \frac{\Delta_a \ln(T\mathsf{kl}(\mu_a + c\Delta_a, \mu_1 - c\Delta_a) \vee e^2)}{\mathsf{kl}(\mu_a, \mu_1) \ln T} \cdot \frac{\mathsf{kl}(\mu_a, \mu_1)}{\mathsf{kl}(\mu_a + c\Delta_a, \mu_1 - c\Delta_a)}$$

$$= \sum_{a\in[K]:\Delta_a>0} \frac{\Delta_a}{\mathsf{kl}(\mu_a, \mu_1)}, \qquad\qquad\qquad \text{(By the continuity of } \mathsf{kl}(\cdot,\cdot))$$

where the first inequality is because of the fact that $\ln\left( \left( \frac{\dot{\mu}_1+\Delta_a}{c^2\Delta_a^2} \wedge \frac{c^2 T}{\frac{\dot{\mu}_1+\Delta_a}{\Delta_a^2}} \right) \vee e^2 \right) \le$

$\frac{1}{c^2} \ln\left( \frac{\dot{\mu}_1+\Delta_a}{\Delta_a^2} \vee e^2 \right)$ due to Lemma 28, and, the second inequality is due to that when $T \to \infty$, $c^4 \ln T = \frac{\ln T}{(\ln\ln T)^4} \to \infty$. $\qquad\square$

# C  Full Proof of Theorem 1

## C.1  A general lemma on the expected arm pulls and its implication to Theorem 1

We first present a general lemma that bounds the number of pulls to arm $a$ by KL-MS; due to its technical nature, we defer its proof to Section C.2 and focus on its implication to Theorem 1 in this section.

**Lemma 9** (Lemma 5 restated). *For any suboptimal arm $a$, let $\varepsilon_1, \varepsilon_2 > 0$ be such that $\varepsilon_1 + \varepsilon_2 < \Delta_a$. Then its expected number of pulls is bounded as:*

$$\mathbb{E}\left[N_{T,a}\right] \le 1 + \frac{\ln\left(T\mathsf{kl}(\mu_a + \varepsilon_1, \mu_1 - \varepsilon_2) \vee e^2\right)}{\mathsf{kl}(\mu_a + \varepsilon_1, \mu_1 - \varepsilon_2)} + \frac{1}{\mathsf{kl}(\mu_a + \varepsilon_1, \mu_1 - \varepsilon_2)} + \frac{1}{\mathsf{kl}(\mu_a + \varepsilon_1, \mu_a)} \quad (8)$$

$$+ 6H \ln\left( \left( \frac{T}{H} \wedge H \right) \vee e^2 \right) + \frac{4}{\mathsf{kl}(\mu_1 - \varepsilon_2, \mu_1)}, \quad (9)$$

*where $H := \frac{1}{(1-\mu_1+\varepsilon_2)(\mu_1-\varepsilon_2)h^2(\mu_1,\varepsilon_2)}$ and $h(\mu_1, \varepsilon_2) := \ln\left( \frac{(1-\mu_1+\varepsilon_2)\mu_1}{(1-\mu_1)(\mu_1-\varepsilon_2)} \right)$.*

We now use Lemma 9 to conclude Theorem 1.

*Proof of Theorem 1.* Fix any $c \in (0, \frac{1}{4}]$. Let $\varepsilon_1 = \varepsilon_2 = c\Delta_a$; note that by the choice of $c$, $\varepsilon_1 + \varepsilon_2 < \Delta_a$. From Lemma 9, $\mathbb{E}\left[N_{T,a}\right]$ is bounded by Eq. (9). We now plug in the value of $\varepsilon_1, \varepsilon_2$, and further upper bound the third to the sixth terms of the right hand side of Eq. (9):

- 
$$\frac{1}{\mathsf{kl}(\mu_a + \varepsilon_1, \mu_1 - \varepsilon_2)} \leq \frac{4(\mu_1 - \varepsilon_2)(1 - \mu_1 + \varepsilon_2) + 4(\Delta_a - \varepsilon_1 - \varepsilon_2)}{(\Delta_a - \varepsilon_1 - \varepsilon_2)^2} \leq \frac{4}{(1-2c)^2} \cdot \frac{\dot{\mu}_1 + \Delta_a}{\Delta_a^2}$$

- 
$$\frac{1}{\mathsf{kl}(\mu_a + \varepsilon_1, \mu_a)} \leq \frac{4\dot{\mu}_a + 4\varepsilon_1}{\varepsilon_1^2} \leq \frac{6}{c^2} \cdot \frac{\dot{\mu}_1 + \Delta_a}{\Delta_a^2}$$

- 
$$\frac{4}{\mathsf{kl}(\mu_1 - \varepsilon_2, \mu_1)} \leq \frac{16\dot{\mu}_1 + 16\varepsilon_2}{\varepsilon_2^2} \leq \frac{16}{c^2} \cdot \frac{\dot{\mu}_1 + \Delta_a}{\Delta_a^2}$$

- By Lemma 22, $H \leq \frac{2\dot{\mu}_1 + 2\varepsilon_2}{\varepsilon_2^2} \leq \frac{2}{c^2} \cdot \frac{\dot{\mu}_1 + \Delta_a}{\Delta_a^2}$, and by Lemma 27, the function $H \mapsto 6H \ln\left(\left(\frac{T}{H} \wedge H\right) \vee e^2\right)$ is monotonically increasing, we have that

$$6H \ln\left(\left(\frac{T}{H} \wedge H\right) \vee e^2\right) \leq \frac{12}{c^2} \cdot \left(\frac{\dot{\mu}_1 + \Delta_a}{c^2 \Delta_a^2}\right) \ln\left(\left(\frac{\dot{\mu}_1 + \Delta_a}{c^2 \Delta_a^2} \wedge \frac{c^2 T \Delta_a^2}{\dot{\mu}_1 + \Delta_a}\right) \vee e^2\right)$$

Combining all the above bounds and Eq. (9), KL-MS satisfies that, for any arm $a$, for any $c \in (0, \frac{1}{4}]$:

$$\mathbb{E}\left[N_{T,a}\right] \leq \frac{\ln(T\mathsf{kl}(\mu_a + c\Delta_a, \mu_1 - c\Delta_a) \vee e^2)}{\mathsf{kl}(\mu_a + c\Delta_a, \mu_1 - c\Delta_a)}$$
$$+ \left(\frac{34}{c^2} + \frac{4}{(1-2c)^2}\right)\left(\frac{\dot{\mu}_1 + \Delta_a}{c^2 \Delta_a^2}\right) \ln\left(\left(\frac{\dot{\mu}_1 + \Delta_a}{c^2 \Delta_a^2} \wedge \frac{c^2 T \Delta_a^2}{\dot{\mu}_1 + \Delta_a}\right) \vee e^2\right) \quad (10)$$

For any $\Delta \geq 0$, we now bound the pseudo-regret of KL-MS as follows:

$$\text{Reg}(T)$$
$$= \sum_{a:\Delta_a > 0} \Delta_a \mathbb{E}\left[N_{T,a}\right]$$
$$= \sum_{a:\Delta_a \in (0,\Delta]} \Delta_a \mathbb{E}\left[N_{T,a}\right] + \sum_{a:\Delta_a > \Delta} \Delta_a \mathbb{E}\left[N_{T,a}\right]$$
$$\leq \Delta T + \sum_{a:\Delta_a > \Delta} \Delta_a \frac{\ln(T\mathsf{kl}(\mu_a + c\Delta_a, \mu_1 - c\Delta_a) \vee e^2)}{\mathsf{kl}(\mu_a + c\Delta_a, \mu_1 - c\Delta_a)}$$
$$+ \left(\frac{34}{c^2} + \frac{4}{(1-2c)^2}\right) \sum_{a:\Delta_a > \Delta} \left(\frac{\dot{\mu}_1 + \Delta_a}{c^2 \Delta_a}\right) \ln\left(\left(\frac{\dot{\mu}_1 + \Delta_a}{c^2 \Delta_a^2} \wedge \frac{c^2 T \Delta_a^2}{\dot{\mu}_1 + \Delta_a}\right) \vee e^2\right),$$

where the last inequality is from Eq. (10). Then we pick $c = \frac{1}{4}$ and conclude the proof of the theorem. $\square$

## C.2 Proof of Lemma 9: arm pull count decomposition and additional notations

In this subsection, we prove Lemma 9. We first recall the following set of useful notations defined in Section 5:

Recall that $u = \lceil \frac{\ln\left(T\mathsf{kl}(\mu_a + \varepsilon_1, \mu_1 - \varepsilon_2) \vee e^2\right)}{\mathsf{kl}(\mu_a + \varepsilon_1, \mu_1 - \varepsilon_2)} \rceil$, and we have defined the following events

$$A_t := \{I_t = a\}$$

$$B_t := \{N_{t,a} < u\}$$
$$C_t := \{\hat{\mu}_{t,\max} \geq \mu_1 - \varepsilon_2\}$$
$$D_t := \{\hat{\mu}_{t,a} \leq \mu_a + \varepsilon_1\}$$

**A useful decomposition of the expected number of pulls to arm $a$.** With the notations above, we bound the expected number of pulling any suboptimal $a$ by decomposing the arm pull indicator $\mathbf{1}\{I_t = a\}$ according to events $B_{t-1}, C_{t-1}^c$ and $D_{t-1}$ in a cascading manner:

$$\mathbb{E}[N_{T,a}] = \mathbb{E}\left[\sum_{t=1}^{T} \mathbf{1}\{I_t = a\}\right] \tag{11}$$

$$= 1 + \mathbb{E}\left[\sum_{t=K+1}^{T} \mathbf{1}\{A_t\}\right] \qquad \text{(Definition of Algorithm 1)}$$

$$= 1 + \mathbb{E}\left[\sum_{t=K+1}^{T} \mathbf{1}\{A_t, B_{t-1}\}\right] + \mathbb{E}\left[\sum_{t=K+1}^{T} \mathbf{1}\{A_t, B_{t-1}^c\}\right] \tag{12}$$

$$\leq 1 + (u - 1) + \mathbb{E}\left[\sum_{t=K+1}^{T} \mathbf{1}\{A_t, B_{t-1}^c\}\right] \qquad \text{(Lemma 18)}$$

$$= u + \underbrace{\mathbb{E}\left[\sum_{t=K+1}^{T} \mathbf{1}\{A_t, B_{t-1}^c, C_{t-1}, D_{t-1}\}\right]}_{F1} \tag{13}$$

$$+ \underbrace{\mathbb{E}\left[\sum_{t=K+1}^{T} \mathbf{1}\{A_t, B_{t-1}^c, C_{t-1}, D_{t-1}^c\}\right]}_{F2} \tag{14}$$

$$+ \underbrace{\mathbb{E}\left[\sum_{t=K+1}^{T} \mathbf{1}\{A_t, B_{t-1}^c, C_{t-1}^c\}\right]}_{F3} \tag{15}$$

Given the above decomposition, the lemma is now an immediate consequence of the definition of $u$, Lemmas 10, 11 and 12 (that bounds $F1, F2, F3$ respectively), which we state and prove in Appendix D.

# D Bounding the number of arm pulls in each case

## D.1 F1

In this section we bound $F1$. This is the case that $\hat{\mu}_{t,a}$ is small and $\hat{\mu}_{t,\max}$ is large, so that $\mathsf{kl}(\hat{\mu}_{t,a}, \hat{\mu}_{t,\max})$ do not significantly underestimate $\mathsf{kl}(\mu_a, \mu_1)$, which will imply that suboptimal arm $a$ will be only pulled a small number of times due to the arm selection rule (Eq. (3)). Note that $u$ is set carefully so that $F1$ is bounded just enough to be lower than the $\frac{\ln T}{\mathsf{kl}(\mu_a, \mu_1)}$ Bernoulli asymptotic lower bound.

**Lemma 10.**

$$F1 \leq \frac{1}{\mathsf{kl}(\mu_a + \varepsilon_1, \mu_1 - \varepsilon_2)}$$

*Proof.* Recall the notations that $A_t = \{I_t = a\}$, $B_{t-1}^c = \{N_{t-1,a} \geq u\}$, $C_{t-1} = \{\hat{\mu}_{t-1,\max} \geq \mu_1 - \varepsilon_2\}$, $D_{t-1} = \{\hat{\mu}_{t-1,a} \leq \mu_a + \varepsilon_1\}$. We have:

$$F1 = \mathbb{E}\left[\sum_{t=K+1}^{T} \mathbf{1}\left\{A_t, B_{t-1}^c, C_{t-1}, D_{t-1}\right\}\right] \tag{16}$$

$$= \sum_{t=K+1}^{T} \mathbb{E}\left[\mathbb{E}\left[\mathbf{1}\left\{A_t, B_{t-1}^c, C_{t-1}, D_{t-1}\right\} \mid \mathcal{H}_{t-1}\right]\right] \quad \text{(Law of total expectation)}$$

$$= \sum_{t=K+1}^{T} \mathbb{E}\left[\mathbf{1}\left\{B_{t-1}^c, C_{t-1}, D_{t-1}\right\} \mathbb{E}\left[\mathbf{1}\left\{A_t\right\} \mid \mathcal{H}_{t-1}\right]\right]$$

$$(B_{t-1}, C_{t-1}, D_{t-1} \text{ are } \mathcal{H}_{t-1}\text{-measurable})$$

$$\leq \sum_{t=K+1}^{T} \mathbb{E}\left[\mathbf{1}\left\{B_{t-1}^c, C_{t-1}, D_{t-1}\right\} \exp(-N_{t-1,a}\mathsf{kl}(\hat{\mu}_{t-1,a}, \hat{\mu}_{t-1,\max}))\right] \quad \text{(By Lemma 23)}$$

$$\leq \sum_{t=K+1}^{T} \mathbb{E}\left[\mathbf{1}\left\{B_{t-1}^c\right\} \exp(-u \cdot \mathsf{kl}(\mu_a + \varepsilon_1, \mu_1 - \varepsilon_2))\right]$$

(Based on $B_{t-1}^c, C_{t-1}$ and $D_{t-1}$, there is $N_{t-1,a} \geq u$ and $\mathsf{kl}\left(\hat{\mu}_{t-1,a}, \hat{\mu}_{t-1,\max}\right) \geq \mathsf{kl}\left(\mu_a + \varepsilon_1, \mu_1 - \varepsilon_2\right)$)

$$\leq T \cdot \exp(-u \cdot \mathsf{kl}(\mu_a + \varepsilon_1, \mu_1 - \varepsilon_2)) \tag{$\mathbf{1}\{\cdot\} \leq 1$}$$

$$\leq T \cdot \frac{1}{T\mathsf{kl}(\mu_a + \varepsilon_1, \mu_1 - \varepsilon_2)} \quad \text{(Recall definition of } u\text{)}$$

$$= \frac{1}{\mathsf{kl}(\mu_a + \varepsilon_1, \mu_1 - \varepsilon_2)} \qquad \qquad \square$$

### D.2 F2

In this section we upper bound $F2$. This is the case when the suboptimal arm $a$'s mean reward is overestimated by at least $\varepsilon_2$. Intuitively this should not happen too many times, due to the concentration between the empirical mean reward and the population mean reward of arm $a$.

**Lemma 11.**

$$F2 \leq \frac{1}{\mathsf{kl}(\mu_a + \varepsilon_1, \mu_a)}$$

*Proof.* Recall the notations that $A_t = \{I_t = a\}$, $B_{t-1}^c = \{N_{t-1,a} \geq u\}$, $C_{t-1} = \{\hat{\mu}_{t-1,\max} \geq \mu_1 - \varepsilon_2\}$, $D_{t-1}^c = \{\hat{\mu}_{t-1,a} > \mu_a + \varepsilon_1\}$. We have:

$$F2 = \mathbb{E}\left[\sum_{t=K+1}^{T} \mathbf{1}\left\{A_t, B_{t-1}^c, C_{t-1}, D_{t-1}^c\right\}\right] \tag{17}$$

$$\leq \mathbb{E}\left[\sum_{k=2}^{\infty} \mathbf{1}\left\{B_{\tau_a(k)-1}^c, C_{\tau_a(k)-1}, D_{\tau_a(k)-1}^c\right\}\right]$$

(implies that only when $t = \tau_a(k)$ for some $k \geq 2$ the inner indicator is non-zero)

$$\leq \mathbb{E}\left[\sum_{k=2}^{\infty} \mathbf{1}\left\{D_{\tau_a(k)-1}^c\right\}\right] \quad \text{(Drop unnecessary conditions)}$$

$$= \mathbb{E}\left[\sum_{k=2}^{\infty} \mathbf{1}\left\{D_{\tau_a(k-1)}^c\right\}\right] \qquad (\hat{\mu}_{\tau_a(k)-1,a} = \hat{\mu}_{\tau_a(k-1),a})$$

$$= \mathbb{E}\left[\sum_{k=1}^{\infty} \mathbf{1}\left\{D_{\tau_a(k)}^c\right\}\right] \qquad \text{(shift time index } t\text{)}$$

$$\leq \sum_{k=1}^{\infty} \exp(-k \cdot \mathsf{kl}(\mu_a + \varepsilon_1, \mu_a)) \qquad\qquad \text{(By Lemma 25)}$$

$$\leq \frac{\exp(-\mathsf{kl}(\mu_a + \varepsilon_1, \mu_a))}{1 - \exp(-\mathsf{kl}(\mu_a + \varepsilon, \mu_a))} \qquad\qquad \text{(Geometric sum)}$$

$$\leq \frac{1}{\mathsf{kl}(\mu_a + \varepsilon_1, \mu_a)} \qquad\qquad \text{(Applying inequality } e^x \geq 1 + x \text{ when } x \geq 0)$$

$$(18)$$

Note that in the first inequality, we use the observation that for every $t \geq K+1$ such that $A_t$ happens, there exists a unique $k \geq 2$ such that $t = \tau_a(k)$. The third inequality is due to the Chernoff's inequality (Lemma 25) on the random variable $\hat{\mu}_{\tau_a(k),a} - \mu_a$. Given any $\tau_a(k)$, $\hat{\mu}_{\tau_a(k),a}$ is the running average reward of the first $k$'s pulling of arm $a$. In each pulling of arm $a$ the reward follows a bounded distribution $\nu_a$ with mean $\mu_a$ independently. $\qquad\square$

### D.3 F3

In this section we upper bound $F3$, which counts the expected number of times steps when arm $a$ is pulled while $\hat{\mu}_{t-1,\max}$ underestimates $\mu_1$ by at least $\varepsilon_2$. Our main result of this section is the following lemma:

**Lemma 12.**

$$F3 \leq 6H \ln\left( \left( \frac{T}{H} \wedge H \right) \vee e^2 \right) + \frac{4}{\mathsf{kl}(\mu_1 - \varepsilon_2, \mu_1)},$$

where we recall that $H = \frac{1}{(1 - \mu_1 + \varepsilon_2)(\mu_1 - \varepsilon_2)h^2(\mu_1, \varepsilon_2)}$.

#### D.3.1 Roadmap of analysis

Before proving the lemma, we sketch the key ideas underlying our proof. First, note that by the KL-MS sampling rule (Eq. (3)), at any time step $t$, $p_{t,1}$ should not be too small ($p_{t,1} = \exp(-N_{t-1,1}\mathsf{kl}(\hat{\mu}_{t-1,1}, \hat{\mu}_{t-1,\max}))/M_t$), and as a result, the conditional probability of pulling arm $a$, $p_{t,a}$ should be not much higher than that of arm 1, $p_{t,1}$; using this along with a "probability transfer" argument similar to [4, 11] (see Lemma 23 for a formal statement) tailored to KL-MS sampling rule, we have:

$$F3 \leq \mathbb{E}\left[ \sum_{t=K+1}^{T} \mathbf{1}\left\{ A_t, C_{t-1}^c \right\} \right] \leq \mathbb{E}\left[ \sum_{t=K+1}^{T} \mathbf{1}\left\{ I_t = 1, C_{t-1}^c \right\} \exp(N_{t-1,1} \cdot \mathsf{kl}(\hat{\mu}_{t-1,1}, \mu_1 - \varepsilon_2)) \right]$$

$$\leq \mathbb{E}\left[ \sum_{t=K+1}^{T} \mathbf{1}\left\{ I_t = 1, \hat{\mu}_{t-1,1} \leq \mu_1 - \varepsilon_2 \right\} \exp(N_{t-1,1} \cdot \mathsf{kl}(\hat{\mu}_{t-1,1}, \mu_1 - \varepsilon_2)) \right]$$

By filtering the time steps when $I_t = 1$, the above can be upper bounded by an expectation over the outcomes in arm 1:

$$\sum_{k=1}^{\infty} \mathbb{E}\left[ \mathbf{1}\left\{ \hat{\mu}_{(k),1} \leq \mu_1 - \varepsilon_2 \right\} \exp(k \cdot \mathsf{kl}(\hat{\mu}_{(k),1}, \mu_1 - \varepsilon_2)) \right]$$

Intuitively, this is well-controlled, as by Chernoff bound (Lemma 25), the probability that $\mathbf{1}\left\{ \hat{\mu}_{(k),1} \leq \mu_1 - \varepsilon_2 \right\}$ is nonzero is exponentially small in $k$; therefore, the expectation of $\mathbf{1}\left\{ \hat{\mu}_{(k),1} \leq \mu_1 - \varepsilon_2 \right\} \exp(k \cdot \mathsf{kl}(\hat{\mu}_{(k),1}, \mu_1 - \varepsilon_2))$ can be controlled. After a careful calculation that utilizes a double-integral argument (that significantly simplifies similar arguments in [11, 24]), we can show that it is at most

$$2H \sum_{k=1}^{\lfloor H \rfloor} \frac{1}{k} + \frac{1}{\mathsf{kl}(\mu_1 - \varepsilon_2, \mu_1)}$$

Summing this over all $k$, we can upper bound $F3$ by

$$F3 \leq O\left( H \ln\left( H \vee e^2 \right) + \frac{1}{\mathsf{kl}(\mu_1 - \varepsilon_2, \mu_1)} \right). \tag{19}$$

A slight generalization of the above argument yields the following useful lemma which further focuses on bounding the expected number of time steps when the number of pulls of arm 1 is in interval $(m, n]$; we defer its proof to Section D.3.5:

**Lemma 13.** *Recall the notations* $A_t = \{I_t = a\}$, $C_{t-1} = \{\hat{\mu}_{t-1,\max} \geq \mu_1 - \varepsilon_2\}$. *Define event* $S_t = \{N_{t,1} > m\}$ *and* $T_t = \{N_{t,1} \leq n\}$ *where* $m \leq n$ *and* $m, n \in \mathbb{N} \cup \{\infty\}$. *Then we have the following inequality:*

$$\mathbb{E}\left[ \sum_{t=K+1}^{T} \mathbf{1}\left\{ A_t, C_{t-1}^c, S_{t-1}, T_{t-1} \right\} \right] \leq \sum_{k=m+1}^{n} \left( \frac{2H}{k} + 1 \right) \exp(-k\mathsf{kl}(\mu_1 - \varepsilon_2, \mu_1))$$

Naively, the bound of $F3$ given by Eq. (19), when combined with previous bounds on $F1$, $F2$, suffice to bound $\mathbb{E}[N_{T,a}]$ by

$$\frac{\ln(T\mathsf{kl}(\mu_a + c\Delta_a, \mu_1 - c\Delta_a) \vee e^2)}{\mathsf{kl}(\mu_a + c\Delta_a, \mu_1 - c\Delta_a)} + O\left( \left( \frac{\dot{\mu}_1 + \Delta_a}{c^2\Delta_a^2} \right) \ln\left( \frac{\dot{\mu}_1 + \Delta_a}{c^2\Delta_a^2} \vee e^2 \right) \right)$$

which establishes KL-MS's asymptotic optimality in the Bernoulli setting and a $O(\sqrt{\dot{\mu}_1 KT \ln T} + K \ln T)$ regret bound. To show a refined $O(\sqrt{\dot{\mu}_1 KT \ln K} + K \ln T)$ regret bound, we prove another bound of $F3$:

$$F3 \leq O\left( H \ln\left( \frac{T}{H} \vee e^2 \right) + \frac{1}{\mathsf{kl}(\mu_1 - \varepsilon_2, \mu_1)} \right). \tag{20}$$

This bound is sometimes stronger than bound (19), since its logarithmic factor depends on $\frac{T}{H}$, which can be substantially smaller than $H$. This alternative bound is crucial to achieve to achieve the $\sqrt{\ln K}$ minimax ratio; see Appendix A and the proof of Theorem 3 therein for details.

To this end, we decompose $F3$ according to whether the number of times arm 1 get pulled exceeds threshold $H$:

$$F3 \leq \mathbb{E}\left[ \sum_{t=K+1}^{T} \mathbf{1}\left\{ A_t, C_{t-1}^c \right\} \right]$$

$$= \underbrace{\mathbb{E}\left[ \sum_{t=K+1}^{T} \mathbf{1}\left\{ A_t, C_{t-1}^c, E_{t-1} \right\} \right]}_{=:F3_1} + \underbrace{\mathbb{E}\left[ \sum_{t=K+1}^{T} \mathbf{1}\left\{ A_t, C_{t-1}^c, E_{t-1}^c \right\} \right]}_{=:F3_2}, \tag{21}$$

where $E_t := \{N_{t,1} \leq H\}$.

Intuitively, $F3_2$ is small as when number of time steps arm 1 is pulled is large, $\hat{\mu}_{t-1,1} \leq \mu_1 - \varepsilon_2$ is unlikely to happen. Indeed, using Lemma 13 with $m = \lfloor H \rfloor, n = \infty$, we immediately have $F3_2 \leq O(\frac{1}{\mathsf{kl}(\mu_1 - \varepsilon_2, \mu_1)})$.

It remains to bound $F3_1$. These terms are concerned with the time steps when arm 1 is pulled at most $\lfloor H \rfloor$ times. Inspired by [33, 23], we introduce an event $\mathcal{E} := \left\{ \forall k \in \left[1, \lfloor H \rfloor\right], \hat{\mu}_{(k),1} \in L_{k,1} \right\}$ (see the definition of $L_{k,1}$ in Eq. (24)) and use it to induce a split:

$$F3_1 \leq \mathbb{E}\left[ \sum_{t=K+1}^{T} \mathbf{1}\left\{ A_t, C_{t-1}^c, E_{t-1}, \mathcal{E} \right\} \right] + \mathbb{E}\left[ \sum_{t=K+1}^{T} \mathbf{1}\left\{ \mathcal{E}^c \right\} \right]$$

$$\leq \mathbb{E}\left[\sum_{t=K+1}^{T} \mathbf{1}\left\{A_t, C_{t-1}^c, E_{t-1}, \hat{\mu}_{t-1,1} \geq \mu_1 - \alpha_{N_{t-1,1}}\right\}\right] + \mathbb{E}\left[\sum_{t=K+1}^{T} \mathbf{1}\left\{\mathcal{E}^c\right\}\right]$$

A probability transferring argument on the first term shows that it is bounded by $O\left(H \ln\left(\frac{T}{H} \vee e^2\right)\right)$; the second term is at most $T\mathbb{P}(\mathcal{E}^c)$, which in turn is at most $H$ using a peeling device and maximal Chernoff inequality (Lemma 24). Combining these two, we prove Eq. (20), which concludes the proof of Lemma 12.

### D.3.2 Proof of Lemma 12

**Additional notations.** In the proof of Lemma 12, we will use the following notations: we denote ramdom variable $X_k := \mu_1 - \hat{\mu}_{(k),1}$, and denote its probability density function by $p_{X_k}(x)$. We also define function $f_k(x) := \exp(k \cdot \mathsf{kl}(\mu_1 - x, \mu_1 - \varepsilon_2))$.

*Proof of Lemma 12.* Recall that we introduce $E_t := \{N_{t,1} \leq H\}$; and according to $E_{t-1}$ we obtain the decomposition Eq. (21) above that $F3 \leq F3_1 + F3_2$.

As we will prove in Lemmas 14 and 15, $F3_1$ and $F3_2$ are bounded by $6H \ln\left(\left(\frac{T}{H} \wedge H\right) \vee e^2\right) + \frac{1}{\mathsf{kl}(\mu_1 - \varepsilon_2, \mu_1)}$ and $\frac{3}{\mathsf{kl}(\mu_1 - \varepsilon_2, \mu_1)}$, respectively. The lemma follows from combining these two bounds by algebra. $\square$

### D.3.3 $F3_1$

**Lemma 14.**

$$F3_1 \leq 6H \ln\left(\left(\frac{T}{H} \wedge H\right) \vee e^2\right) + \frac{1}{\mathsf{kl}(\mu_1 - \varepsilon_2, \mu_1)}$$

*Proof.* We consider three cases.

**Case 1: $H < 1$.** In this case, $E_t$ cannot happen for $t \geq K + 1$ since we have pulled each arm once in the first $K$ rounds and $N_{K,a}$ for any arm should be at least 1. Therefore

$$F3_1 = 0 \leq 6H \ln\left(\left(\frac{T}{H} \wedge H\right) \vee e^2\right) + \frac{1}{\mathsf{kl}(\mu_1 - \varepsilon_2, \mu_1)}$$

**Case 2: $H > \frac{T}{e}$.** According to Lemma 22, we can upper bound $F3_1$ by

$$F3_1 \leq T < 4H$$

$$\leq 6H \ln\left(\left(\frac{T}{H} \wedge H\right) \vee e^2\right) + \frac{1}{\mathsf{kl}(\mu_1 - \varepsilon_2, \mu_1)}$$

**Case 3: $1 \leq H \leq \frac{T}{e}$.** It suffices to prove the following two inequalities:

$$F3_1 \leq 6H \ln\left(\frac{T}{H} \vee e^2\right) + \frac{1}{\mathsf{kl}(\mu_1 - \varepsilon_2, \mu_1)} \tag{22}$$

$$F3_1 \leq 6H \ln(H \vee e^2) + \frac{1}{\mathsf{kl}(\mu_1 - \varepsilon_2, \mu_1)} \tag{23}$$

**Case 3 – Proof of Eq. (22).** To show Eq. (22), we first set up some useful notations. Recall from Section 2 that we denote $\tau_1(s) = \min\{t \geq 1 : N_{t,1} = s\}$ and $\hat{\mu}_{(s),1} := \hat{\mu}_{\tau_1(s),1}$. For $s \in \mathbb{N}$, we first define interval $L_{s,1}$ as:

$$L_{s,1} := \left\{\mu \in [0,1] : \mathsf{kl}(\mu, \mu_1) \leq \frac{2\ln(T/s)}{s} \text{ or } \mu \geq \mu_1\right\}. \tag{24}$$

For notational convenience, we also define $\alpha_s = \mu_1 - \inf L_{s,1}$ and therefore $L_{s,1} = [\mu_1 - \alpha_s, 1]$.

Define $\mathcal{E}$ as $\left\{ \forall k \in \left[1, \lfloor H \rfloor\right], \hat{\mu}_{(k),1} \in L_{k,1} \right\}$. We denote event $\mathcal{E}_k := \left\{ \hat{\mu}_{(k),1} \in L_{k,1} \right\}$; in this notation, $\mathcal{E} = \bigcap_{k=1}^{\lfloor H \rfloor} \mathcal{E}_k$, that is, $\mathcal{E}$ happens iff all $\mathcal{E}_k$ holds simultaneously for all $k$ less or equal to $H$. Note that Lemma 24 implies that $\mathbb{P}(\mathcal{E}^c) \leq \frac{2H}{T}$.

Therefore,

$$
F3_1 \leq \mathbb{E}\left[ \sum_{t=K+1}^{T} \mathbf{1}\left\{ A_t, C_{t-1}^c, E_{t-1}, \mathcal{E} \right\} \right] + \mathbb{E}\left[ \sum_{t=K+1}^{T} \mathbf{1}\left\{ \mathcal{E}^c \right\} \right]
$$

$$
\leq \mathbb{E}\left[ \sum_{t=K+1}^{T} \mathbf{1}\left\{ A_t, C_{t-1}^c, E_{t-1}, \mathcal{E}_{N_{t-1,1}} \right\} \right] + T\mathbb{P}\left( \mathcal{E}^c \right)
$$

$$
\leq \mathbb{E}\left[ \sum_{t=K+1}^{T} \mathbf{1}\left\{ A_t, C_{t-1}^c, E_{t-1}, \mathcal{E}_{N_{t-1,1}} \right\} \right] + 2H, \tag{25}
$$

where in the second inequality, we use the observation that if $\mathcal{E}$ happens and $N_{t-1,1} \leq H$, $\mathcal{E}_{N_{t-1,1}}$ also happens; in the third inequality, we recall that $\mathbb{P}(\mathcal{E}^c) \leq \frac{2H}{T}$.

We continue upper bounding Eq. (25). For the first term in Eq. (25), we use a "probability transfer" argument (Lemma 23) to bound the probability of pulling the suboptimal arm by the probability of pulling optimal times an inflation term.

$$
\mathbb{E}\left[ \sum_{t=K+1}^{T} \mathbf{1}\left\{ A_t, C_{t-1}^c, E_{t-1}, \mathcal{E}_{N_{t-1,1}} \right\} \right] \tag{26}
$$

$$
= \mathbb{E}\left[ \sum_{t=K+1}^{T} \mathbf{1}\left\{ C_{t-1}^c, \mathcal{E}_{N_{t-1,1}}, E_{t-1} \right\} \cdot \mathbb{E}\left[ \mathbf{1}\left\{ A_t \right\} \mid \mathcal{H}_{t-1} \right] \right] \qquad \text{(Law of total expectation)}
$$

$$
\leq \mathbb{E}\left[ \sum_{t=K+1}^{T} \mathbf{1}\left\{ C_{t-1}^c, \mathcal{E}_{N_{t-1,1}}, E_{t-1} \right\} \cdot \exp(N_{t-1,1} \cdot \mathsf{kl}(\hat{\mu}_{t-1,1}, \hat{\mu}_{t-1,\max})) \mathbb{E}\left[ \mathbf{1}\left\{ I_t = 1 \right\} \mid \mathcal{H}_{t-1} \right] \right]
$$

$$
\text{(By Lemma 23)}
$$

$$
= \mathbb{E}\left[ \sum_{t=K+1}^{T} \mathbf{1}\left\{ I_t = 1, C_{t-1}^c, \mathcal{E}_{N_{t-1,1}}, E_{t-1} \right\} \cdot \exp(N_{t-1,1} \cdot \mathsf{kl}(\hat{\mu}_{t-1,1}, \hat{\mu}_{t-1,\max})) \right]
$$

$$
\text{(Law of total expectation)}
$$

Then we make a series of manipulations to reduce the above to bounding the expectation of some function of the random observations drawn from the optimal arm. First, note that for the summation inside the expectation above, each nonzero term corresponds to a time step $t$ such that $t = \tau_1(k)$ for some unique $k \geq 2$, therefore,

$$
\leq \mathbb{E}\left[ \sum_{k=2}^{\infty} \mathbf{1}\left\{ C_{\tau_1(k)-1}^c, \mathcal{E}_{N_{\tau_1(k)-1,1}}, E_{\tau_1(k)-1} \right\} \cdot \exp(N_{\tau_1(k)-1,1} \cdot \mathsf{kl}(\hat{\mu}_{\tau_1(k)-1,1}, \hat{\mu}_{\tau_1(k)-1,\max})) \right]
$$
$$
\tag{27}
$$

$$
\leq \mathbb{E}\left[ \sum_{k=2}^{\infty} \mathbf{1}\left\{ C_{\tau_1(k)-1}^c, \mathcal{E}_{N_{\tau_1(k)-1,1}}, E_{\tau_1(k)-1} \right\} \cdot \exp(N_{\tau_1(k)-1,1} \cdot \mathsf{kl}(\hat{\mu}_{\tau_1(k)-1,1}, \mu_1 - \varepsilon_2)) \right]
$$

$$
\text{(when the condition } C_{\tau_1(k)-1}^c \text{ holds, } \hat{\mu}_{\tau_1(k)-1,1} \leq \hat{\mu}_{\tau_1(k)-1,\max} < \mu_1 - \varepsilon_2 )
$$

$$\leq \mathbb{E}\left[\sum_{k=2}^{\infty} \mathbf{1}\left\{\mathcal{E}_{N_{\tau_1(k)-1,1}}, E_{\tau_1(k)-1}\right\} \cdot \exp(N_{\tau_1(k)-1,1} \cdot \mathsf{kl}(\hat{\mu}_{\tau_1(k)-1,1}, \mu_1 - \varepsilon_2))\right]$$

$$\text{(Dropping } C^c_{\tau_1(k)-1})$$

$$= \mathbb{E}\left[\sum_{k=2}^{\infty} \mathbf{1}\left\{\mathcal{E}_{k-1}, k-1 \leq H\right\} \exp((k-1) \cdot \mathsf{kl}(\hat{\mu}_{(k-1),1}, \mu_1 - \varepsilon_2))\right]$$

$$(N_{\tau_1(k)-1} = k-1 \text{ and } \hat{\mu}_{\tau_1(k)-1,1} = \hat{\mu}_{\tau(k-1),1})$$

$$= \mathbb{E}\left[\sum_{k=1}^{\infty} \mathbf{1}\left\{\mathcal{E}_k, k \leq H\right\} \exp(k \cdot \mathsf{kl}(\hat{\mu}_{(k),1}, \mu_1 - \varepsilon_2))\right] \qquad \text{(shift index } k \text{ by 1)}$$

$$= \mathbb{E}\left[\sum_{k=1}^{\lfloor H \rfloor} \mathbf{1}\left\{\varepsilon_2 \leq \mu_1 - \hat{\mu}_{(k),1} \leq \alpha_k\right\} \cdot \exp(k \cdot \mathsf{kl}(\hat{\mu}_{(k),1}, \mu_1 - \varepsilon_2))\right] + \sum_{k=\lfloor H \rfloor+1}^{\infty} 0$$

$$\text{(Under the conditions } \mathcal{E}_k, E_{\tau_1(k+1)-1}, \text{ when } k \geq \lfloor H \rfloor + 1, E_{\tau_1(k+1)-1} \text{ is always false)}$$

$$= \sum_{k=1}^{\lfloor H \rfloor} \mathbb{E}\left[\mathbf{1}\left\{\varepsilon_2 \leq X_k \leq \alpha_k\right\} \cdot f_k(X_k)\right] \qquad \text{(Recall } X_k = \mu_1 - \hat{\mu}_{(k),1})$$

$$(28)$$

Here the Eq. (28) is the sum of expectation of the function $f_k(X_k)$ over a bounded range $\{\varepsilon_2 \leq X_k \leq \alpha_k\}$ from $k = 1$ to $\lfloor H \rfloor$. Continuing Eq. (28),

$$\mathbb{E}\left[\sum_{t=K+1}^{T} \mathbf{1}\left\{A_t, C^c_{t-1}, E_{t-1}, \mathcal{E}_{N_{t-1,1}}\right\}\right] \tag{29}$$

$$\leq \sum_{k=1}^{\lfloor H \rfloor} \mathbb{E}[f_k(X_k)\mathbf{1}[\{\varepsilon_2 \leq X_k \leq \alpha_k\}] \tag{30}$$

$$= \sum_{k=1}^{\lfloor H \rfloor} \int_{\varepsilon_2}^{\alpha_k} f_k(x) p_{X_k}(x)\,\mathrm{d}x \qquad (p_{X_k}(\cdot) \text{ is the p.d.f. of } X_k)$$

$$= \sum_{k=1}^{\lfloor H \rfloor} \int_{\varepsilon_2}^{\alpha_k} \left(f_k(\varepsilon_2) + \int_{\varepsilon_2}^{x} f'_k(y)\,\mathrm{d}y\right) p_{X_k}(x)\,\mathrm{d}x \qquad (f_k(x) = f_k(\varepsilon_2) + \int_{\varepsilon_2}^{x} f'_k(y)\,\mathrm{d}y)$$

$$= \underbrace{\sum_{k=1}^{\lfloor H \rfloor} \int_{\varepsilon_2}^{\alpha_k} \int_{\varepsilon_2}^{x} f'_k(y) p_{X_k}(x)\,\mathrm{d}y\,\mathrm{d}x}_{A} + \underbrace{\sum_{k=1}^{\lfloor H \rfloor} \int_{\varepsilon_2}^{\alpha_k} p_{X_k}(x)\,\mathrm{d}x}_{B} \tag{31}$$

We denote the first term in Eq. (31) as $A$ and the second one as $B$. Next we are going to handle $A$ and $B$ separately. Starting from the easier one,

$$B = \sum_{k=1}^{\lfloor H \rfloor} \int_{\varepsilon_2}^{\alpha_k} p_{X_k}(x)\,\mathrm{d}x \tag{32}$$

$$\leq \sum_{k=1}^{\lfloor H \rfloor} \mathbb{P}(X_k \geq \varepsilon_2) \tag{33}$$

$$= \sum_{k=1}^{\lfloor H \rfloor} \mathbb{P}(\hat{\mu}_{(k),1} \leq \mu_1 - \varepsilon_2) \tag{34}$$

$$\leq \sum_{k=1}^{\lfloor H \rfloor} \exp(-k \cdot \mathsf{kl}(\mu_1 - \varepsilon_2, \mu_1)) \qquad \text{(Applying Lemma 25)}$$

$$\leq \sum_{k=1}^{\infty} \exp\left(-k \cdot \mathsf{kl}(\mu_1 - \varepsilon_2, \mu_1)\right) \qquad (35)$$

$$\leq \frac{\exp\left(-\mathsf{kl}(\mu_1 - \varepsilon_2, \mu_1)\right)}{1 - \exp\left(-\mathsf{kl}(\mu_1 - \varepsilon_2, \mu_1)\right)} \qquad \text{(Geometric sum)}$$

$$= \frac{1}{\exp\left(\mathsf{kl}(\mu_1 - \varepsilon_2, \mu_1)\right) - 1} \qquad (36)$$

$$\leq \frac{1}{\mathsf{kl}(\mu_1 - \varepsilon_2, \mu_1)} \qquad (e^x \geq x + 1 \text{ when } x \geq 0)$$

$$(37)$$

On the other hand,

$$A = \sum_{k=1}^{\lfloor H \rfloor} \int_{\varepsilon_2}^{\alpha_k} \int_{\varepsilon_2}^{x} f_k'(y) p_{X_k}(x) \, \mathrm{d}y \, \mathrm{d}x \qquad (38)$$

$$= \sum_{k=1}^{\lfloor H \rfloor} \int_{\varepsilon_2}^{\alpha_k} \int_{y}^{\alpha_k} f_k'(y) p_{X_k}(x) \, \mathrm{d}x \, \mathrm{d}y \qquad \text{(Switching the order of integral)}$$

$$= \sum_{k=1}^{\lfloor H \rfloor} \int_{\varepsilon_2}^{\alpha_k} k \frac{\mathrm{d}\mathsf{kl}(\mu_1 - y, \mu_1 - \varepsilon_2)}{\mathrm{d}y} f_k(y) \mathbb{P}(y \leq X_k \leq \alpha_k) \, \mathrm{d}y \qquad \text{(Calculate inner integral)}$$

$$\leq \sum_{k=1}^{\lfloor H \rfloor} \int_{\varepsilon_2}^{\alpha_k} k \frac{\mathrm{d}\mathsf{kl}(\mu_1 - y, \mu_1 - \varepsilon_2)}{\mathrm{d}y} f_k(y) \exp\left(-k \cdot \mathsf{kl}(\mu_1 - y, \mu_1)\right) \, \mathrm{d}y \qquad \text{(Apply Lemma 25)}$$

$$\leq \sum_{k=1}^{\lfloor H \rfloor} \int_{\varepsilon_2}^{\alpha_k} k \frac{\mathrm{d}\mathsf{kl}(\mu_1 - y, \mu_1 - \varepsilon_2)}{\mathrm{d}y} \, \mathrm{d}y \quad (f_k(y) \exp\left(-k \cdot \mathsf{kl}(\mu_1 - y, \mu_1)\right) \leq 1 \text{ when } y \in [\varepsilon_2, \alpha_k])$$

$$= \sum_{k=1}^{\lfloor H \rfloor} k\mathsf{kl}(\mu_1 - \alpha_k, \mu_1 - \varepsilon_2) \qquad \text{(Fundamental Theorem of Calculus)}$$

$$\leq \sum_{k=1}^{\lfloor H \rfloor} 2 \ln \frac{T}{k} \qquad \text{(Recall definition of } \alpha_k)$$

$$\leq 2\lfloor H \rfloor \ln T - 2 \int_{1}^{\lfloor H \rfloor} \ln k \, \mathrm{d}k \qquad \text{(Integral inequality Lemma 20)}$$

$$= 2\lfloor H \rfloor \ln T - 2(k \ln k - k)\big|_{1}^{\lfloor H \rfloor} \qquad \text{(the anti-derivative of } \ln x \text{ is } x \ln x - x)$$

$$= 2\lfloor H \rfloor \ln T - 2\lfloor H \rfloor \ln\left(\lfloor H \rfloor\right) + 2\lfloor H \rfloor \qquad (39)$$

$$= 2\lfloor H \rfloor \ln\left(\frac{T}{\lfloor H \rfloor}\right) + 2\lfloor H \rfloor \qquad (40)$$

$$\leq 2H \ln\left(\frac{T}{H} \vee e^2\right) + 2H \qquad (x \ln \frac{T}{x} \text{ is monotonically increasing when } x \in (0, \frac{T}{e}))$$

$$(41)$$

The fist inequality is due to the Lemma 25. In the second inequality, we use the fact that when $y \in [\varepsilon_2, \alpha_k]$, $f_k(y) \exp\left(-k \cdot \mathsf{kl}(\mu_1 - y, \mu_1)\right) \leq 1$. This is because

$$f_k(y) \exp\left(-k \cdot \mathsf{kl}(\mu_1 - y, \mu_1)\right) = \exp(k \cdot (\mathsf{kl}(\mu_1 - y, \mu_1 - \varepsilon_2) - \mathsf{kl}(\mu_1 - y, \mu_1))) \leq 1$$

In the third one we use the definition of $\alpha_k$ to bound $\mathsf{kl}(\mu_a - \alpha_k, \mu_1 - \varepsilon_2)$ by $\ln\left(\frac{T}{k}\right)$. In the fourth inequality, we apply integral inequality Lemma 20 by letting $f(x) := \ln(x)$, $a = 2$ and $b = \lfloor H \rfloor$. For the last inequality, we use the fact that $x \mapsto x \ln\left(\frac{T}{x}\right)$ is monotonically increasing when $x \in (0, \frac{T}{e})$.

We conclude that $F3_1$ is bounded by

$$F3_1 \leq A + B + 2H \tag{42}$$

$$\leq 2H \ln\left(\frac{T}{H} \vee e^2\right) + 2H + \frac{1}{\mathsf{kl}(\mu_1 - \varepsilon_2, \mu_1)} + 2H \tag{43}$$

$$\leq 6H \ln\left(\frac{T}{H} \vee e^2\right) + \frac{1}{\mathsf{kl}(\mu_1 - \varepsilon_2, \mu_1)} \tag{44}$$

**Case 3 – Proof of Eq.** (23). Applying Lemma 13 by letting $m = 0$ and $n = \lfloor H \rfloor$, we have that

$$F3_1 = \mathbb{E}\left[\sum_{t=K+1}^{T} \mathbf{1}\left\{A_t, C_{t-1}^c, E_{t-1}\right\}\right] \tag{45}$$

$$\leq \sum_{k=1}^{\lfloor H \rfloor} \frac{2\exp(-k\mathsf{kl}(\mu_1 - \varepsilon_2, \mu_1))}{k(\mu_1 - \varepsilon_2)(1 - \mu_1 + \varepsilon_2)h^2(\mu_1, \varepsilon_2)} + \sum_{k=1}^{\lfloor H \rfloor} \exp(-k\mathsf{kl}(\mu_1 - \varepsilon_2, \mu_1)) \tag{46}$$

$$\leq 2H \sum_{k=1}^{\lfloor H \rfloor} \frac{1}{k} + \frac{1}{\mathsf{kl}(\mu_1 - \varepsilon_2, \mu_1)} \tag{47}$$

$$\leq 6H \ln(H \vee e^2) + \frac{1}{\mathsf{kl}(\mu_1 - \varepsilon_2, \mu_1)} \tag{48}$$

where in the second inequality, we use that $\exp(-k\mathsf{kl}(\mu_1 - \varepsilon_2, \mu_1)) \leq 1$ and the definition of $H$, as well as the fact that $\sum_{k=1}^{\lfloor H \rfloor} \exp(-kt) \leq \sum_{k=1}^{\infty} \exp(-kt) = \frac{e^{-t}}{1-e^{-t}} \leq \frac{1}{t}$; in the third inequality, we use the algebraic fact that for $t > 0$, $\sum_{k=1}^{\lfloor H \rfloor} \frac{1}{k} \leq (1 + \ln(\lfloor H \rfloor)) \leq 2(\ln(\lfloor H \rfloor) \vee 1) \leq 2\ln(H \vee e^2)$.

Therefore, when $H \in (1, \frac{T}{e})$, $F3_1$ can be bounded using Eq. (22) and Eq. (23) simultaneously, concluding the proof in Case 3.

In summary, in all three cases, $F3_1$ is upper bounded by $6H \ln\left(\left(\frac{T}{H} \wedge H\right) \vee e^2\right) + \frac{1}{\mathsf{kl}(\mu_1 - \varepsilon_2, \mu_1)}$; this concludes the proof. $\qquad \square$

### D.3.4 $F3_2$

As mentioned in the proof roadmap, intuitively, $F3_2$ is small, since when number of times arm 1 is pulled is large, $\hat{\mu}_{t-1,1} \leq \mu_1 - \varepsilon_2$ is unlikely to happen. Here, we control $F3_2$ using Lemma 13.

**Claim 15.**

$$F3_2 \leq \frac{3}{\mathsf{kl}(\mu_1 - \varepsilon_2, \mu_1)}$$

*Proof.* $F3_2$ is the case where the number of arm pulling of optimal arm 1 is lower bounded by $H$.

$$F3_2 = \mathbb{E}\left[\sum_{t=K+1}^{T} \mathbf{1}\left\{A_t, C_{t-1}^c, E_{t-1}^c\right\}\right] \tag{49}$$

$$\leq \sum_{k=\lfloor H \rfloor+1}^{\infty} \frac{2\exp(-k\mathsf{kl}(\mu_1 - \varepsilon_2, \mu_1))}{k(\mu_1 - \varepsilon_2)(1 - \mu_1 + \varepsilon_2)h^2(\mu_1, \varepsilon_2)} + \frac{1}{\mathsf{kl}(\mu_1 - \varepsilon_2, \mu_1)} \qquad \text{(Lemma 13)}$$

$$\leq \sum_{k=\lfloor H \rfloor+1}^{\infty} \frac{2\exp(-k\mathsf{kl}(\mu_1 - \varepsilon_2, \mu_1))}{H(\mu_1 - \varepsilon_2)(1 - \mu_1 + \varepsilon_2)h^2(\mu_1, \varepsilon_2)} + \frac{1}{\mathsf{kl}(\mu_1 - \varepsilon_2, \mu_1)} \qquad (\lfloor H \rfloor + 1 \geq H)$$

$$\leq \sum_{k=\lfloor H \rfloor+1}^{\infty} 2\exp(-k\mathsf{kl}(\mu_1 - \varepsilon_2, \mu_1)) + \frac{1}{\mathsf{kl}(\mu_1 - \varepsilon_2, \mu_1)} \qquad \text{(By the definition of } H)$$

$$\leq \frac{2\exp(-(\lfloor H \rfloor+1)\mathsf{kl}(\mu_1 - \varepsilon_2, \mu_1))}{1 - \exp(-\mathsf{kl}(\mu_1 - \varepsilon_2, \mu_1))} + \frac{1}{\mathsf{kl}(\mu_1 - \varepsilon_2, \mu_1)} \qquad \text{(Geometric sum)}$$

$$\leq \frac{2}{1 - \exp(-\mathsf{kl}(\mu_1 - \varepsilon_2, \mu_1))} + \frac{1}{\mathsf{kl}(\mu_1 - \varepsilon_2, \mu_1)} \qquad (\exp(-x) \leq 1 \text{ when } x \leq 0)$$

$$\leq \frac{3}{\mathsf{kl}(\mu_1 - \varepsilon_2, \mu_1)} \qquad (50)$$

The first inequality is true because Lemma 13 by letting $m = \lfloor H \rfloor$ and $n = \infty$, as well as the fact that for $t > 0$, $\sum_{k=\lfloor H \rfloor+1}^{\infty} \exp(-kt) \leq \sum_{k=1}^{\infty} \exp(-kt) = \frac{e^{-t}}{1-e^{-t}} \leq \frac{1}{t}$. $\qquad\square$

### D.3.5 Proof of Lemma 13

*Proof of Lemma 13.* For any fixed $k$, recall that we denoted $f_k(x) = \exp(k \cdot \mathsf{kl}(\mu_1 - x, \mu_1 - \varepsilon_2))$, $X_k = \mu_1 - \hat{\mu}_{\tau_1(k),1}$ and the pdf of $X_k$ as $p_{X_k}(x)$.

$$\mathbb{E}\left[\sum_{t=u+1}^{T} \mathbf{1}\left\{A_t, C_{t-1}^c, S_{t-1}, T_{t-1}\right\}\right] \qquad (51)$$

$$= \sum_{t=u+1}^{T} \mathbb{E}\left[\mathbf{1}\left\{C_{t-1}^c, S_{t-1}, T_{t-1}\right\} \mathbb{E}\left[A_t \mid \mathcal{H}_{t-1}\right]\right] \qquad \text{(Law of total expectation)}$$

$$\leq \sum_{t=u+1}^{T} \mathbb{E}\left[\mathbf{1}\left\{C_{t-1}^c, S_{t-1}, T_{t-1}\right\} \cdot \exp(N_{t-1,1} \cdot \mathsf{kl}(\hat{\mu}_{t-1,1}, \hat{\mu}_{t-1,\max}))\mathbb{E}\left[I_t = 1 \mid \mathcal{H}_{t-1}\right]\right]$$

$$\text{(Lemma 23)}$$

$$\leq \sum_{t=u+1}^{T} \mathbb{E}\left[\mathbf{1}\left\{C_{t-1}^c, S_{t-1}, T_{t-1}\right\} \cdot \exp(N_{t-1,1} \cdot \mathsf{kl}(\hat{\mu}_{t-1,1}, \mu_1 - \varepsilon_2))\mathbb{E}\left[I_t = 1 \mid \mathcal{H}_{t-1}\right]\right]$$

$$\text{(when } C_{t-1}^c \text{ happens, } \mathsf{kl}(\hat{\mu}_{t-1,1}, \hat{\mu}_{t-1,\max}) \leq \mathsf{kl}(\hat{\mu}_{t-1,1}, \mu_1 - \varepsilon_2))$$

$$= \sum_{t=u+1}^{T} \mathbb{E}\left[\mathbf{1}\left\{I_t = 1, C_{t-1}^c, S_{t-1}, T_{t-1}\right\} \cdot \exp(N_{t-1,1} \cdot \mathsf{kl}(\hat{\mu}_{t-1,1}, \mu_1 - \varepsilon_2))\right]$$

$$\text{(Law of total expectation)}$$

$$\leq \mathbb{E}\left[\sum_{k=2}^{\infty} \mathbf{1}\left\{C_{\tau_1(k)-1}^c, k - 1 \in (m, n]\right\} \cdot \exp(N_{\tau_1(k)-1,1} \cdot \mathsf{kl}(\hat{\mu}_{(k-1),1}, \mu_1 - \varepsilon_2))\right]$$

(for any $t$ such that $\mathbf{1}\{I_t = 1\}$ is nonzero, $t = \tau_1(k)$ for some unique $k$; $N_{\tau(k)-1,1} = k - 1$, and $\hat{\mu}_{\tau(k)-1,1} = \hat{\mu}_{(k-1),1}$)

$$= \mathbb{E}\left[\sum_{k=2}^{\infty} \mathbf{1}\left\{C_{\tau_1(k)-1}^c, k \in (m+1, n+1]\right\} \cdot \exp((k-1) \cdot \mathsf{kl}(\hat{\mu}_{(k-1),1}, \mu_1 - \varepsilon_2))\right] \qquad \text{(algebra)}$$

$$\leq \mathbb{E}\left[\sum_{k=m+2}^{n+1} \mathbf{1}\left\{\mu_1 \geq \mu_1 - \hat{\mu}_{(k-1),1} > \varepsilon_2\right\} \cdot \exp((k-1) \cdot \mathsf{kl}(\hat{\mu}_{(k-1),1}, \mu_1 - \varepsilon_2))\right] \qquad (52)$$

$$= \mathbb{E}\left[\sum_{k=m+1}^{n} \mathbf{1}\left\{\mu_1 \geq \mu_1 - \hat{\mu}_{(k),1} > \varepsilon_2\right\} \cdot f_k(\mu_1 - \hat{\mu}_{(k),1})\right], \qquad \text{(shift } k \text{ by 1)}$$

$$(53)$$

here, for the second to last inequality, we use the fact that when $S_{\tau_1(k)-1}$ happens, $k - 1 > m$, and when $T_{\tau_1(k)-1}$ happens, $k - 1 \leq n$. In the last inequality, we use the fact that when $C_{\tau_1(k)-1}^c$ happens,

$\hat{\mu}_{\tau_1(k)-1,\max} < \mu_1 - \varepsilon_2$. Combining this with the fact that $\hat{\mu}_{(k-1),1} = \hat{\mu}_{\tau_1(k)-1,1} \le \hat{\mu}_{\tau_1(k)-1,\max}$, we have $\mu_1 - \hat{\mu}_{(k-1),1} > \varepsilon_2$.

Hence Eq. (53) becomes

$$
\begin{aligned}
(53) &= \mathbb{E}\left[\sum_{k=m+1}^{n} \mathbf{1}\left\{\mu_1 \ge X_k > \varepsilon_2\right\} \cdot f_k(X_k)\right] \\
&= \sum_{k=m+1}^{n} \int_{\varepsilon_2}^{\mu_1} f_k(x) p_{X_k}(x)\, dx \\
&= \sum_{k=m+1}^{n} \int_{\varepsilon_2}^{\mu_1} p_{X_k}(x)\left(f_k(\varepsilon_2) + \sum_{k=m+1}^{n} \int_{\varepsilon_2}^{x} f_k'(y)\, dy\right) dx \\
&\qquad\qquad\qquad\qquad\qquad\qquad (f_k(x) = f_k(\varepsilon_2) + \int_{\varepsilon_2}^{x} f_k'(y)\, dy)) \\
&= \sum_{k=m+1}^{n} \int_{\varepsilon_2}^{\mu_1} p_{X_k}(x) f_k(\varepsilon_2)\, dx + \sum_{k=m+1}^{n} \int_{\varepsilon_2}^{\mu_1}\int_{\varepsilon_2}^{x} p_{X_k}(x) f_k'(y)\, dy\, dx \\
&= \underbrace{\sum_{k=m+1}^{n} \int_{\varepsilon_2}^{\mu_1} p_{X_k}(x) f_k(\varepsilon_2)\, dx}_{A} + \underbrace{\sum_{k=m+1}^{n} \int_{\varepsilon_2}^{\mu_1}\int_{y}^{\mu_1} p_{X_k}(x) f_k'(y)\, dx\, dy}_{B} \\
&\qquad\qquad\qquad\qquad\qquad\qquad\qquad\qquad\qquad\qquad \text{(Exchange the order of integral)}
\end{aligned}
$$

**For $A$:**

$$
A = \sum_{k=m+1}^{n} \int_{\varepsilon_2}^{\mu_1} p_{X_k}(x) f_k(\varepsilon_2)\, dx \tag{54}
$$

$$
\le \sum_{k=m+1}^{n} \int_{\varepsilon_2}^{\mu_1} p_{X_k}(x) f_k(\varepsilon_2)\, dx \tag{55}
$$

$$
\le \sum_{k=m+1}^{n} \exp\left(-k \cdot \mathsf{kl}\left(\mu_1 - \varepsilon_2, \mu_1\right)\right) f_k(\varepsilon_2) \qquad \text{(By Lemma 25)} \tag{56}
$$

$$
= \sum_{k=m+1}^{n} \exp\left(-k \cdot \mathsf{kl}\left(\mu_1 - \varepsilon_2, \mu_1\right)\right) \tag{56}
$$

where the last equality is because $f_k(\varepsilon_2) = 1$. **For $B$:**

$$
B \tag{57}
$$

$$
= \sum_{k=m+1}^{n} \int_{\varepsilon_2}^{\mu_1}\int_{\varepsilon_2}^{x} f_k'(y) p_{X_k}(x)\, dy\, dx \tag{58}
$$

$$
= \sum_{k=m+1}^{n} \int_{\varepsilon_2}^{\mu_1}\int_{y}^{\mu_1} f_k'(y) p_{X_k}(x)\, dx\, dy \qquad \text{(Switching the order of integral)}
$$

$$
= \sum_{k=m+1}^{n} \int_{\varepsilon_2}^{\mu_1} k\frac{\mathsf{dkl}(\mu_1 - y, \mu_1 - \varepsilon_2)}{dy} f_k(y)\mathbb{P}(y \le x \le \mu_1)\, dy \qquad \text{(Calculate inner integral)}
$$

$$
= \sum_{k=m+1}^{n} \int_{\varepsilon_2}^{\mu_1} f_k(y) \cdot k\frac{\mathsf{dkl}(\mu_1 - y, \mu_1 - \varepsilon_2)}{dy} \cdot \exp(-k \cdot \mathsf{kl}(\mu_1 - y, \mu_1))\, dy \quad \text{(Apply Lemma 25)}
$$

$$= \sum_{k=m+1}^{n} \int_{\varepsilon_2}^{\mu_1} \exp\left(k(\mathsf{kl}(\mu_1 - y, \mu_1 - \varepsilon_2) - \mathsf{kl}(\mu_1 - y, \mu_1))\right) \cdot k \frac{\mathrm{dkl}(\mu_1 - y, \mu_1 - \varepsilon_2)}{\mathrm{d}y} \, \mathrm{d}y \quad (59)$$

$$= \sum_{k=m+1}^{n} \int_{\varepsilon_2}^{\mu_1} k \exp\left(-k\mathsf{kl}(\mu_1 - \varepsilon_2, \mu_1)\right) \cdot \quad (60)$$

$$\exp\left(k\,(y - \varepsilon_2)\ln\left(\frac{(1 - \mu_1)(\mu_1 - \varepsilon_2)}{(1 - \mu_1 + \varepsilon_2)\mu_1}\right)\right) \frac{\mathrm{dkl}(\mu_1 - y, \mu_1 - \varepsilon_2)}{\mathrm{d}y} \, \mathrm{d}y$$

(By Lemma 29 with $\phi(x) = x\ln(x) + (1 - x)\ln(1 - x)$, which induces $B_\phi(z, x) = \mathsf{kl}(z, x)$)

$$= \sum_{k=m+1}^{n} \int_{\varepsilon_2}^{\mu_1} k \exp\left(-k\mathsf{kl}(\mu_1 - \varepsilon_2, \mu_1) - k\,(y - \varepsilon_2)\,h(\mu_1, \varepsilon_2)\right) \frac{\mathrm{dkl}(\mu_1 - y, \mu_1 - \varepsilon_2)}{\mathrm{d}y} \, \mathrm{d}y$$

$$\text{(Recall } \ln(\tfrac{(1 - \mu_1 + \varepsilon_2)\mu_1}{(1 - \mu_1)(\mu_1 - \varepsilon_2)}) = h(\mu_1, \varepsilon_2))$$

$$= \sum_{k=m+1}^{n} \exp(-k\mathsf{kl}(\mu_1 - \varepsilon_2, \mu_1)) \cdot \quad (61)$$

$$\left(\underbrace{\int_{\varepsilon_2}^{\mu_1} k \exp\left(-k\,(y - \varepsilon_2)\,h(\mu_1, \varepsilon_2)\right) \frac{\mathrm{dkl}(\mu_1 - y, \mu_1 - \varepsilon_2)}{\mathrm{d}y} \, \mathrm{d}y}_{\mathrm{INT}}\right) \quad (62)$$

Here, in the third to the last equation we have applied Lemma 29 and $\phi(x) = x\ln(x) + (1 - x)\ln(1 - x)$, $B_\phi(z, x)$ becomes $\mathsf{kl}(z, x)$. We set $z := (\mu_1 - y, 1 - \mu_1 + y)$, $x := (\mu_1 - \varepsilon_2, 1 - \mu_1 + \varepsilon_2)$ and $y := (\mu, 1 - \mu_1)$. Under this setting, according to Lemma 29, we have $\mathsf{kl}(\mu_1 - y, \mu_1 - \varepsilon_2) - \mathsf{kl}(\mu_1 - y, \mu_1) = -\mathsf{kl}(\mu_1 - \varepsilon_2, \mu_1) + (y - \varepsilon_2)\ln\left(\frac{(1 - \mu_1)(\mu_1 - \varepsilon_2)}{(1 - \mu_1 + \varepsilon_2)\mu_1}\right)$.

Next, we need to give an upper bound to the integral part INT carefully. By applying the observation below, the integral will become

$$\mathrm{INT} = \int_{\varepsilon_2}^{\mu_1} k \exp\left(-k\,(y - \varepsilon_2)\,h(\mu_1, \varepsilon_2)\right) \frac{\mathrm{dkl}(\mu_1 - y, \mu_1 - \varepsilon_2)}{\mathrm{d}y} \, \mathrm{d}y \quad (63)$$

$$= \int_{\varepsilon_2}^{\mu_1} k \exp\left(-k\,(y - \varepsilon_2)\,h(\mu_1, \varepsilon_2)\right) \frac{\mathrm{dkl}(\mu_1 - y, \mu_1 - \varepsilon_2)}{\mathrm{d}(\mu_1 - y)} \frac{\mathrm{d}(\mu_1 - y)}{\mathrm{d}y} \, \mathrm{d}y \quad (64)$$

$$= -\int_{\varepsilon_2}^{\mu_1} k \exp\left(-k\,(y - \varepsilon_2)\,h(\mu_1, \varepsilon_2)\right) \left(\ln\left(\frac{\mu_1 - y}{1 - \mu_1 + y}\right) - \ln\left(\frac{\mu_1 - \varepsilon_2}{1 - \mu_1 + \varepsilon_2}\right)\right) \mathrm{d}y \quad (65)$$

$$= \int_{\varepsilon_2}^{\mu_1} k \exp\left(-k\,(y - \varepsilon_2)\,h(\mu_1, \varepsilon_2)\right) \int_{\mu_1 - y}^{\mu_1 - \varepsilon_2} \left(\frac{1}{x} + \frac{1}{1 - x}\right) \mathrm{d}x \, \mathrm{d}y \qquad (\ln\tfrac{a}{b} = \int_a^b \tfrac{1}{x}\,\mathrm{d}x)$$

$$= \int_0^{\mu_1 - \varepsilon_2} \int_{\mu_1 - x}^{\mu_1} k \exp\left(-k\,(y - \varepsilon_2)\,h(\mu_1, \varepsilon_2)\right) \left(\frac{1}{x} + \frac{1}{1 - x}\right) \mathrm{d}y \, \mathrm{d}x$$

(Change the order of integral)

$$= \int_0^{\mu_1 - \varepsilon_2} \frac{\exp\left(k\varepsilon_2 h(\mu_1, \varepsilon_2)\right)}{h(\mu_1, \varepsilon_2)} \left(\exp\left(-k\,(\mu_1 - x)\,h(\mu_1, \varepsilon_2)\right) - \exp\left(-k\mu_1 h(\mu_1, \varepsilon_2)\right)\right) \cdot$$

$$\quad (66)$$

$$\left(\frac{1}{x} + \frac{1}{1 - x}\right) \mathrm{d}x \qquad \text{(Calculate inner integral)}$$

$$= \frac{\exp\left(-k\,(\mu_1 - \varepsilon_2)\,h(\mu_1, \varepsilon_2)\right)}{h(\mu_1, \varepsilon_2)} \cdot \quad (67)$$

$$\left(\underbrace{\int_0^{\mu_1-\varepsilon_2} \frac{\exp\left(kxh(\mu_1,\varepsilon_2)\right)-1}{x}\,\mathrm{d}x}_{part\ I} + \underbrace{\int_0^{\mu_1-\varepsilon_2} \frac{\exp\left(kxh(\mu_1,\varepsilon_2)\right)-1}{1-x}\,\mathrm{d}x}_{part\ II}\right) \tag{68}$$

**Part I** For part I, we can bound it by

$$Part\ I = \int_0^{\mu_1-\varepsilon_2} \frac{\exp\left(kxh(\mu_1,\varepsilon_2)\right)-1}{x}\,\mathrm{d}x \tag{69}$$

$$= \int_0^{(\mu_1-\varepsilon_2)kh(\mu_1,\varepsilon_2)} kh(\mu_1,\varepsilon_2)\frac{\exp\left(y\right)-1}{y}\frac{1}{kh(\mu_1,\varepsilon_2)}\,\mathrm{d}y$$
$$\text{(change variable } y = kxh(\mu_1,\varepsilon_2))$$

$$= \int_0^{(\mu_1-\varepsilon_2)kh(\mu_1,\varepsilon_2)} \frac{\exp\left(y\right)-1}{y}\,\mathrm{d}y \tag{70}$$

$$\leq 2\frac{\exp\left((\mu_1-\varepsilon_2)kh(\mu_1,\varepsilon_2)\right)}{(\mu_1-\varepsilon_2)kh(\mu_1,\varepsilon_2)} \quad \text{(Using Lemma 19 by letting } t = (\mu_1-\varepsilon_2)kh(\mu_1,\varepsilon_2))$$
$$\tag{71}$$

**Part II** For part II,

$$part\ II = \int_0^{\mu_1-\varepsilon_2} \frac{\exp\left(kxh(\mu_1,\varepsilon_2)\right)-1}{1-x}\,\mathrm{d}x \tag{72}$$

$$\leq \int_0^{\mu_1-\varepsilon_2} \frac{\exp\left(kxh(\mu_1,\varepsilon_2)\right)-1}{1-\mu_1+\varepsilon_2}\,\mathrm{d}x \quad \text{(Bound denominator by } 1-\mu_1+\varepsilon_2)$$

$$= \frac{1}{1-\mu_1+\varepsilon_2}\left(\frac{1}{kh(\mu_1,\varepsilon_2)}\exp\left(kxh(\mu_1,\varepsilon_2)\right)-x\right)\Big|_0^{\mu_1-\varepsilon_2} \quad \text{(calculate integral)}$$

$$\leq \frac{\exp\left(k\left(\mu_1-\varepsilon_2\right)h(\mu_1,\varepsilon_2)\right)}{(1-\mu_1+\varepsilon_2)kh(\mu_1,\varepsilon_2)} \tag{73}$$

Hence from Eq.(71) and Eq.(73), by multiplying the first factor in the Eq. (68), we can bound INT by

$$\text{INT} \leq \frac{\exp\left(-k(\mu_1-\varepsilon_2)h(\mu_1,\varepsilon_2)\right)}{h(\mu_1,\varepsilon_2)}\,(\text{part I + part II}) \tag{74}$$

$$\leq \frac{\exp\left(-k(\mu_1-\varepsilon_2)h(\mu_1,\varepsilon_2)\right)}{h(\mu_1,\varepsilon_2)}\left(2\frac{\exp\left((\mu_1-\varepsilon_2)kh(\mu_1,\varepsilon_2)\right)}{(\mu_1-\varepsilon_2)kh(\mu_1,\varepsilon_2)} + \frac{\exp\left(k\left(\mu_1-\varepsilon_2\right)h(\mu_1,\varepsilon_2)\right)}{(1-\mu_1+\varepsilon_2)kh(\mu_1,\varepsilon_2)}\right) \tag{75}$$

$$\leq \frac{2}{kh^2(\mu_1,\varepsilon_2)}\cdot\left(\frac{1}{\mu_1-\varepsilon_2}+\frac{1}{1-\mu_1+\varepsilon_2}\right) = \frac{2}{kh^2(\mu_1,\varepsilon_2)}\cdot\left(\frac{1}{(\mu_1-\varepsilon_2)(1-\mu_1+\varepsilon_2)}\right) \tag{76}$$

Therefore, we can upper bound $B$ by

$$B \leq \sum_{k=m+1}^{n} \exp(-k\mathsf{kl}(\mu_1-\varepsilon_2,\mu_1))\cdot\text{INT} \tag{77}$$

$$= \sum_{k=m+1}^{n} \frac{2\exp(-k\mathsf{kl}(\mu_1-\varepsilon_2,\mu_1))}{k(\mu_1-\varepsilon_2)(1-\mu_1+\varepsilon_2)h^2(\mu_1,\varepsilon_2)} \tag{78}$$

In a summary, by combining Eq. (56) and Eq. (78), we have

$$\sum_{t=K+1}^{T} \mathbb{P}\left(A_t, B_{t-1}^c, C_{t-1}^c, S_t, T_t\right)$$

$$\le A + B$$

$$\le \sum_{k=m+1}^{n} \left( \frac{2}{k(\mu_1 - \varepsilon_2)(1 - \mu_1 + \varepsilon_2)h^2(\mu_1, \varepsilon_2)} + 1 \right) \exp(-k\mathsf{kl}(\mu_1 - \varepsilon_2, \mu_1)). \qquad \square$$

# E  Auxiliary Lemmas

## E.1  Control Variance over Bounded Distribution

**Lemma 16.** *Let $\nu$ be a distribution supported on $[0, 1]$ with mean $\mu$. Then, the variance of $\nu$ is no larger than $\dot\mu$.*

*Proof.* For a random variable $X \sim \nu$,

$$\begin{aligned}
\mathrm{Var}_{X \sim \nu}[X] &= \mathbb{E}\left[X^2\right] - \left(\mathbb{E}[X]\right)^2 \\
&\le \mathbb{E}\left[X\right] - \left(\mathbb{E}[X]\right)^2 && (X \ge X^2 \text{ when } X \in [0, 1]) \\
&= \mu - \mu^2 \\
&= \dot\mu && (\text{Recall that } \dot\mu = \mu(1 - \mu))
\end{aligned}$$

$$\square$$

## E.2  Controlling the Moment Generating Function

**Lemma 17.** *Let $\nu$ be a distribution with mean $\mu$ and support set $\mathcal{S} = [0, 1]$. Then, moment generating function of $X \sim \nu$ is smaller than $1 - \mu + e^\lambda$. More specifically,*

$$\mathbb{E}_{X \sim \nu}\left[e^{\lambda X}\right] \le \mu e^\lambda + (1 - \mu) \tag{79}$$

*Proof.* Since $e^y$ is a convex function, we apply Jensen's inequality on two point $y = 0$ and $y = \lambda$ with weights $1 - x$ and $x$ respectively.

$$\begin{aligned}
\exp\left((1 - x) \cdot 0 + x \cdot \lambda\right) &\le (1 - x) \cdot e^0 + x \cdot e^\lambda \\
\Rightarrow \mathbb{E}\left[\exp\left(\lambda x\right)\right] &\le (1 - \mu) + \mu \cdot e^\lambda && \square
\end{aligned}$$

## E.3  Upper Bounding the Sum of Probability of Cumulative Arm Pulling

**Lemma 18.** *Let $\{E_t\}_{t=1}^{T}$ be a sequence of events determined at the time step $t$ and $N := B_{t_1 - 1}$. $M$ is an integer such that $1 \le N \le M \le T$. Let $t_1, t_2$ be time indices in $\mathbb{N}$ such that $t_1 < t_2$ and $F_t := \left\{\sum_{i=1}^{t} \mathbf{1}\left\{E_i\right\} < M\right\}$ which is the event of upper bounding cumulative count Then, it holds deterministically that*

$$\sum_{t=t_1}^{t_2} \mathbf{1}\left\{E_t, F_{t-1}\right\} \le M - N \tag{80}$$

## E.4  Useful Integral Bound

**Lemma 19.** *Let $f(t) = \int_0^t \frac{\exp(x) - 1}{x} \, \mathrm{d}x$. We have the inequality $f(t) \le 2 \cdot \frac{\exp(t)}{t}$.*

*Proof.* According to the Taylor expansion of $\exp(x)$ at $x = 0$, we have

$$\frac{\exp(x) - 1}{x} = \frac{\sum_{i=0}^{\infty} \frac{x^i}{i!} - 1}{x} = \sum_{i=0}^{\infty} \frac{x^i}{(i+1)!}$$

Then for $f(t)$,

$$
\begin{aligned}
f(t) &= \int_0^t \sum_{i=0}^{\infty} \frac{x^i}{(i+1)!} \, \mathrm{d}x \\
&= \sum_{i=0}^{\infty} \int_0^t \frac{x^i}{(i+1)!} \, \mathrm{d}x \\
&= \sum_{i=0}^{\infty} \frac{t^{i+1}}{(i+1) \cdot (i+1)!} \\
&\leq 2 \cdot \sum_{i=0}^{\infty} \frac{t^{i+1}}{(i+2)!} \\
&= 2 \cdot \sum_{i=2}^{\infty} \frac{t^{i-1}}{i!} \\
&\leq 2 \cdot \frac{\exp(t)}{t} \qquad \qquad \square
\end{aligned}
$$

**Lemma 20.** *Given an integrable function $f(x)$ which is monotonically increasing in the range $\mathbb{R}^+$. For two integers $1 \leq a < b$, we have the following inequality*

$$
\int_{a-1}^b f(x) \, \mathrm{d}x \leq \sum_{i=a}^b f(i) \leq f(a) + \int_a^b f(x) \, \mathrm{d}x
$$

*Proof.* For the LHS inequality,

$$
\begin{aligned}
\sum_{i=a}^b f(i) &= \sum_{i=a}^b f(i) \cdot (i+1-i) \\
&\geq \sum_{i=a}^b \int_{i-1}^i f(x) \, \mathrm{d}x \\
&= \int_{a-1}^b f(x) \, \mathrm{d}x
\end{aligned}
$$

For the RHS,

$$
\begin{aligned}
\sum_{i=a}^b f(i) &= \sum_{i=a}^b f(i) \cdot (i+1-i) \\
&= f(a) + \sum_{i=a+1}^b f(i) \cdot (i-(i-1)) \\
&\leq f(a) + \sum_{i=a+1}^b \int_{i-1}^i f(x) \, \mathrm{d}x \\
&= f(a) + \int_a^b f(x) \, \mathrm{d}x \qquad \qquad \square
\end{aligned}
$$

### E.5 Bounding $H$

**Lemma 21.** *Given $h(\mu_1, \varepsilon_2) = \ln\left( \frac{(1-\mu_1+\varepsilon_2)\mu_1}{(1-\mu_1)(\mu_1-\varepsilon_2)} \right)$ with $0 < \varepsilon_2 < \mu_1$, there exists an inequality*

$$
h(\mu_1, \varepsilon_2) \geq \frac{\varepsilon_2(1+\varepsilon_2)}{\mu_1(1-\mu_1+\varepsilon_2)}
$$

*Proof.* Using concavity of logarithm function which is for two nonnegative point $x, y$

$$\forall x, y > 0, \ln y \leq \ln x + \frac{y - x}{x}$$

We apply this property to get the lower bound $h(\mu_a, \varepsilon_2)$ by

$$
\begin{aligned}
h(\mu_1, \varepsilon_2) &= \ln\left(\frac{(1 - \mu_1 + \varepsilon_2)\mu_1}{(1 - \mu_1)(\mu_1 - \varepsilon_2)}\right) \\
&= \ln \mu_1 - \ln(\mu_1 - \varepsilon_2) + \ln(1 - \mu_1 + \varepsilon_2) - \ln(1 - \mu_1) \\
&\geq \frac{\varepsilon_2}{\mu_1} + \frac{\varepsilon_2}{1 - \mu_1 + \varepsilon_2} \qquad\qquad \text{(concavity property of logarithm)} \\
&= \frac{\varepsilon_2(1 + \varepsilon_2)}{\mu_1(1 - \mu_1 + \varepsilon_2)} \qquad\qquad\qquad\qquad\qquad\qquad\qquad \square
\end{aligned}
$$

**Lemma 22.** *Given* $H := \frac{1}{(1 - \mu_1 + \varepsilon_2)(\mu_1 - \varepsilon_2)h^2(\mu_1, \varepsilon_2)}$, $h(\mu_1, \varepsilon_2) := \ln\left(\frac{(1 - \mu_1 + \varepsilon_2)\mu_1}{(1 - \mu_1)(\mu_1 - \varepsilon_2)}\right)$, $0 \leq \varepsilon_2 \leq \frac{1}{2}\mu_1$ *and* $0 < \mu_1 \leq 1$. *$H$ is bounded by the following inequality*

$$H \leq \frac{2\dot{\mu}_1}{\varepsilon_2^2} + \frac{2}{\varepsilon_2}$$

*where* $\dot{\mu}_1 := (1 - \mu_1)\mu_1$.

*Proof.* According to Lemma 21, $h(\mu_1, \varepsilon_2)$ is lower bounded by

$$h(\mu_1, \varepsilon_2) \geq \frac{\varepsilon_2(1 + \varepsilon_2)}{\mu_1(1 - \mu_1 + \varepsilon_2)}$$

To upper bound $H$,

$$
\begin{aligned}
H &\leq \frac{1}{(\mu_1 - \varepsilon_2)(1 - \mu_1 + \varepsilon_2)\left(\frac{\varepsilon_2(1 + \varepsilon_2)}{\mu_1(1 - \mu_1 + \varepsilon_2)}\right)^2} \\
&= \frac{\mu_1^2(1 - \mu_1 + \varepsilon_2)}{(\mu_1 - \varepsilon_2)\varepsilon_2^2(1 + \varepsilon_2)^2} \\
&= \frac{\mu_1}{\mu_1 - \varepsilon_2} \cdot \left(\frac{1}{1 + \varepsilon_2}\right)^2 \cdot \frac{(1 - \mu_1 + \varepsilon_2)\mu_1}{\varepsilon_2^2} \\
&\leq 2 \cdot 1 \cdot \frac{(1 - \mu_1 + \varepsilon_2)\mu_1}{\varepsilon_2^2} \qquad\qquad\qquad (0 \leq \varepsilon_2 \leq \frac{\mu_1}{2}) \\
&\leq \frac{2\dot{\mu}}{\varepsilon_2^2} + \frac{2}{\varepsilon_2} \qquad\qquad\qquad\qquad\qquad\qquad\qquad \square
\end{aligned}
$$

### E.6 Probability Transferring Inequality

**Lemma 23.** *Let $\mathcal{H}_{t-1}$ be the $\sigma$-field generated by historical trajectory up to time (and including) $t - 1$, which is defined as $\sigma\left(\{I_i, r_i\}_{i=1}^{t-1}\right)$ ($I_i$ is the arm pulling at the time round $i$ and $r_i$ is its return reward). Given the algorithm 1, the probability of pulling a sub-optimal arm $a$ has the following relationship.*

$$\mathbb{P}(I_t = a | \mathcal{H}_{t-1}) \leq \exp(-N_{t-1,a}\mathsf{kl}(\hat{\mu}_{t-1,a}, \hat{\mu}_{t-1,\max}))$$

*Also,*

$$\mathbb{P}(I_t = a | \mathcal{H}_{t-1}) \leq \exp(N_{t-1,1}\mathsf{kl}(\hat{\mu}_{t-1,1}, \hat{\mu}_{t-1,\max}))\mathbb{P}(I_t = 1 \mid \mathcal{H}_{t-1})$$

*Proof.* For the first item, recall the definition of $p_{t,a} = \exp\left(-N_{t-1,a}\mathsf{kl}(\hat{\mu}_{t-1,a}, \hat{\mu}_{t-1,\max})\right)/M_t$.

$$
\begin{aligned}
\mathbb{P}(I_t = a | \mathcal{H}_{t-1}) &= p_{t,a} \\
&= \frac{\exp(-N_{t-1,a}\mathsf{kl}(\hat{\mu}_{t-1,a}, \hat{\mu}_{t-1,\max}))}{M_t} \\
&\leq \exp(-N_{t-1,a}\mathsf{kl}(\hat{\mu}_{t-1,a}, \hat{\mu}_{t-1,\max}))
\end{aligned}
$$

Since $M_t \geq 1$ from the fact that $KL(\hat{\mu}_{t-1,a}, \hat{\mu}_{t-1,\max}) = 0$ when $a = \arg\max_{i \in [K]} \hat{\mu}_{t-1,i}$, recall the definition of $M_t$, we have $M_t \geq 1$.

For the second item, recall the algorithm setting, there exists the following relationship

$$
\begin{aligned}
&\mathbb{P}(I_t = a | \mathcal{H}_{t-1}) \\
&= \frac{\exp(-N_{t-1,a}\mathsf{kl}(\hat{\mu}_{t-1,a}, \hat{\mu}_{t-1,\max})}{M_t} \\
&= \frac{\exp(-N_{t-1,a}\mathsf{kl}(\hat{\mu}_{t-1,a}, \hat{\mu}_{t-1,\max}))}{\exp(-N_{t-1,1}\mathsf{kl}(\hat{\mu}_{t-1,1}, \hat{\mu}_{t-1,\max}))} \cdot \frac{\exp(-N_{t-1,1}\mathsf{kl}(\hat{\mu}_{t-1,1}, \hat{\mu}_{t-1,\max}))}{M_t} \\
&= \frac{\exp(-N_{t-1,a}\mathsf{kl}(\hat{\mu}_{t-1,a}, \hat{\mu}_{t-1,\max}))}{\exp(-N_{t-1,1}\mathsf{kl}(\hat{\mu}_{t-1,1}, \hat{\mu}_{t-1,\max}))} \cdot \mathbb{P}(I_t = 1 \mid \mathcal{H}_{t-1}) \\
&\leq \frac{1}{\exp(-N_{t-1,1}\mathsf{kl}(\hat{\mu}_{t-1,1}, \hat{\mu}_{t-1,\max}))} \cdot \mathbb{P}(I_t = 1 \mid \mathcal{H}_{t-1}) \\
&= \exp(N_{t-1,1}\mathsf{kl}(\hat{\mu}_{t-1,1}, \hat{\mu}_{t-1,\max}))\mathbb{P}(I_t = 1 \mid \mathcal{H}_{t-1})
\end{aligned}
$$

The first inequality is due to $\mathsf{kl}(\hat{\mu}_{t-1,a}, \hat{\mu}_{t-1,\max}) \geq 0$ and $\exp(-N_{t-1,a}\mathsf{kl}(\hat{\mu}_{t-1,a}, \hat{\mu}_{t-1,\max})) \leq 1$. $\qquad\square$

### E.7 Bounding the Deviation of Running Averages from the Population Mean

**Lemma 24.** *The distribution of random variable $X$ is $\nu_i$ which is a distribution with bounded support $[0, 1]$ and mean $\mu$. Suppose that there is a sequence of sample $\{X_i\}_{i=1}^{k}$ draw i.i.d. from $\nu_i$. Denote $\sum_{i=1}^{s} X_i/s$ as $\hat{\mu}_s$.*

*Let $\epsilon > 0$, assume $T \geq k \geq 1$. Then,*

$$
\mathbb{P}\left(\exists 1 \leq s \leq k : \mathsf{kl}(\hat{\mu}_s, \mu) \geq \frac{2\ln(T/s)}{s}\right) \leq \frac{2k}{T}
$$

*Proof.* We apply the peeling device $\frac{k}{2^{n+1}} < s \leq \frac{k}{2^n}$ to upper bound the upper left term

$$
\mathbb{P}\left(\exists s \leq k : \mathsf{kl}(\hat{\mu}_s, \mu) \geq \frac{2\ln(T/s)}{s}\right) \tag{81}
$$

$$
\leq \sum_{n=0}^{\infty} \mathbb{P}\left(\exists s : s \in [k] \cap (\frac{k}{2^{n+1}}, \frac{k}{2^n}], \mathsf{kl}(\hat{\mu}_s, \mu) \geq \frac{2\ln(T/s)}{s}\right) \tag{82}
$$

$$
\leq \sum_{n=0}^{\infty} \mathbb{P}\left(\exists s : s \in [k] \cap (\frac{k}{2^{n+1}}, \frac{k}{2^n}], \mathsf{kl}(\hat{\mu}_s, \mu) \geq \frac{2^{n+1}\ln(2^n T/k)}{k}\right)
$$

(Relax $s$ to the maximum in each subcase) $\tag{83}$

For $n \geq \lfloor\log_2 k\rfloor + 1$, $[k] \cap (\frac{k}{2^{n+1}}, \frac{k}{2^n}] = \emptyset$, which means that the event $\left\{\exists s : s \in [k] \cap (\frac{k}{2^{n+1}}, \frac{k}{2^n}], \mathsf{kl}(\hat{\mu}_s, \mu) \geq \frac{2^{n+1}\ln(2^n T/k)}{k}\right\}$ cannot happen and its probability is 0 trivially. Therefore,

$$
(83) = \sum_{n=0}^{\lfloor\log_2 k\rfloor} \mathbb{P}\left(\exists s : s \in [k] \cap (\frac{k}{2^{n+1}}, \frac{k}{2^n}], \mathsf{kl}(\hat{\mu}_s, \mu) \geq \frac{2^{n+1}\ln\left(2^n T/k\right)}{k}\right) + \sum_{n=\lfloor\log_2 k\rfloor+1}^{\infty} 0
$$

$$
\leq \sum_{n=0}^{\lfloor\log_2 k\rfloor} \mathbb{P}\left(\exists s \geq \frac{k}{2^{n+1}}, \mathsf{kl}(\hat{\mu}_s, \mu) \geq \frac{2^{n+1}\ln\left(2^n T/k\right)}{k}\right)
$$

$$
= \sum_{n=0}^{\lfloor \log_2 k \rfloor} \mathbb{P}\left( \exists s \geq \lceil \frac{k}{2^{n+1}} \rceil, \mathsf{kl}(\hat{\mu}_s, \mu) \geq \frac{2^{n+1} \ln\left(2^n T/k\right)}{k} \right)
$$

$$
\leq \sum_{n=0}^{\lfloor \log_2 k \rfloor} \exp\left( -\lceil \frac{k}{2^{n+1}} \rceil \cdot \frac{2^{n+1} \ln\left(2^n T/k\right)}{k} \right) \qquad \text{(Maximal Inequality Lemma 25)}
$$

$$
= \sum_{n=0}^{\infty} \exp\left( -\ln \frac{2^n T}{k} \right)
$$

$$
= \sum_{n=0}^{\infty} \frac{k}{2^n T}
$$

$$
= \frac{2k}{T}
$$

The first inequality relies on the Lemma 25, for each choice of $n$, we set $y$ to be $\frac{2^{n+1} \ln\left(2^{n+1} T/k\right)}{k}$. $\quad \square$

The following lemma is standard in the literature, see e.g. [33]; we include a proof for completeness.

**Lemma 25.** *Given a natural number $N$ in $\mathbb{N}^+$, and a sequence of R.V.s $\{X_i\}_{i=1}^{\infty}$ is drawn from a distribution $\nu$ with bounded support $[0,1]$ and mean $\mu$. Let $\hat{\mu}_n = \frac{1}{n} \sum_{i=1}^{n} X_i, n \in \mathbb{N}$, which is the empirical mean of the first $n$ samples.*

*Then, for $y \geq 0$*

$$
\mathbb{P}(\exists n \geq N, \mathsf{kl}(\hat{\mu}_n, \mu) \geq y, \hat{\mu}_n < \mu) \leq \exp(-Ny) \tag{84}
$$

$$
\mathbb{P}(\exists n \geq N, \mathsf{kl}(\hat{\mu}_n, \mu) \geq y, \hat{\mu}_n > \mu) \leq \exp(-Ny) \tag{85}
$$

*Consequently, the following inequalities are also true:*

$$
\mathbb{P}(\hat{\mu}_N < \mu - \varepsilon) \leq \exp(-N \cdot \mathsf{kl}(\mu - \varepsilon, \mu)) \tag{86}
$$

$$
\mathbb{P}(\hat{\mu}_N > \mu + \varepsilon) \leq \exp(-N \cdot \mathsf{kl}(\mu + \varepsilon, \mu)) \tag{87}
$$

*Proof.* First, we prove a useful fact that for any $\lambda \in \mathbb{R}$, $S_n(\lambda) := \exp\left(n\hat{\mu}_n\lambda - ng_\mu(\lambda)\right)$ (abbrev. $S_n$) is a super-martingale sequence when $n \in \mathbb{N}^+$ and $n \geq N$, where $g_\mu(\lambda) := \ln\left(1 - \mu + \mu e^\lambda\right)$ is the log moment generating function of Bernoulli($\mu$).

Then, we have the following inequalities to finish the proof of the above fact:

$$
\begin{aligned}
\mathbb{E}\left[ S_{n+1} \mid S_n, \dots, S_1 \right] &= \mathbb{E}\left[ S_{n+1} \mid S_n \right] \\
&= \mathbb{E}\left[ \exp\left((n+1)\hat{\mu}_{n+1}\lambda - (n+1)g_\mu(\lambda)\right) \mid S_n \right] \\
&= \mathbb{E}\left[ S_n \cdot \exp\left(X_{n+1}\lambda - g_\mu(\lambda)\right) \mid S_n \right] \\
&= S_n \cdot \frac{\mathbb{E}\left[ \exp\left(X_{n+1}\lambda\right) \right]}{\exp\left(g_\mu(\lambda)\right)} \\
&\leq S_n \cdot \frac{1 - \mu + \mu e^\lambda}{1 - \mu + \mu e^\lambda} = S_n \qquad \text{(Lemma 17)}
\end{aligned}
$$

here, for the first equality, note that $S_{n+1}$, which is determined by $\hat{\mu}_{n+1}$ and $\hat{\mu}_{n+1}$ is conditionally independent of the trajectory up to time step $n-1$ given the condition $S_n$. The second and third equalities are due to the definitions of $S_{n+1}$ and $S_n$ respectively. In the first inequality, we apply Lemma 17 to upper bound the numerator $\mathbb{E}[\exp\left(X_{n+1}\lambda\right)]$ by $1 - \mu + \mu e^\lambda$.

We now prove Eq. (84) and Eq. (85) respectively.

For Eq. (84), we consider two cases:

**Case 1:** $y > \mathsf{kl}(0, \mu) = \ln \frac{1}{1-\mu}$. In this case, event $\mathsf{kl}(\hat{\mu}_n, \mu) \geq y$ can never happen. Therefore, LHS $= 0 \leq$ RHS.

**Case 2:** $y \leq \mathsf{kl}(0, \mu)$**.** In this case, there exists a unique $z_0 \in [0, \mu)$ such that $\mathsf{kl}(z_0, \mu) = y$. We denote $\lambda_0 := \ln \frac{z_0(1-\mu)}{(1-z_0)\mu} < 0$.

Observe that

$$y = \mathsf{kl}(z_0, \mu) = z_0 \lambda_0 - g_\mu(\lambda_0)$$

Therefore, LHS of Eq. (84) is equal to

$$\mathbb{P}(\exists n \geq N, \mathsf{kl}(\hat{\mu}_n, \mu) \geq y, \hat{\mu}_n < \mu) \tag{88}$$
$$= \mathbb{P}\left(\exists n \geq N, \hat{\mu}_n \leq z_0\right) \tag{89}$$
$$\leq \mathbb{P}\left(\exists n \geq N, n\hat{\mu}_n \lambda_0 - n g_\mu(\lambda_0) \geq n z_0 \lambda_0 - n g_\mu(\lambda_0)\right) \qquad (\lambda_0 < 0 \text{ and } \hat{\mu}_n \leq z_0)$$
$$\leq \mathbb{P}\left(\exists n \geq N, n\hat{\mu}_n \lambda_0 - n g_\mu(\lambda_0) \geq ny\right) \qquad (\text{By the definition of } z_0)$$
$$\leq \mathbb{P}\left(\exists n \geq N, \exp(n\lambda_0\hat{\mu}_n - n g_\mu(z_0)) \geq \exp(Ny)\right) \tag{90}$$
$$= \mathbb{P}(\exists n \geq N, S_n(\lambda_0) \geq \exp(Ny)) \tag{91}$$
$$\leq \frac{\mathbb{E}[S_N(\lambda_0)]}{\exp(Ny)} \leq \exp(-Ny) \qquad (\text{Ville's maximal inequality})$$

For Eq. (85), we consider two cases:

**Case 1:** $y > \mathsf{kl}(1, \mu) = \ln \frac{1}{\mu}$**.** In this case, event $\mathsf{kl}(\hat{\mu}_n, \mu) \geq y$ can never happen. Therefore, LHS $= 0 \leq$ RHS.

**Case 2:** $y \leq \mathsf{kl}(1, \mu)$**.** In this case, there exists a unique $z_1 \in (\mu, 1]$ such that $\mathsf{kl}(z_1, \mu) = y$. Let $\lambda_1 := \ln \left(\frac{z_1(1-\mu)}{(1-z_1)\mu}\right) > 0$. Observe that

$$y = \mathsf{kl}(z_1, \mu) = z_1 \lambda_1 - g_\mu(\lambda_1)$$

Then we have

$$\mathbb{P}(\exists n \geq N, \mathsf{kl}(\hat{\mu}_n, \mu) \geq y, \hat{\mu}_n > \mu)$$
$$= \mathbb{P}\left(\exists n \geq N, \hat{\mu}_n \geq z_1\right)$$
$$\leq \mathbb{P}\left(\exists n \geq N, n\hat{\mu}_n \lambda_1 - n g_\mu(\lambda_1) \geq n z_1 \lambda_1 - n g_\mu(\lambda_1)\right) \qquad (\lambda_1 > 0 \text{ and } \hat{\mu}_n \geq z_1)$$
$$\leq \mathbb{P}\left(\exists n \geq N, n\hat{\mu}_n \lambda_1 - n g_\mu(\lambda_1) \geq ny\right) \qquad (\text{By the definition of } z_1)$$
$$\leq \mathbb{P}\left(\exists n \geq N, \exp(n\lambda_1\hat{\mu}_n - n g_\mu(\lambda_1)) \geq \exp(Ny)\right)$$
$$= \mathbb{P}(\exists n \geq N, S_n(\lambda_1) \geq \exp(Ny))$$
$$\leq \frac{\mathbb{E}[S_N(\lambda_1)]}{\exp(Ny)} \leq \exp(-Ny) \qquad (\text{Ville's maximal inequality})$$

where the first inequality is due to the fact that $\lambda_1 > 0$ and the condition $\hat{\mu}_n \geq z_1$ which is equivalent to the event $\left\{\mathsf{kl}\left(\hat{\mu}_n, \mu\right) \geq \mathsf{kl}\left(z_1, \mu\right), \hat{\mu}_n > \mu\right\}$.

Finally we derive Eq. (86) and (87) from Eq. (84) and Eq. (85) respectively.

For Eq. (86), by letting $y = \mathsf{kl}(\mu - \varepsilon, \mu)$ we have that

$$\mathbb{P}\left(\hat{\mu}_N < \mu - \varepsilon\right) = \mathbb{P}\left(\mathsf{kl}(\hat{\mu}_N, \mu) > \mathsf{kl}(\mu - \varepsilon, \mu), \hat{\mu}_n < \mu\right) = \mathbb{P}\left(\mathsf{kl}(\hat{\mu}_N, \mu) > y, \hat{\mu}_n < \mu\right)$$
$$\leq \mathbb{P}\left(\exists n \geq N, \mathsf{kl}(\hat{\mu}_n, \mu) \geq y, \hat{\mu}_n < \mu)\right)$$
$$\leq \exp(-Ny) = \exp(-N \cdot \mathsf{kl}(\mu - \varepsilon, \mu))$$

For Eq. (87), by letting $y = \mathsf{kl}(\mu + \varepsilon, \mu)$ we have that

$$\mathbb{P}\left(\hat{\mu}_N > \mu + \varepsilon\right) = \mathbb{P}\left(\mathsf{kl}(\hat{\mu}_N, \mu) > \mathsf{kl}(\mu + \varepsilon, \mu), \hat{\mu}_n > \mu\right) = \mathbb{P}\left(\mathsf{kl}(\hat{\mu}_N, \mu) > y, \hat{\mu}_n > \mu\right)$$
$$\leq \mathbb{P}\left(\exists n \geq N, \mathsf{kl}(\hat{\mu}_n, \mu) \geq y, \hat{\mu}_n > \mu)\right)$$
$$\leq \exp(-Ny) = \exp(-N \cdot \mathsf{kl}(\mu + \varepsilon, \mu) \qquad \square$$

## E.8 Lower Bound of KL

**Lemma 26.** *Given a KL-divergence* $\mathsf{kl}(\mu_i, \mu_j)$ *between two Bernoulli distribution* $\nu(\mu_i)$ *and* $\nu(\mu_j)$ *where* $\mu_i, \mu_j \in [0,1]$. *Denote* $\dot{\mu}_i := \mu_i(1-\mu_i)$, $\dot{\mu}_j := \mu_j(1-\mu_j)$ *and* $\Delta := |\mu_j - \mu_i|$, *we have a lower bound to* $\mathsf{kl}(\mu_i, \mu_j)$.

$$\mathsf{kl}(\mu_i, \mu_j) \geq \frac{1}{4}\frac{\Delta^2}{\dot{\mu}_j + \Delta} \geq \frac{1}{8}\left(\frac{\Delta^2}{\dot{\mu}_i + \Delta} \vee \frac{\Delta^2}{\dot{\mu}_j + \Delta}\right)$$

*Proof.* It suffices to show $\mathsf{kl}(\mu_i, \mu_j) \geq \frac{1}{4}\left(\frac{\Delta^2}{\dot{\mu}_j + \Delta}\right)$ since by 1-Lipshizness of $z \mapsto z(1-z)$ we have $\dot{\mu}_i \leq \dot{\mu}_j + 1 \cdot |\mu_i - \mu_j| = \dot{\mu}_j + \Delta$ and $\frac{1}{4}\left(\frac{\Delta^2}{\dot{\mu}_i + \Delta}\right) \geq \frac{1}{4}\left(\frac{\Delta^2}{\dot{\mu}_j + 2\Delta}\right) \geq \frac{1}{8}\left(\frac{\Delta^2}{\dot{\mu}_j + \Delta}\right)$. Then we split $\mu_j$ into two cases.

**Case 1**: $\mu_j \leq \frac{1}{2}$.

In this case, $\mu_j = \mu_j \cdot 1 \leq \mu_j \cdot 2(1 - \mu_j) = 2\dot{\mu}_j$.

$$\mathsf{kl}(\mu_i, \mu_j) \geq \frac{\Delta^2}{2(\mu_i \vee \mu_j)} = \frac{\Delta^2}{2(\mu_j + \Delta)} \geq \frac{\Delta^2}{2(2\dot{\mu}_j + \Delta)} \geq \frac{\Delta^2}{4\dot{\mu}_j + 2\Delta} \geq \frac{1}{4}\left(\frac{\Delta^2}{\dot{\mu}_j + \Delta}\right)$$

**Case 2**: $\mu_j > \frac{1}{2}$

In this case, we have $1 - \mu_j \leq (1 - \mu_j)2\mu_j = 2\dot{\mu}_j$.

Using the following inequality

$$\begin{aligned}
&\mathsf{kl}(\mu_i, \mu_j)\\
=&\mathsf{kl}(1 - \mu_i, 1 - \mu_j)\\
\geq&\frac{\Delta^2}{2\left((1-\mu_i) \vee (1-\mu_j)\right)}\\
\geq&\frac{\Delta^2}{2(1-\mu_i)}\\
\geq&\frac{\Delta^2}{2(1-\mu_j+\Delta)}\\
\geq&\frac{\Delta^2}{2\left(2\dot{\mu}_j + \Delta\right)}\\
=&\frac{\Delta^2}{4\dot{\mu}_j + 2\Delta}\\
\geq&\frac{1}{4}\left(\frac{\Delta^2}{\dot{\mu}_j + \Delta}\right)
\end{aligned}$$
$\square$

## E.9 Algebraic Lemmas

**Lemma 27.** *Let* $q \geq p > 0$ *and* $b > 0$, *and define* $f_{p,q}(x) := \frac{\ln(bx^p \vee e^q)}{x}$. *Then* $f(x)$ *is monotonically decreasing in* $\mathbb{R}_+$. *Specifically, both* $f_{1,2}(x) := \frac{\ln(bx \vee e^2)}{x}$ *and* $f_{2,2}(x) := \frac{\ln(bx^2 \vee e^2)}{x}$ *are monotonically decreasing.*

*Proof.* Note that

$$f_{p,q}(x) = \begin{cases} \frac{q}{x} & bx^p \leq e^q \\ \frac{\ln(bx^p)}{x} & bx^p > e^q \end{cases}$$

**Algorithm 2** The KL-UCB algorithm (taken from Lattimore and Szepesvári [29, Section 10.2])

1: **Input:** $K \geq 2$
2: **for** $t = 1, 2, \cdots, n$ **do**
3:   **if** $t \leq K$ **then**
4:     Pull the arm $I_t = t$ and observe reward $y_t \sim \nu_i$.
5:   **else**
6:     For every $a \in [K]$, compute

$$\mathrm{UCB}_t(a) = \max \left\{ \mu \in [0,1] : \mathsf{kl}(\hat{\mu}_{t-1,a}, \mu) \leq \frac{\ln f(t)}{N_{t-1,a}} \right\},$$

     where $f(t) = 1 + t \ln^2 t$.
7:     Choose arm $I_t = \mathrm{argmax}_{a \in [K]} \mathrm{UCB}_t(a)$
8:     Receive reward $y_t \sim \nu_{I_t}$
9:   **end if**
10: **end for**

---

- When $x \in (0, \frac{e^{\frac{q}{p}}}{b^{\frac{1}{p}}})$, $bx^p < e^q$. In this case, $f_{p,q}$ is monotonically decreasing as $f_{p,q}(x)$ is inverse proportional to $x$.

- When $x \in [\frac{e^{\frac{q}{p}}}{b^{\frac{1}{p}}}, +\infty)$, $bx^p \geq e^q$. In this case,

$$f'_{p,q}(x) = \frac{p - \ln(bx^p)}{x^2} \leq \frac{p - q}{x^2} \leq 0,$$

which implies that $f_{p,q}$ is also monotonically decreasing in this region. $\quad\square$

**Lemma 28.** *For $C \geq 1$ and $a > 0$,*
$$\ln(Ca \vee e^2) \leq C \ln(a \vee e^2)$$

*Proof.* From Lemma 27, $f_{1,2}(x) := \frac{\ln(bx \vee e^2)}{x}$ is monotonically decreasing. Therefore, we have
$$\frac{\ln(Ca \vee e^2)}{Ca} \leq \frac{\ln(a \vee e^2)}{a},$$
this yields the lemma. $\quad\square$

### E.10  Bregman divergence identity

**Lemma 29** (Lemma 6.6 in Orabona [37]). *Let $B_\phi$ the Bregman divergence w.r.t. $\phi : X \to \mathbb{R}$. Then, for any three points $x, y \in \mathrm{interior}(X)$ and $z \in X$, the following equality holds:*
$$B_\phi(z, x) + B_\phi(x, y) - B_\phi(z, y) = \langle \nabla\phi(y) - \nabla\phi(x), z - x \rangle,$$
*where $B_\phi(z, x) := \phi(z) - \phi(x) - \langle \nabla\phi(x), z - x \rangle$.*

## F  Refined worst-case guarantees for existing algorithms

### F.1  KL-UCB's refined regret guarantee

In this section, we show that KL-UCB [13] also can enjoy a worst-case regret bound of the form $\sqrt{\hat{\mu}_1 TK \ln T}$ in the bandits with $[0, 1]$ bounded reward setting. We first recall the KL-UCB algorithm, Algorithm 2, and we take the version of [29, Section 10.2].

The following theorem is a refinement of the guarantee of KL-UCB in [29, Theorem 10.6].

**Theorem 30** (KL-UCB: refined guarantee). *For any $K$-arm bandit problem with reward distributions supported on $[0, 1]$, KL-UCB (Algorithm 2) has regret bounded as follows. For any $\Delta > 0$ and*

$c \in (0, \frac{1}{4}]$,

$$\text{Reg}(T) \leq T\Delta + \sum_{a:\Delta_a > \Delta} \frac{\Delta_a \ln(1 + T \ln^2 T)}{\mathsf{KL}(\mu_a + c\Delta_a, \mu_1 - c\Delta_a)} + O\left( \sum_{a:\Delta_a > \Delta} \frac{\dot{\mu}_1 + \Delta_a}{c^2 \Delta_a} \right). \tag{92}$$

*and consequently,*

$$\text{Reg}(T) \leq O\left( \sqrt{\dot{\mu}_1 TK \ln T} + K \ln T \right). \tag{93}$$

*Proof sketch.* To show Eq. (92), fix any suboptimal arm $a$; it suffices to show that

$$\mathbb{E}\left[N_{T,a}\right] \leq \frac{\Delta_a \ln(1 + T \ln^2 T)}{\mathsf{KL}(\mu_a + c\Delta_a, \mu_1 - c\Delta_a)} + O\left( \sum_{a:\Delta_a > \Delta} \frac{\dot{\mu}_1 + \Delta_a}{c^2 \Delta_a} \right). \tag{94}$$

To this end, following Lattimore and Szepesvári [29, proof of Theorem 10.6], let $\varepsilon_1, \varepsilon_2 > 0$ be such that $\varepsilon_1 + \varepsilon_2 < \Delta_a$.

Define

$$\tau = \min\left\{ t : \max_{s \in \{1,\dots,T\}} \mathsf{kl}(\hat{\mu}_{1,(s)}, \mu_1 - \varepsilon_2) - \frac{\ln f(t)}{s} \leq 0 \right\},$$

and

$$\kappa = \sum_{s=1}^{T} \mathbf{1}\left\{ \mathsf{kl}(\hat{\mu}_{a,(s)}, \mu_1 - \varepsilon_2) \leq \frac{\ln f(T)}{s} \right\}.$$

A close examination of Lattimore and Szepesvári [29, proof of Lemma 10.7] reveals that a stronger bound on $\mathbb{E}[\tau]$ holds, i.e.,

$$\mathbb{E}[\tau] \leq \frac{2}{\mathsf{kl}(\mu_1 - \varepsilon_2, \mu_1)}$$

and similarly, a close examination of Lattimore and Szepesvári [29, proof of Lemma 10.8] reveals that a stronger bound on $\mathbb{E}[\kappa]$ holds,

$$\mathbb{E}[\kappa] \leq \frac{\ln f(T)}{\mathsf{kl}(\mu_a + \varepsilon_1, \mu_1 - \varepsilon_2)} + \frac{1}{\mathsf{kl}(\mu_a + \varepsilon_1, \mu_a)}$$

Therefore, by Lattimore and Szepesvári [29, proof of Theorem 10.6], we have

$$\mathbb{E}\left[N_{T,a}\right] \leq \mathbb{E}[\tau] + \mathbb{E}[\kappa] \leq \frac{\ln f(T)}{\mathsf{kl}(\mu_a + \varepsilon_1, \mu_1 - \varepsilon_2)} + \frac{1}{\mathsf{kl}(\mu_a + \varepsilon_1, \mu_a)} + \frac{2}{\mathsf{kl}(\mu_1 - \varepsilon_2, \mu_1)} \tag{95}$$

We now set $\varepsilon_1 = \varepsilon_2 = c\Delta_a$. Observe that by Lemma 26,

$$\frac{1}{\mathsf{kl}(\mu_a + \varepsilon_1, \mu_a)} \lesssim \frac{\dot{\mu}_a + \varepsilon_1}{\varepsilon_1^2} \lesssim \frac{\dot{\mu}_1 + \Delta_a}{c^2 \Delta_a^2},$$

and

$$\frac{2}{\mathsf{kl}(\mu_1 - \varepsilon_2, \mu_1)} \lesssim \frac{\dot{\mu}_1 + \varepsilon_2}{\varepsilon_2^2} \lesssim \frac{\dot{\mu}_1 + \Delta_a}{c^2 \Delta_a^2}$$

Plugging these two inequalities into Eq. (95) yields Eq. (94).

As for Eq. (93), we note that $\frac{1}{\mathsf{kl}(\mu_a + c\Delta_a, \mu_a - c\Delta_a)} \lesssim \frac{\dot{\mu}_1 + \Delta_a}{c^2 \Delta_a^2}$, and therefore, Eq. (92) implies that for any $\Delta > 0$,

$$\text{Reg}(T) \leq \Delta T + \sum_{a:\Delta_a > \Delta} \frac{\dot{\mu}_1 + \Delta_a}{c^2 \Delta_a} \ln f(T)$$

$$\leq \Delta T + K \frac{\dot{\mu}_1 + \Delta}{c^2 \Delta} \ln f(T)$$

Choosing $\Delta = \sqrt{\dot{\mu}_1 \frac{K \ln f(T)}{T}}$ yields Eq. (93). $\qquad\square$

## F.2 KL-UCB++'s refined regret guarantee

In this section, we show a worst-case regret guarantee of KL-UCB++ of order $\tilde{O}\left(\sqrt{\dot{\mu}_1 K^3 T \ln T} + K^2 \ln T\right)$ by adapting the original KL-UCB++ analysis (Theorem 2 of [33]).

First, we derive a refined bound of the number of suboptimal arm pulling, corresponding to Eq. (24) in [33], which we state in the following theorem.

**Theorem 31** (KL-UCB++: refined upper bound of suboptimal arm pulling). *For any suboptimal arm $a$, the expected number of its pulling up to time step $T$, namely $\mathbb{E}\left[N_a(T)\right]$, is bounded by*

$$\mathbb{E}\left[N_a(T)\right] \leq \frac{\ln(T)}{\mathsf{kl}(\mu_a + \delta, \mu_1 - \delta)} + O\left(\frac{(K + \ln\ln(T))(\dot{\mu}_1 + \Delta_a)}{\delta^2}\right), \tag{96}$$

*for any $\delta \in [\frac{88K}{T} + \sqrt{\frac{88\dot{\mu}_1 K}{T}}, \frac{\Delta_a}{3}]$.*

*Proof.* First we decompose the expected number of arm pulling w.r.t. suboptimal arm $a$, $\mathbb{E}\left[N_a(T)\right]$ as

$$\mathbb{E}\left[N_a(T)\right] \leq 1 + \underbrace{\sum_{t=K}^{T-1} \mathbb{P}\left(U_1(t) \leq \mu_1 - \delta\right)}_{A} + \underbrace{\sum_{t=K}^{T-1} \mathbb{P}\left(\mu_1 - \delta < U_a(t) \text{ and } I_{t+1} = a\right)}_{B}$$

Following [33], we can bound each term in $A$ as:

$$\mathbb{P}\left(U_1(t) \leq \mu_1 - \delta\right)$$

$$\leq \underbrace{\mathbb{P}\left(\exists 1 \leq n \leq f(\delta), \hat{\mu}_{1,n} \leq \mu_1, \mathsf{kl}(\hat{\mu}_{1,n}, \mu_1) \geq \frac{g(n)}{n}\right)}_{A_1} + \underbrace{\mathbb{P}\left(\exists f(\delta) \leq n \leq T, \hat{\mu}_{1,n} \leq \mu_1 - \delta\right)}_{A_2},$$

here, with foresight, we choose $f(\delta) = \frac{1}{\mathsf{kl}(\mu_1-\delta,\mu_1)} \ln \frac{\mathsf{kl}(\mu_1-\delta,\mu_1)T}{K}$.

Note that $A_2 \leq \exp(-f(\delta)\mathsf{kl}(\mu_1 - \delta, \mu_1))$ by the maximal inequality (Lemma 25).

$$A_2 \leq \frac{K}{T\mathsf{kl}(\mu_1 - \delta, \mu_1)}. \tag{97}$$

For bounding $A_1$, we rely on the following inequality borrowed from [33, page 7]: for any $N$ such that $\frac{T}{KN} \geq e^{3/2}$,[5]

$$\mathbb{P}\left(\exists 1 \leq n \leq N, \hat{\mu}_{1,n} \leq \mu_1, \mathsf{kl}(\hat{\mu}_{1,n}, \mu_1) \geq \frac{g(n)}{n}\right)$$

$$\leq 4e^2 \frac{\ln(\frac{T}{KN}(1 + \ln^2(\frac{T}{KN})))}{\ln(\frac{T}{KN})} \cdot \frac{N\mathsf{kl}(\mu_1 - \delta, \mu_1)}{\ln(\frac{T}{KN})} \cdot \frac{K}{T\mathsf{kl}(\mu_1 - \delta, \mu_1)}.$$

Therefore, setting $N = f(\delta)$ we have the following inequality when $\frac{T}{Kf(\delta)} \geq e^{3/2}$:

$$\mathbb{P}\left(\exists 1 \leq n \leq f(\delta), \hat{\mu}_{1,n} \leq \mu_1, \mathsf{kl}(\hat{\mu}_{1,n}, \mu_1) \geq \frac{g(n)}{n}\right)$$

$$\leq 4e^2 \underbrace{\frac{\ln(\frac{T}{Kf(\delta)}(1 + \ln^2(\frac{T}{Kf(\delta)})))}{\ln(\frac{T}{Kf(\delta)})}}_{C} \cdot \underbrace{\frac{f(\delta)\mathsf{kl}(\mu_1 - \delta, \mu_1)}{\ln(\frac{T}{Kf(\delta)})}}_{D} \cdot \frac{K}{T\mathsf{kl}(\mu_1 - \delta, \mu_1)}.$$

Also, based on the assumption that $\delta \geq \frac{88K}{T} + \sqrt{\frac{88\dot{\mu}_1 K}{T}}$, we have that $\frac{T\mathsf{kl}(\mu_1-\delta,\mu_1)}{K} \geq e^{3/2}$ and $\frac{T}{Kf(\delta)} > 1$ (we defer the justification at the end of this paragraph). Now:

---

[5]we only replaced their $f(u)$ with $N$. The proof still goes through since the proof has no assumption except $\frac{T}{Kf(u)} \geq e^{3/2}$.

- For $C$, we apply the elementary inequality that $\frac{\ln(x(1+\ln^2 x))}{\ln x} \le 2$ for $x > 1$ with $x = \frac{T}{Kf(\delta)}$; therefore, $C \le 2$.

- For $D = \frac{\ln\frac{\mathsf{kl}(\mu_1-\delta)T}{K}}{\ln\left(\frac{\mathsf{kl}(\mu_1-\delta)T}{K}/\ln\frac{\mathsf{kl}(\mu_1-\delta)T}{K}\right)}$, we apply the elementary inequality that $\frac{\ln(x)}{\ln(x/\ln x)} \le 2$ for $x \ge e^{3/2}$ with $x = \frac{T\mathsf{kl}(\mu_1-\delta,\mu_1)}{K}$; therefore, $D \le 2$.

Now we are going to justify the condition that $\delta \ge \frac{88K}{T} + \sqrt{\frac{88\dot{\mu}_1 K}{T}}$ ensures these two elementary inequalities being true. In proving $\frac{T\mathsf{kl}(\mu_1-\delta,\mu_1)}{K} \ge e^{3/2}$, we use the KL lower bound lemma (lemma 26). More specifically,

$$\frac{T\mathsf{kl}(\mu_1-\delta,\mu_1)}{K} \ge e^{3/2} \tag{98}$$

$$\Longleftarrow \frac{T\delta^2}{4K(\dot{\mu}_1+\delta)} \ge e^{3/2} \tag{Lemma 26}$$

$$\Longleftarrow \delta^2 \ge \frac{44K(\dot{\mu}_1+\delta)}{T} \tag{99}$$

$$\Longleftarrow \delta^2 \ge 2 \cdot \max\left\{\frac{44\dot{\mu}_1 K}{T}, \frac{44K\delta}{T}\right\} \tag{100}$$

$$\Longleftarrow \delta \ge \max\left\{\frac{88K}{T}, \sqrt{\frac{88\dot{\mu}_1 K}{T}}\right\} \tag{101}$$

$$\Longleftarrow \delta \ge \frac{88K}{T} + \sqrt{\frac{88\dot{\mu}_1 K}{T}} \tag{102}$$

In summary, from the above derivation, $\delta \ge \frac{88K}{T} + \sqrt{\frac{88\dot{\mu}_1 K}{T}}$ implies that $\frac{T\mathsf{kl}(\mu_1-\delta,\mu_1)}{K} \ge e^{3/2}$. In this case, furthermore we have $\frac{T}{Kf(\delta)} = \frac{T\mathsf{kl}(\mu_1-\delta,\mu_1)/K}{\ln(T\mathsf{kl}(\mu_1-\delta,\mu_1)/K)} \ge \frac{2}{3}e^{3/2} > 1$.

Therefore we bound $A_1$ by

$$A_1 \le 4e^2 \cdot 2 \cdot 2 \cdot \frac{K}{T\mathsf{kl}(\mu_1-\delta,\mu_1)} \le 16e^2 \frac{K}{T\mathsf{kl}(\mu_1-\delta,\mu_1)}. \tag{103}$$

Combining Eq (97) and (103), we derive the upper bound for $A$:

$$A \le \sum_{t=K}^{T-1}\left(16e^2+1\right)\frac{K}{T\mathsf{kl}(\mu_1-\delta,\mu_1)} \le \left(16e^2+1\right)\frac{K}{\mathsf{kl}(\mu_1-\delta,\mu_1)} \le O\left(\frac{K(\dot{\mu}_1+\delta)}{\delta^2}\right), \tag{104}$$

where in the last inequality we use Lemma 26.

To bound $B$, we reuse the same idea in [33] but change the definition of $n(\delta)$ to accommodate our new analysis,

$$n(\delta) = \left\lceil \frac{\ln\left(\frac{T}{K}\left(1+\ln^2(\frac{T}{K})\right)\right)}{\mathsf{kl}(\mu_a+\delta,\mu_1-\delta)} \right\rceil$$

applying the same analysis in [33] (specifically, from their Eq. (28) to Eq.(29)), we bound $B$ by

$$B \le n(\delta)-1+\sum_{n=n(\delta)}^{T}\mathbb{P}\left(\mathsf{kl}\left(\hat{\mu}_{a,(n)},\mu_1-\delta\right) \le \mathsf{kl}(\mu_a+\delta,\mu_1-\delta)\right) \tag{105}$$

$$\le n(\delta)-1+\sum_{n=n(\delta)}^{T}\mathbb{P}\left(\hat{\mu}_{a,(n)} \ge \mu_a+\delta\right) \tag{106}$$

$$\leq n(\delta) - 1 + \sum_{n=1}^{T} \exp\left(-n\mathsf{kl}(\mu_a + \delta, \mu_a)\right) \qquad \text{(Lemma 25)}$$

$$\leq n(\delta) - 1 + \frac{1}{\exp\left(\mathsf{kl}(\mu_a + \delta, \mu_a)\right) - 1} \qquad \text{(Geometric sum)}$$

$$\leq n(\delta) - 1 + \frac{1}{\mathsf{kl}(\mu_a + \delta, \mu_a)} \qquad (e^x \geq x + 1 \text{ when } x \geq 0)$$

$$\leq \frac{\ln\left(\frac{T}{K}\left(1 + \ln^2\left(\frac{T}{K}\right)\right)\right)}{\mathsf{kl}\left(\mu_a + \delta, \mu_1 - \delta\right)} + \frac{4\left(\dot{\mu}_a + \delta\right)}{\delta^2} \qquad \text{(Lemma 26)}$$

$$= \frac{\ln(T)}{\mathsf{kl}\left(\mu_a + \delta, \mu_1 - \delta\right)} + \frac{\ln\left(\frac{1}{K}\left(1 + \ln^2\left(\frac{T}{K}\right)\right)\right)}{\mathsf{kl}\left(\mu_a + \delta, \mu_1 - \delta\right)} + \frac{4\left(\dot{\mu}_1 + \Delta_a + \delta\right)}{\delta^2}$$
$$\text{(By the 1-Lipshitzness of } \mu \mapsto \dot{\mu})$$

$$\leq \frac{\ln(T)}{\mathsf{kl}\left(\mu_a + \delta, \mu_1 - \delta\right)} + O\left(\frac{\ln\left(\frac{1}{K}\left(1 + \ln^2\left(\frac{T}{K}\right)\right)\right)}{\delta^2/(\dot{\mu}_1 + \Delta_a)} + \frac{4\left(\dot{\mu}_1 + \Delta_a + \delta\right)}{\delta^2}\right) \qquad \text{(Lemma 26)}$$

$$\leq \frac{\ln(T)}{\mathsf{kl}\left(\mu_a + \delta, \mu_1 - \delta\right)} + O\left(\ln\ln T \cdot \frac{\dot{\mu}_1 + \Delta_a}{\delta^2}\right). \qquad (107)$$

Combining Eq.(104) and Eq.(107), we get the final inequality Eq.(96). $\qquad \square$

Based on the above refinement and replace $\delta$ by $c\Delta_a$, we can have the following theorem.

**Theorem 32** (KL-UCB++: refined guarantee). *For any $K$-arm bandit problem with reward distributions supported on $[0, 1]$, KL-UCB++ has regret bounded as follows:*

$$\mathrm{Reg}(T) \leq O\left(\sqrt{\dot{\mu}_1 T K^3 \ln T} + K^2 \ln T\right). \qquad (108)$$

*Proof.* Define $S = \left\{a \in [K] : \frac{88K}{T} + \sqrt{\frac{88\dot{\mu}_1 K}{T}} \leq \frac{\Delta_a}{3}\right\}$. For $a \in S$, applying Theorem 31 with $\delta = \frac{\Delta_a}{3}$, and observe that by Lemma 26,

$$\frac{1}{\mathsf{kl}(\mu_a + \delta, \mu_1 - \delta)} \lesssim \frac{\dot{\mu}_a + \delta}{\delta^2} \lesssim \frac{\dot{\mu}_1 + \Delta_a}{\Delta_a^2},$$

we get:

$$\mathbb{E}[N_a(T)] \lesssim \frac{(\dot{\mu}_1 + \Delta_a) \ln T}{\Delta_a^2} + O\left(\frac{(K + \ln\ln(T))(\dot{\mu}_1 + \Delta_a)}{\Delta_a^2}\right) \qquad (109)$$

$$\lesssim O\left(\frac{(K + \ln T)(\dot{\mu}_1 + \Delta_a)}{\Delta_a^2}\right) \qquad (110)$$

Therefore, for any $\Delta > 0$, the regret given a timespan of $T$ is bounded by

$$\mathrm{Reg}(T) \leq \sum_{a:\Delta_a \leq \Delta} \Delta_a \mathbb{E}\left[N_a(T)\right] + \sum_{a:\Delta_a > \Delta, a \in S} \Delta_a \mathbb{E}\left[N_a(T)\right] + \sum_{a:\Delta_a > \Delta, a \notin S} \Delta_a \mathbb{E}\left[N_a(T)\right]$$

$$\leq T\Delta + \sum_{a:\Delta_a > \Delta, a \in S} O\left(\frac{(K + \ln T)(\dot{\mu}_1 + \Delta_a)}{\Delta_a}\right) + \sum_{a:\Delta_a > \Delta, a \notin S} O\left(T\left(\frac{K}{T} + \sqrt{\frac{\dot{\mu}_1 K}{T}}\right)\right)$$

$$\leq T\Delta + O\left(\frac{K(K + \ln T)(\dot{\mu}_1 + \Delta)}{\Delta}\right) + O\left(K^2 + \sqrt{\dot{\mu}_1 K^3 T}\right)$$

Choosing $\Delta = \sqrt{\frac{\dot{\mu}_1 K (K + \ln(T))}{T}}$ yields Eq. (108). $\qquad\qquad\square$

### F.3 The worst-case regret bound of UCB-V

In this section, we will show that the problem dependent regret bound presented in UCB-V[7] can also be adaptive to $\dot{\mu}_1$ in the bandits with $[0, 1]$ bounded reward setting. The starting point is that we will obtain a lemma (Lemma 33) to bound the arm pulling for all suboptimal arms like what we did in our paper.

**Lemma 33.** *Let $N_i(T)$ to be the number of the arm pulling in terms of the arm $i$ until the time step $T$ (inclusively) in the algorithm UCB-V from [7]. Then we can bound $\mathbb{E}\left[N_i(T)\right]$ by the following inequality*

$$\mathbb{E}\left[N_i(T)\right] \lesssim \left(\frac{\dot{\mu}_i^2}{\Delta_i^2} + \frac{1}{\Delta_i}\right) \log T \tag{111}$$

*Proof.* Inside the proof of Theorem 3 in [7], by setting $c = 1$, for each arm $i$, we obtain the following inequality for any $\zeta > 0$:

$$\mathbb{E}\left[N_i(T)\right] \leq 1 + 8\mathcal{E}_T \left(\frac{\dot{\mu}_i^2}{\Delta_i^2} + \frac{2}{\Delta_i}\right) + Te^{-\mathcal{E}_T}\left(\frac{24\dot{\mu}_i}{\Delta_i^2} + \frac{4}{\Delta_i}\right) + \sum_{t=u+1}^{T} \beta\left(\mathcal{E}_t, T\right), \tag{112}$$

where $u := \lceil 8\zeta\left(\frac{\dot{\mu}_k^2}{\Delta_k^2} + \frac{2}{\Delta_k}\right)\log T\rceil$, $\mathcal{E}_T := \zeta \log T$ and $\beta\left(\mathcal{E}_t, t\right) := \inf_{1 < \alpha \leq 3}\left(\frac{\log t}{\log \alpha} \wedge t\right) e^{-\frac{\mathcal{E}_t}{\alpha}}$. We pick $\zeta = 1.1$. The last term is bounded by

$$\sum_{t=u+1}^{T} \beta\left(\mathcal{E}_t, t\right) \leq \sum_{t=u+1}^{T} 3 \cdot \inf_{1 < \alpha \leq 3}\left(\frac{\log t}{\log \alpha} \wedge t\right) e^{-\frac{\mathcal{E}_t}{\alpha}} \leq \sum_{t=u+1}^{T} 3 \cdot \frac{\log t}{\log(1.1)} e^{-\frac{\mathcal{E}_t}{1.1}} \tag{113}$$

$$\leq \frac{3}{\log(1.1)} \sum_{t=u+1}^{T} \frac{\log t}{t^{1.1}} \lesssim \sum_{t=1}^{\infty} \frac{\log t}{t^{1.1}} \lesssim 1 \tag{114}$$

Therefore, we have the following inequality

$$\mathbb{E}\left[N_i(T)\right] \lesssim \left(\frac{\dot{\mu}_i^2}{\Delta_i^2} + \frac{1}{\Delta_i}\right) \log T \tag{115}$$

$\qquad\qquad\square$

By using the lemma 33 we just obtained, we can obtain the following theorem about worst-case regret bound of UCB-V.

**Theorem 34.** *The regret of the algorithm UCB-V[7] is bounded by:*

$$\mathrm{Reg}(T) \lesssim \sqrt{\dot{\mu}_1 K T \ln(T)} + K \ln(T) \tag{116}$$

*Proof.*

$$\mathrm{Reg}(T) = \sum_{i:\Delta_i \leq \Delta} \Delta_i \mathbb{E}[N_i(T)] + \sum_{i:\Delta_i > \Delta} \Delta_i \mathbb{E}[N_i(T)]$$

$$\leq T\Delta + \sum_{i:\Delta_i > \Delta} \Delta_i \mathbb{E}[N_i(T)]$$

$$\lesssim T\Delta + \sum_{i:\Delta_i > \Delta} \left(\frac{\dot{\mu}_i}{\Delta_i} + 1\right)\log(T) \qquad\qquad \text{(By Eq. (115))}$$

$$\leq T\Delta + \sum_{i:\Delta_i \in [\Delta, 1/4]} \left(\frac{\dot{\mu}_i}{\Delta_i} + 1\right)\log(T) + \sum_{i:\Delta_i > 1/4} \left(\frac{\dot{\mu}_i}{\Delta_i} + 1\right)\log(T)$$

$$\lesssim T\Delta + \sum_{i:\Delta_i\in[\Delta,1/4]} \frac{\dot\mu_i}{\Delta_i}\log(T) + K\log(T) \ .$$

To bound the second term above, we consider two cases.

**Case 1**: $\mu_1 < \frac{3}{4}$ .

In this case, one can show that $\dot\mu_i \lesssim \dot\mu_1$. Thus,

$$\sum_{i:\Delta_i\in[\Delta,1/4]} \frac{\dot\mu_i}{\Delta_i}\ln(T) \lesssim K\frac{\dot\mu_1}{\Delta}\ln(T) \ .$$

**Case 2**: $\mu_1 \geq \frac{3}{4}$ .

We observe that if $i$ satisfies $\Delta_i \in [\Delta, 1/4]$, then $\dot\mu_i = \mu_i(1-\mu_i) \leq 1-\mu_i = 1-\mu_i+\mu_1-\mu_1 = 1-\mu_1+\Delta_i \lesssim \dot\mu_1 + \Delta_i$. Thus,

$$\sum_{i:\Delta_i\in[\Delta,1/4]} \frac{\dot\mu_i}{\Delta_i}\ln(T) \lesssim \sum_{i:\Delta_i\in[\Delta,1/4]} \left(\frac{\dot\mu_1}{\Delta_i}+1\right)\ln(T) \leq K\frac{\dot\mu_1}{\Delta}\ln(T) + K\ln(T) \ .$$

Altogether, we have

$$\mathrm{Reg}(T) \lesssim T\Delta + K\frac{\dot\mu_1}{\Delta}\ln(T) + K\ln(T) \ .$$

Let us choose $\Delta = \sqrt{\frac{K\dot\mu_1}{T}} \wedge \frac{1}{4}$. If $T > K\dot\mu_1$, then we obtain the desired bound. If $T \leq K\dot\mu_1$, we get $\Delta = 1/4$, so

$$\mathrm{Reg}(T) \lesssim n + K\dot\mu_1\ln(T) + K\ln(T) \leq K\dot\mu_1 + K\dot\mu_1\ln(T) + K\ln(T) \ ,$$

which is less than the desired bound. This concludes the proof. $\qquad\square$

## G  Improved minimax analysis of the sub-Gaussian MS

We sketch how to change the proof of the sub-Gaussian MS regret bound in Bian and Jun [11] so it can achieve the minimax ratio of $\sqrt{\ln(K)}$.

It suffices to show that $\forall a : \mu_a < \mu_1, \mathbb{E}[N_{T,a}] \lesssim \frac{\sigma^2}{\varepsilon^2}\ln(\frac{T\varepsilon^2}{\sigma^2} \vee e^2)$. To bound $\mathbb{E}[N_{T,a}]$, recall that there are three terms to bound: $(F1)$, $(F2)$, and $(F3)$. Recall the symbols in Bian and Jun [11]:

- $\sigma^2$: the sub-Gaussian parameter.
- $u := \left\lceil \frac{2\sigma^2(1+c)^2\ln(T\Delta_a^2/(2\sigma^2)\vee e^2)}{\Delta_a^2} \right\rceil$ for some $c > 0$.
- $\varepsilon > 0$: an analysis parameter that will be chosen later to be $\Delta_a$ up to a constant factor.

The reason why one does not obtain the minimax ratio of $\sqrt{\ln(K)}$ is that the bound obtained in Bian and Jun [11] for $(F3)$ is $O(\frac{\sigma^2}{\varepsilon^2}\ln(\frac{\sigma^2}{\varepsilon^2} \vee e^2))$ rather than $O(\frac{\sigma^2}{\varepsilon^2}\ln(\frac{T\varepsilon^2}{\sigma^2} \vee e^2))$. To achieve the latter bound for $(F3)$, first we choose the splitting threshold $\frac{\sigma^2}{\varepsilon^2}$ which takes the same role as $H$ for KL-MS in the $[0,1]$-bounded reward case and $F3$ will be separated into $F3_1$ and $F3_2$. $F3_1$ is the case where $F3$ is with the extra condition that $N_{t-1,1} \leq \frac{\sigma^2}{\varepsilon^2}$ for $1 \leq t \leq T$ and $F3_2$ the case where $F3$ is with the extra condition that $N_{t-1,1} > \frac{\sigma^2}{\varepsilon^2}$ for $1 \leq t \leq T$. It is easy to bound $F3_2$ using a similar argument as our Claim 15 that $F3_2 \lesssim \frac{\sigma^2}{\varepsilon^2}$.

For $F3_1$, we define the following event

$$\mathcal{E} := \left\{ \forall k \in [1, \lfloor\frac{\sigma^2}{\varepsilon^2}\rfloor], \hat\mu_{(k),1} \geq \mu_1 - \sqrt{\frac{4\sigma^2\ln(T/k)}{k}} \right\}$$

where $\hat\mu_{(k),1}$ is the empirical mean of arm 1 (the true best arm) after $k$ arm pulls.

We have

$$F3_1 = \mathbb{E}\left[\sum_{t=K+1}^{T} \mathbf{1}\left\{I_t = a, N_{t-1,a} > u, \hat{\mu}_{t-1,\max} < \mu_1 - \varepsilon, N_{t-1,1} \le \frac{\sigma^2}{\varepsilon^2}\right\}\right]$$

$$= \mathbb{E}\left[\sum_{t=K+1}^{T} \mathbf{1}\left\{I_t = a, N_{t-1,a} > u, \hat{\mu}_{t-1,\max} < \mu_1 - \varepsilon, N_{t-1,1} \le \frac{\sigma^2}{\varepsilon^2}, \mathcal{E}\right\}\right] + \mathbb{E}\left[\sum_{t=K+1}^{T-1} \mathbf{1}\left\{\mathcal{E}^c\right\}\right]$$

$$\le \mathbb{E}\left[\sum_{t=K+1}^{T} \mathbf{1}\left\{I_t = a, N_{t-1,a} > u, \hat{\mu}_{t-1,\max} < \mu_1 - \varepsilon, \mathcal{E}\right\}\right] + T \cdot \mathbb{P}(\mathcal{E}^c)$$

Note that one can show that $T \cdot \mathbb{P}(\mathcal{E}^c) \lesssim \frac{\sigma^2}{\varepsilon^2}$ using a similar argument to Lemma 24. One can also see that the first term above corresponds to the first term of Eq. (25) in KL-MS, and one can use a similar technique therein to bound the first term above by $\frac{\sigma^2}{\varepsilon^2} \ln(\frac{T\varepsilon^2}{\sigma^2} \vee e^2)$ up to a constant factor.

Adding the bounds of $F3_1$ and $F3_2$ together, we conclude that $F3 \lesssim \frac{\sigma^2}{\varepsilon^2} \ln(\frac{T\varepsilon^2}{\sigma^2} \vee e^2)$.

## H    Additional Experiments

### H.1    Regret comparison

We compare KL-MS with the Bernoulli Thompson Sampling and MS [11]. Bernoulli Thompson Sampling chooses beta distribution as the prior (Beta(0.5, 0.5)) and the posterior. The reward environment is borrowed from [25], where there are two reward environments. Both are two-arm bandit, one has the mean reward $[0, 20, 0, 25]$ and the other has the mean reward $[0.80, 0.90]$. From Figure 1 and Figure 2 we find that the performance of KL-MS is better than MS by a margin, although worse than Bernoulli Thompson Sampling. Nevertheless, we will see in the next section that Bernoulli Thompson Sampling tends to generate somewhat unreliable logged data for offline evaluation.

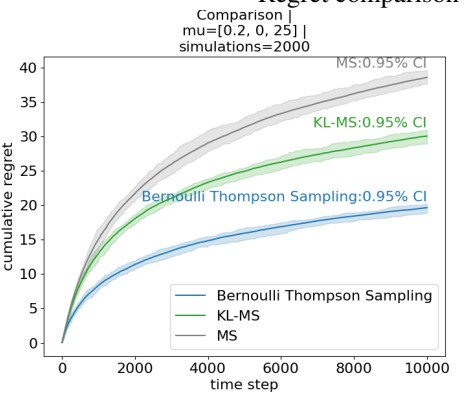

Figure 1: $\mu = [0.20, 0.25], T = 10,000$      Figure 2: $\mu = [0.80, 0.90], T = 10,000$

### H.2    Offline evaluation

This section presents our simulation results on offline evaluation using logged data. We use the logged data generated by our algorithm, KL-MS, and standard Thompson Sampling, to estimate the expected reward of the policy that takes an action uniformly at random in $[K]$, which is equal to $\bar{\mu} = \frac{1}{K}\sum_{i=1}^{K}\mu_i$. The logged data are of the form $(I_t, p_{t,I_t}, r_t)_{t=1}^{T}$, where $I_t$ is the action taken, $p_{t,I_t}$ is the action probability (which can be exact or approximate), $r_t$ is the received reward, all at time

step $t$. We consider the IPW estimator [22] that estimates $\mu$, defined as

$$\hat{\mu} = \frac{1}{T} \sum_{t=1}^{T} \frac{1/K}{p_{t,I_t}} r_t.$$

We set $T$, the time horizon of the interaction log, to be $1,000$ or $10,000$. For Thompson sampling, we use Monte Carlo (MC) to estimate the action probabilities; we vary the number of MC samples $M$ in $\{10^3, 10^4, 10^5\}$. Note that MC estimation of action probabilities induces a high time cost: in our simulations, for $T = 10^3$, KL-MS uses $0.43$s to generate its logged data; in contrast, BernoulliTS with $M = 10^3$ uses $15.21$s to generate its logged data. This suggest that setting $M = 10^4$ or $10^5$ may be impractical in applications.

Figures 3 to 14 shows the histogram of the IPW estimates of the average reward induced by logged data generated by KL-MS and Bernoulli-TS with MC estimation of action probabilities, based on $N = 2000$ independent trials in the same reward environment used in the previous experiment. Repeatedly, We have two 2-armed bandit problems, whose mean rewards are $[0.20, 0.25]$ and $[0.8, 0.9]$ respectively. Tables 2 to 9 report the MSE and the bias estimate of the respective estimator. It can be seen from the figures and tables that: (1) the logged data induced by KL-MS consistently give more accurate estimates of $\mu$, compared to that of BernoulliTS with MC estimation of action probabilities; (2) the offline evaluation performance of the logged data induced by BernoulliTS is sensitive to the number of MC samples $M$; while the performance of setting $M = 10^4$ or $10^5$ is on par with KL-MS, the estimation error of the more-practical $M = 10^3$ setting is evidently higher. (3) When time step $T$ is increasing, the error between the IPW estimator induced by BernoulliTS logged data and the true performance become larger while KL-MS remains the same level of error which is smaller than the BernoulliTS.

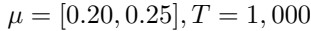

$\mu = [0.20, 0.25], T = 1,000$

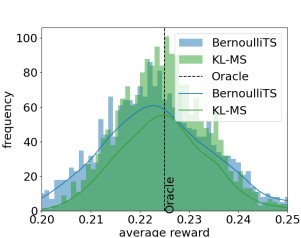
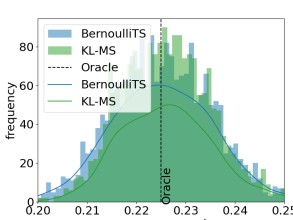
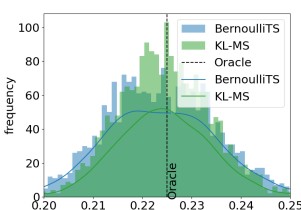

Figure 3: $M = 10^3$        Figure 4: $M = 10^4$        Figure 5: $M = 10^5$

$\mu = [0.80, 0.90], T = 1,000$

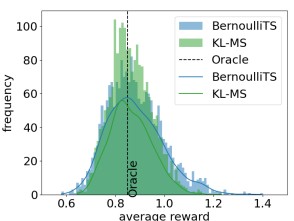
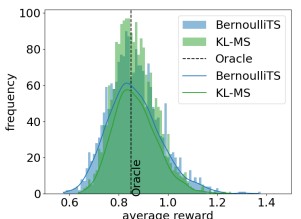
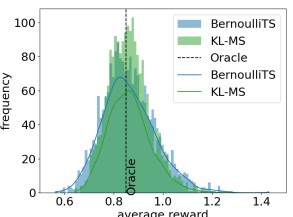

Figure 6: $M = 10^3$        Figure 7: $M = 10^4$        Figure 8: $M = 10^5$

$$\mu = [0.20, 0.25], T = 10,000$$

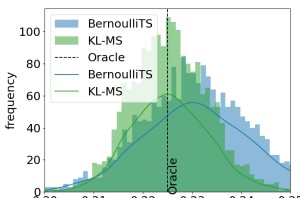

Figure 9: $M = 10^3$

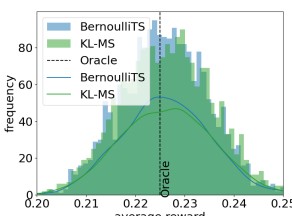

Figure 10: $M = 10^4$

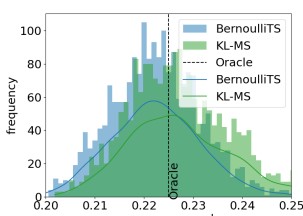

Figure 11: $M = 10^5$

$$\mu = [0.80, 0.90], T = 10,000$$

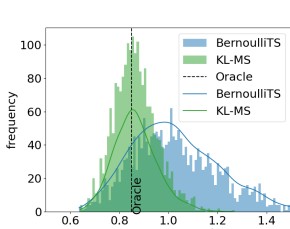

Figure 12: $M = 10^3$

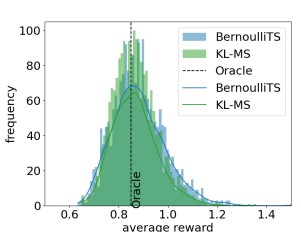

Figure 13: $M = 10^4$

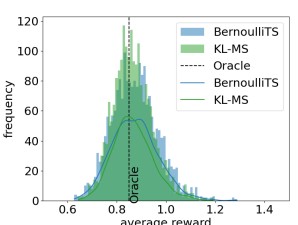

Figure 14: $M = 10^5$

Table 2: MSEs for $\mu = [0.20, 0.25]$, $T = 1,000$

|  | $M$ | | |
|---|---|---|---|
|  | $10^3$ | $10^4$ | $10^5$ |
| BernoulliTS | 0.00014 | 0.00012 | 0.00014 |
| KL-MS | 0.00001 | 0.00001 | 0.00001 |

Table 3: Bias for $\mu = [0.20, 0.25]$, $T = 1,000$

|  | $M$ | | |
|---|---|---|---|
|  | $10^3$ | $10^4$ | $10^5$ |
| BernoulliTS | -0.00059 | 0.00106 | -0.00068 |
| KL-MS | -0.00096 | 0.00118 | 0.00011 |

Table 4: MSEs for $\mu = [0.80, 0.90]$, $T = 1,000$

|  | $M$ | | |
|---|---|---|---|
|  | $10^3$ | $10^4$ | $10^5$ |
| BernoulliTS | 0.01464 | 0.01143 | 0.01228 |
| KL-MS | 0.00733 | 0.00782 | 0.00749 |

Table 5: Bias for $\mu = [0.80, 0.90]$, $T = 1,000$

|  | $M$ | | |
|---|---|---|---|
|  | $10^3$ | $10^4$ | $10^5$ |
| BernoulliTS | 0.02911 | 0.01741 | 0.01636 |
| KL-MS | 0.01304 | 0.01412 | 0.01355 |

Table 6: MSEs for $\mu = [0.20, 0.25]$, $T = 10,000$

|  | $M$ | | |
|---|---|---|---|
|  | $10^3$ | $10^4$ | $10^5$ |
| BernoulliTS | 0.00017 | 0.00010 | 0.00009 |
| KL-MS | 0.00007 | 0.00006 | 0.00011 |

Table 7: Bias for $\mu = [0.20, 0.25]$, $T = 10,000$

|  | $M$ | | |
|---|---|---|---|
|  | $10^3$ | $10^4$ | $10^5$ |
| BernoulliTS | 0.00637 | 0.00142 | -0.00240 |
| KL-MS | 0.00052 | 0.00066 | 0.00220 |

Table 8: MSEs for $\mu = [0.80, 0.90]$, $T = 10,000$

|  | $M$ | | |
|---|---|---|---|
|  | $10^3$ | $10^4$ | $10^5$ |
| BernoulliTS | 0.06842 | 0.01276 | 0.01220 |
| KL-MS | 0.00898 | 0.00804 | 0.00929 |

Table 9: Bias for $\mu = [0.80, 0.90]$, $T = 10,000$

|  | $M$ | | |
|---|---|---|---|
|  | $10^3$ | $10^4$ | $10^5$ |
| BernoulliTS | 0.17947 | 0.03401 | 0.04313 |
| KL-MS | 0.02046 | 0.01731 | 0.01123 |