# OpenReview forum: "Kullback-Leibler Maillard Sampling for Multi-armed Bandits with Bounded Rewards"
_NeurIPS.cc/2023/Conference — NeurIPS 2023 poster_

### Official Review · Reviewer_nCy6 · 2023-06-25

**Soundness:** 4 excellent
**Presentation:** 3 good
**Contribution:** 2 fair
**Rating:** 6
**Confidence:** 4

**Summary:**

In this paper, the author analyzes the MED algorithm proposed by Honda \& Takemura (2011) for Bernoulli distributions in the context of general bounded distributions, and under the name KL-Maillard Sampling. This work is a follow-up of a previous work that proposed Maillard Sampling for sub-gaussian distributions.

KL-MS is a bandit algorithm that samples an arm at each time step with probability $p(t) \propto \exp(-N_a(t) \text{kl}(\mu_{t-1,a}, \mu_{t-1, \text{max}}))$, where kl is the KL divergence corresponding to Bernoulli distributions. The interest of this simple strategy is that one can explicitly compute the probability to pull each arm, which is useful for instance in the context of off-policy evaluation.

Contrarily to the initial work of Honda &  Takemura that focused on instance-dependent bounds, and in the spirit of the previous MS paper, the authors provide both optimal instance-dependent and minimax bounds for the algorithm for Bernoulli distributions. The same guarantees naturally hold for general bounded distributions, losing the optimality for problem-dependent guarantees compared to the original MED algorithm using the ``tight'' divergence. It is also proved that the worst-case guarantees scale in the standard deviation of the best arm, which is on par with what is known from the sub-gaussian case.

**Strengths:**

* The paper is well-written, clear, and easy to follow. Furthermore, it seems technically sound, and the proofs are carefully detailed. The literature review is well-covered regarding the scope of the paper.

*  The analysis of asymptotic optimality of MED for the Bernoulli case is largely simplified compared to the original proof of Honda \& Takemura (2011). Furthermore, the worst case optimality is a novel result compared to this work. Compared to the previous MS paper, it is also interesting that the optimal minimax ratio is achieved without tweaking the algorithm. To prove these results, the the authors perfectly used the analysis tricks introduced recently in the TS literature (e.g all the cited works from Jin et al.).

* The main novel element/insight of the paper compared to its two major inspirations is the refined analysis of the "under-exploration" term (F3) in the regret analysis, that lead to the $\log(K)$ vs. $\log(T)$ improvement of the minimax ratio of MS without having to change the algorithm, making MS+ obsolete. The changes in the proof for this are rather substantial so the contribution is valuable.

* Maybe the most surprising result in the paper is the minimax bound scaling with the variance term $\mu_1(1-\mu_1)$. Therefore I would have appreciated some explanation in the main text as to where Theorem 4 comes from. If I understand correctly, it follows from a tighter version of Pinsker's inequality (Lemma 28) which is worth highlighting. While interesting, this trick could certainly be applied to the analysis of other bandit algorithms (as done for KL-UCB in an Appendix), so the result cannot really be interpreted as an indicator of the superior performance of KL-MS.


**Weaknesses:**

* I am a bit uncomfortable with the re-branding of MED as KL-MS. This was understandable for the initial MS paper since Honda & Takemura tackled bounded distributions, but here the algorithm exactly matches MED for Bernoulli distributions. Furthermore, it is folklore that for algorithms using divergences the Bernoulli divergence can be used for general bounded distributions. Hence, there is no real reason for this re-branding in my opinion. However, I insist on the fact that I find the novel elements of analysis intereting.

* Regarding the analysis, it seems that it differs from the original paper only from term (F3). The way the authors handle this term is very interesting and a valuable contribution in itself, but this should be better highlighted in the paper. However, even this part seems largely inspired by recent papers from Jian et al. for the analysis of Thompson Sampling, so I wonder if there is a really novel theoretical contribution in the paper.

* Minor: the authors should cite a recent follow-up of Honda and co-authors on the MED algorithm: https://arxiv.org/abs/2303.06058
This do not alter the contribution of this paper since the authors focus on problem-dependent guarantees of MED for a broader class of distributions, but answer some of the questions presented in the conclusion of the paper.

**Questions:**

I don't have specific questions for this work, everything seems clear to me.

---

> ### Author Rebuttal · Authors · 2023-08-10
>
> We thank the reviewers for taking the time to review our work and provide valuable feedback thoroughly.
>
> *(1) The shared common mechanism with MED and KL-MS.*
>
> We agree with the reviewer that under Bernoulli environments, our algorithm and MED are identical. The main reason we see our algorithm as a 'Maillard sampling'-style algorithm is that its action probabilities are calculated only based on empirical mean rewards of all arms. In contrast, MED algorithms and their variants compute action probabilities based on a notion of 'empirical divergence' that relies on the full empirical distribution of rewards.
>
> *(2) The novelty of our paper should be appropriately highlighted.*
>
> We include the most challenging part in Appendix D.3.1, "Roadmap of analysis of F3," and highlight some critical techniques that can be of independent interest. For example, extending MS to the KL version is not trivial and requires much work. Also, choosing a separating point in F3 takes a lot of work to find. Please also see our global response for a recap of our analysis and technical highlights.
>
>
> *(3) Reference [1]*
>
> Thank you for the reference. Indeed, [1] gives randomized and off-policy-amenable algorithms that achieve asymptotic optimality for unbounded rewards, which answers our question in lines 298-300. However, it does not (yet) yield easily interpretable finite-time regret bounds, which we believe is an important research direction.
>
> [1] D. Baudry, K. Suzuki, and J. Honda. A general recipe for the analysis of randomized multi-armed
> bandit algorithms, 2023.

---

> > ### Comment · Reviewer_nCy6 · 2023-08-11
> > **post-rebuttal comment**
> >
> > Thank you for your response. Along with the other reviews, it confirms my positive evaluation of the paper.
> >
> > (1) I see, so for you the terminology "Maillard Sampling" may refer to algorithms that are easier-to-compute (but sometimes sub-optimal) proxies of MED in some sense?
> >
> > (2) I see. I still believe hat most of the difficulty in handling this term has been addressed in other works (of Jin et al.), but due to the technicality of the arguments this is not a limitation of the paper and I agree that the proof must have required some work.
> >
> > (3) I agree with your comment. This is related to your discussion with reviewer Gstc, but for the bounded case I believe that there is indeed some work to adapt your analysis to MED, because for instance the existing concentration inequality on KL-inf scales as (n x exp(-n ..)) (while for Bernoulli kl we have $\exp(-n...)$), and this multiplicative n itself would worsens the worst-case bound. Obtaining tighter concentration may be challenging, and it is not even clear that this is possible. Hence, the MS framework has an interest in the sense that it makes worst-case analysis easier with the tools developed by JIn et al. for exponential families, since it only requires to concentrate empirical means.

---

> > > ### Author Response · Authors · 2023-08-20
> > >
> > > (1) Yes, that was our intention. We also see your point that, if we go with a generous interpretation of MED in the sense of [Baudry, Suzuki, and Honda, 2023, Eq. (4)], By choosing $D_\pi(F_k(t), \mu^\star(t))$ to be $\mathsf{kl}(\mu_k(t), \mu^\star)$, MED specializes to our KL-MS.
> > >
> > > (2, 3)  We wish to point out that only a small part of our proof is inspired by [Jin et al. 2022] (specifically, our application of Lemma 25 in bounding $F3_1$ is inspired by their usage of Lemma A.4 to prove Lemma A.1, which is for refining the minimax ratio from $\sqrt{ \dot\mu_1 \ln T }$ to $\sqrt{\dot \mu_1 \ln K}$; even that time-uniform concentration inequality of empirical rewards was originally due to [Menard and Garivier, 2017], to the best of our knowledge). For the high-level case splits on bounding $\mathbb{E}[N_{T, a}]$, [Jin et al. 2022] uses a standard split in frequentist analysis of Thompson Sampling [e.g. Agrawal and Goyal, 2017, Eq. (2)], depending on whether the posterior sample of arm i exceeds $\mu_1 - \varepsilon$; In contrast, our split of F1, F2, F3 is similar to (and perhaps simplifies) the analysis of MS [Maillard, 2013, Bian and Jun, 2022] and MED [Honda and Takemura, 2010].
> > >
> > > (3) We acknowledge the reviewer’s finding and appreciate the explanation.

---

### Official Review · Reviewer_QYoJ · 2023-06-29

**Soundness:** 3 good
**Presentation:** 3 good
**Contribution:** 2 fair
**Rating:** 5
**Confidence:** 4

**Summary:**

This paper considers a classic bandit problem, where the algorithm should explicitly output the random distribution of the next pulling arm (as a comparason, in classic case, the algortihm only needs to generate one arm from this random distribution and outputs that arm). Existing results only work on the case that the random rewards are unbounded and subgaussian. In this paper, the authors extend the existing works to the case that the rewards are bounded in $[0,1]$, design a KL-MS algorithm (using a KL-divergence approach). They show that the regret upper bound of KL-MS is near optimal. They also use some experiments to show that the performance of KL-MS (in outputing the random distribution precisely) outperforms existing baselines.

**Strengths:**

The regret bound in this paper is nearly tight.
The writting is clear for me to understand.

**Weaknesses:**

My first concern is about the model setting, i.e., why we require to know the exact random distribution of pulling the arm? Though there is an example of estimating the average reward, I do not think this is a well-motivated one. Can you provide more examples about the why we require that distribution in reality? Besides, why not just use a UCB-type algorithm, which can easily give you the exact probability distribution, as long as tight analysis? I know that there are works, e.g., MOSS, to achieve the tight $O(\sqrt{KT})$ regret upper bound, but not very sure whether there are UCB algorithms that achieve $O(\sqrt{\mu(1-\mu)KT})$ regret upper bound (though I think after using some variance-based concentration, the steps are straightforward).

My second question is about the experiments of this paper (in appendix H). I do not see any comparason about the regrets between different algorithms, and I am wondering the regret performances of KL-MS.

Finally, do you think the idea (of either MS or KL-MS) could be applied to infinite-arm case (e.g., linear bandits), for example, return a distribution supported on an infinite set?

**Questions:**

Please see the above "Weaknesses"

---

> ### Author Rebuttal · Authors · 2023-08-10
>
> We thank the reviewers for taking the time to review our work and provide valuable feedback thoroughly.
>
> *(1) Why do we need the exact action distribution in reality?*
>
> Our motivation comes from the broad field of off-policy evaluation and optimization for contextual bandits and reinforcement learning, where learners use previously-collected logged data to make inferences about the unknown environment [1]. For example, in an online advertisements recommendation system, before deploying a new policy, the platform would like to evaluate its performance using historical data (collected by previous policies), possibly for safety considerations. As we demonstrated in the experiments, Thompson Sampling, albeit having good regret performance, when combined with Monte Carlo estimation of action probabilities, yields logged data that produces reward estimates that are less reliable than that of KL-MS (which instead maintains closed-form action probabilities).
>
> *(2) Regret comparison between algorithms.*
>
> Since we focus on the performance of offline policy evaluation, we put the most essential plots in Appendix H. We now include two experimental evaluations on the comparison between KL-MS, MS and Thompson Sampling in the global response for your reference.
>
> *(3) Generalizing KL-MS to an infinite-arm case.*
>
> We agree that this is an interesting topic and leave it for future work.
>
> [1] Saito, Y., Udagawa, T., Kiyohara, H., Mogi, K., Narita, Y., & Tateno, K. (2021, September). Evaluating the robustness of off-policy evaluation. In Proceedings of the 15th ACM Conference on Recommender Systems (pp. 114-123).

---

> > ### Comment · Reviewer_QYoJ · 2023-08-14
> > **Thank you**
> >
> > Thanks for your reply.
> >
> > For (1), I am still wondering whether we can use UCB method to achieve the same goal. Can you give me some insights about why UCB-based policies do not work in your example?

---

> > > ### Author Response · Authors · 2023-08-20
> > >
> > > Note that deterministic exploration algorithms such as UCB generates logged data that cannot be reliably combined with the IPW estimator for offline evaluation. More precisely, logged data is generated by a UCB-based policy with $p_{t,I_t}=1$ for all t. Considering the offline evaluation setup in Appendix I, the IPW estimator will be $\hat\mu := \sum_{t=1}^T \frac{r_t}{K T}$. However, such an IPW estimator is biased with respect to the estimation target $\mu = \frac{1}{K} \sum_{i=1}^K \mu_i$. Consider a UCB-type algorithm, the fraction of the optimal arm in the historical arm pull $N_{T,1}/T $ will go $1$, therefore when we let $T \rightarrow \infty$, $\hat\mu = \sum_{t=1}^T \frac{r_t}{K T} \rightarrow \sum_{t=1}^T \frac{\mu_1}{K T}=\frac{r_1}{K}$ which is not equal to $\mu$.

---

### Official Review · Reviewer_Mj4M · 2023-07-04

**Soundness:** 3 good
**Presentation:** 2 fair
**Contribution:** 3 good
**Rating:** 5
**Confidence:** 4

**Summary:**

The submission considers the vanilla setting of stochastic K-armed bandits and studies a strategy introduced by Maillard (2013), which relies on exponential weights and outputs at each round probabilities of taking each action. This is often convenient in offline policy evaluation, when estimates based on inverse propensity weighting [IPW] are constructed. Distribution-dependent and distribution-free regret bounds are provided, either in a general non-parametric model of all probability distributions over [0,1], or in the much specific model of Bernoulli distributions. The general distribution-dependent regret bounds asymptotically match the gap-based bounds of UCB (Theorem 1 and Remark 2) and is actually optimal in the Bernoulli model (Theorem 5). The general distribution-free bound improves on the one for UCB by featuring a \sqrt{\mu^\star (1-\mu^\star)} term (Theorem 3). Another main result is formed by Figure 1 and Table 1: there are actually few randomized strategies (Thompson sampling, MED) and none of them exhibits closed-form expressions for probabilities of plays.

This shows that the core result of this article is: a strategy for the vanilla case of stochastic K-armed bandits, with decent (though not optimal) distribution-dependent and distribution-free regret bounds, and based on determining actual probabilities of playing arms, which is useful for offline policy evaluation.


**Strengths:**

The idea of constraining the strategy to output probability distributions while getting decent bounds is nice and may turn useful---I have witnessed several recent articles critically using IPW in bandit contexts. By 'decent bounds', I mean bounds that are as good as, or slightly better than, UCB, but not optimal (as IMED and recent versions of KL-UCB achieve).

The exposition is clear and I enjoyed reading the main body of the submission.


**Weaknesses:**

1.
Exponential weights are actually difficult to compute in practice with a good accuracy, at least for suboptimal arms that are played often. Perhaps this case does not arise (suboptimal arms are played only logarithmically many times and the probabilities are easy to compute with a good accuracy), but the accuracy in the computation should be commented, especially given the critiques against Thompson sampling on these issues on page 2.

2.
The comparison to previous works could be clarified and reorganized in pages 4--5. In particular, it would have been better to recall first the typical (e.g., for UCB) as well as the optimal distribution-dependent and distribution-free regret bounds, in the Bernoulli model and in the model P(0,1) of all distributions over [0,1], when no constraint of outputting probability distributions is imposed. The sub-UCB criterion could be omitted, I don't think it adds anything. The literature review seems a bit outdated. In particular, a new reference is critically missing: https://www.jmlr.org/papers/volume23/20-717/20-717.pdf / it shows that in the model P(0,1) there exists a strategy called KL-UCB-switch that achieves simultaneously the optimal distribution-dependent and the optimal distribution-free bounds.

3.
The regret bounds are unprecise: (i) they involve O(...) terms and (ii) even the main terms are difficult to read because of +/- c \Delta_a terms in the kl, (iii) not mentioning the additive T \Delta term, where \Delta is a parameter that must then be << \ln T / T and therefore should vanish. Even worse, Lemma 9 and Theorem 5 are proved by taking \Delta = 0 in the bound of Theorem 1, while Theorem 1 assumes \Delta > 0.

4.
The proof sketches are too vague: pages 8-9 merely indicate a proof structure in terms of decompositions of events and other immediate considerations, but the actual boundings of the probabilities of interest is not explained in the main body (but is detailed in the extremely long appendix). At least two or three salient (new?) ingredients of these proofs should have been given in the main body. What I read on pages 8-9 is too high level and actually takes almost 2 pages without learning anything specific to the reader. I regretfully couldn't check the proofs and get a sense of their correctness, but I don't feel guilty for this, as nothing or almost helped me doing this in the main body. Better editorial choices could have been made as far as proof sketches are concerned.


Other comments / remarks / typos along the text:
- Lines 9-10: I wouldn't insist on the distribution-dependent optimality for Bernoulli distributions (which is a minor point) but rather on getting decent (better than UCB) bounds
- Line 28: also Garivier and Cappé 2011
- Footnote 1: yes, but for known L and U
- Caption of Figure 1: difficult to understand in itself, I have to read lines 770--773 in appendix to understand (these explanations should thus be moved in the main body)
- Line 48: the empirical averages \mu_{t,a} were not formally defined
- Lines 54-55: sounds like an overstatement; the submission proposes decent but not optimal distribution-dependent and distribution-free regret bounds
- Table 1: Tsallis-INF is among the few strategies that output distributions, so it should not have been excluded on the ground of a minor issue given its non-optimal bound for Bernoulli distribution; this Table 1 is generally not helpful given that it contains too many strategies that do not output distributions
- Lines 65-67: I would be less enthusiastic; getting rid of the \sqrt{\ln K} is uneasy but was achieved for some algorithms that are optimal from the distribution-dependent viewpoint (see https://www.jmlr.org/papers/volume23/20-717/20-717.pdf) though I appreciate as well the \sqrt{\mu^*(1-\mu^*)} term, which indeed may be smaller than \sqrt{\ln K} in some situations
- Line 69: drop 'with'
- Line 82: OK for this definition in case of absolute continuity, otherwise, = +\infty
- Lines 108-109: syntax issue
- Line 119: > b, not \geq b; this quantity is called K_inf
- Lines 154-155: some strategies like KL-UCB-switch (see https://www.jmlr.org/papers/volume23/20-717/20-717.pdf) would be sub-UCB and enjoy a \sqrt{KT} distribution-free regret bound---I thus disagree with the statement made here
- Lines 205-206: same kind of comments
- Lines 239-241: there is no such issue if the Bernoullization trick of lines 115-119 is implemented
- Line 258: rather (6) instead of (8)


**Questions:**

I would like to read authors' opinion on the four main weaknesses that I raised.

**Limitations:**

(They are well-addressed in the conclusion, Section 6, and include extensions of the optimality results of Maillard sampling to general exponential families and even to the non-parametric setting of all distributions over [0,1].)

---

> ### Author Rebuttal · Authors · 2023-08-10
>
> We thank the reviewers for taking the time to review our work and provide valuable feedback thoroughly. We genuinely appreciate reviewers carefully examining our results and offering insightful comments to improve the quality of our research. We have carefully considered each of the weaknesses raised by the reviewer and committed to making the necessary revisions to solidify our arguments.
>
> Allow us to address each of the specific points the reviewer raised.
>
> *(1) The accuracy of computation of action probabilities of KL-MS and comparison against Thompson sampling.*
>
> If we understand your concern, 'Exponential weights are actually difficult to compute in practice with a good accuracy' correctly, the imprecision of the exponential weight calculation comes from floating-point number underflow when evaluating the exponential function. This numerical imprecision is insignificant because it only affects the action probabilities by a minuscule additive factor of the smallest float value in the computer.
>
> Also, if we consider the inconsistency brought by using the log data generated by KL-MS and TS, using TS inevitably introduces bias since the agent cannot access the actual generation probability, while in the KL-MS, it is not.
>
> Please let us know if we have missed anything (we would appreciate it if you could elaborate on your concerns by, e.g., giving some references).
>
> *(2) The sub-UCB criterion could be omitted because it does not add anything.*
>
> Our paper focuses on a finite-time (i.e. nonasymptotic) analysis, and the sub-UCB criterion _does_ play an essential role in this view. More precisely, any algorithm satisfying the sub-UCB criterion guarantees not to suffer a higher regret order than UCB-like algorithms at any time.
>
> A typical example that illustrates the relevance of the sub-UCB property is in [1]’s section "Failure of MOSS", where it gives a bandit instance of K arms such that any sub-UCB algorithm has a regret at most O(K \ln K), while in contrast, MOSS, a non-sub-UCB algorithm, has a regret of at least Omega(K^2), which is significantly larger when K is large. As another example, lines 148-150 (see also footnote 4) show that MED's best-known regret guarantee does not imply that MED is sub-UCB. Finally, recall from Table 1 that not all algorithms are sub-UCB.
>
> *(3) The comparison to previous works could be clarified and reorganized in pages 4--5.*
>
> The current organization of the previous works is based on different criteria an algorithm satisfies. Thank you for your suggestion, we will add a discussion on the earlier UCB algorithm. Thank you for your reference on KL-UCB Switch. We will also add relevant discussions in the final version.
>
> *(4) Imprecise regret bound in Theorem 1.*
>
> We appreciate the reviewer's careful reading.
>
> * We had a typo in theorem 1 (and Lemma 8): '$\Delta > 0$' should instead be '$\Delta \geq 0$' (note that the proof of Theorem 1 continues to hold when $\Delta = 0$, specifically the equation display in line 439).
>
> * The + / - $\Delta_a$ terms in the main term is standard in the analysis of Bernoulli bandits, similar to the $\varepsilon_1$, $\varepsilon_2$ factors in standard KL-UCB analysis [Lattimore and Szepesvari, Bandit Algorithms, Theorem 10.6].
>
> * Note that in the downstream applications of Theorem 1, we do not always choose $\Delta \ll \ln T / T$ (although $\Delta \ll \ln T / T$ makes sense for showing asymptotic $\ln T$-style regret bounds): for example, in the proof of worst-case regret bound Theorem 3, we chose $\Delta = \sqrt{\dot\mu_1 K \ln K / T}$.
>
> * Throughout this paper, big O only hides absolute constants. The main reason we present KL-MS's regret bound in this form is its flexibility in deriving asymptotic and nonasymptotic regret guarantees: to derive asymptotic guarantees, we choose $\Delta = 0$ and treat the terms inside the big-O as lower-order terms; to derive nonasymptotic properties such as sub-UCB or worst-case regret guarantees, we are generous in giving up constant factors and apply Lemma 8. We will correct the typos and add these clarifications in the final version.
>
> *(5) the proof sketches are too vague.*
>
> We refer the reviewer to the global response for a recap of our proof sketches and technical highlights.
>
> *(6) Minor comments.*
>
> Thank you for these; we will make a pass over our paper to incorporate them.
>
> * Reorganizing Table 1: if we include Tsallis-INF, it also seems to make sense to include other randomized exploration algorithms, such as: EXP3, EXP3-IX, and the Boltzmann-Gumbel exploration algorithm [5] (and many others), which, although can maintain closed-form action distributions, do not have sharp regret guarantees in the stochastic setting, such as asymptotic optimality in Bernoulli case and sub-UCB. Nevertheless, we are open to any suggestions that can help elucidate comparisons between our algorithm and previous works.
>
>
> [1] T. Lattimore. Refining the confidence level for optimistic bandit strategies. Journal of Machine Learning Research, 19(20):1–32, 2018.
> URL http://jmlr.org/papers/v19/17-513.html
>
> [2] Agrawal, S., & Goyal, N. (2017). Near-optimal regret bounds for thompson sampling. Journal of the ACM (JACM), 64(5), 1-24.
>
> [3] Garivier, A., Hadiji, H., Menard, P., & Stoltz, G. (2022). KL-UCB-switch: optimal regret bounds for stochastic bandits from both a distribution-dependent and a distribution-free viewpoints. The Journal of Machine Learning Research, 23(1), 8049-8114.
>
> [4] Cesa-Bianchi, N., Gentile, C., Lugosi, G., & Neu, G. (2017). Boltzmann exploration done right. Advances in neural information processing systems, 30.

---

> > ### Comment · Reviewer_Mj4M · 2023-08-11
> >
> > I acknowledge reading the entire thread of reviews and corresponding rebuttals.
> >
> > On this specific rebuttal, I'm satisfied with answers 1-2-3.
> > For answer 4, I believe that the KL-UCB-Switch paper is a good example of a paper with precise bounds not relying on O(...) terms, but perhaps this is too high a standard.
> > I still believe that better proof sketches could have been provided, beyond the mere descriptions of the proof structures.
> >
> > All in all I am ready to increase my score to 5 and will update my report accordingly.

---

> > > ### Author Response · Authors · 2023-08-20
> > >
> > > We can give an exact bound of KL-MS’s regret by replacing the Big-O term in Eq. (3) with exact constants. Specifically, the exact form of Eq. (3) is
> > > $$\mathrm{Reg}(T)
> > > \leq   T\Delta
> > >     +  \sum_{a: \Delta_a > \Delta} \frac{\Delta_a \ln(T \mathsf{kl}(\mu_a + c \Delta_a, \mu_1 - c \Delta_a) \vee e^2 )}
> > >                                         {\mathsf{kl}(\mu_a + c \Delta_a, \mu_1 - c \Delta_a)}
> > > +\left( \frac{34}{c^2} +  \frac{8}{(1-2c)^2} \right)
> > >         \cdot  \sum_{a: \Delta_a > \Delta}
> > >                 \left( \frac{\dot\mu_1 + \Delta_a}{\Delta_a} \right) \ln
> > >                     \left(
> > >                         \left(  \frac{\dot\mu_1 + \Delta_a}{\Delta_a^2} \wedge \frac{T\Delta_a^2}{\dot\mu_1 + \Delta_a}
> > >                         \right)
> > >                         \vee e^2
> > >                     \right)
> > > $$
> > > . As an example, if we choose $c=\dfrac{1}{4}$, the final regret bound given by Eq. (3) would be
> > > $$\mathrm{Reg}(T)
> > > \leq   T\Delta
> > >     + \sum_{a: \Delta_a > \Delta} \frac{\Delta_a \ln(T \mathsf{kl}(\mu_a + c \Delta_a, \mu_1 - c \Delta_a) \vee e^2 )}
> > >                                         {\mathsf{kl}(\mu_a + c \Delta_a, \mu_1 - c \Delta_a)}
> > > +576 \cdot  \sum_{a: \Delta_a > \Delta}
> > >                 \left( \frac{\dot\mu_1 + \Delta_a}{\Delta_a} \right) \ln
> > >                     \left(
> > >                         \left(  \frac{\dot\mu_1 + \Delta_a}{\Delta_a^2} \wedge \frac{T\Delta_a^2}{\dot\mu_1 + \Delta_a}
> > >                         \right)
> > >                         \vee e^2
> > >                     \right)
> > > $$.
> > >
> > >
> > > To see this, we first note that Lemma 10 is exact in that it does not hide constant factors. By tracking the exact constants in the proof of Theorem 1 (lines 435-438), we have that
> > > $$
> > > \mathbb{E}\left[ N_{T,a} \right]
> > >     \leq
> > >     \frac{\ln(T \mathsf{kl}(\mu_a + c \Delta_a, \mu_1 - c \Delta_a) \vee e^2 )}{\mathsf{kl}(\mu_a + c \Delta_a, \mu_1 - c \Delta_a)}
> > >     +
> > >     \left( \frac{34}{c^2}+\frac{8}{(1-2c)^2} \right)
> > >     \cdot
> > >     \left( \frac{\dot\mu_1 + \Delta_a}{c^2 \Delta_a^2} \right)
> > >     \ln\left(
> > >         \left(
> > >             \frac{\dot\mu_1 + \Delta_a}{c^2 \Delta_a^2} \wedge \frac{c^2 T  \Delta_a^2}{\dot\mu_1 + \Delta_a}
> > >         \right) \vee e^2
> > >     \right).
> > > $$

---

> > > > ### Comment · Reviewer_Mj4M · 2023-08-21
> > > >
> > > > Stating more explicitly such exact bounds in the paper looks like a good idea to me, e.g., by keeping the O(...) form in the theorem and providing a pointer to later equations providing exact bounds.

---

### Official Review · Reviewer_GStc · 2023-07-10

**Soundness:** 3 good
**Presentation:** 3 good
**Contribution:** 3 good
**Rating:** 6
**Confidence:** 3

**Summary:**

The paper studies the classical regret-minimization problem in the stochastic multi-armed bandit framework. In particular, the manuscript's focus is on randomized algorithms with an aim to develop one with closed-form arm-selection probabilities at each step. Data collected by such algorithms can be used for offline policy evaluation. The manuscript proposes an algorithm called KL-MS for bounded-support distributions, that achieves KL-style regret guarantees. It is asymptotically optimal (in the instance-dependent stochastic sense) for bernoulli bandits, and order optimal for more general bounded-support distributions. It also enjoys an optimal worst-case regret guarantee. The paper also presents numerical study comparing the offline evaluation when the data is collected using the proposed KL-MS and Thompson Sampling with Monte-carlo on bernoulli bandits.





**Strengths:**

The paper is written well and easy-to-read. I particularly enjoyed the various remarks and discussions after the results, providing insights into the results and comparing the analysis to existing ones. While in some settings randomness should also be treated as a resource (and hence be used with care), there are indeed benefits to using randomized algorithms in other settings. For example, as highlighted in the paper, the data collected by randomized algorithms can be used for offline evaluation. The paper develops an algorithm that simultaneously satisfies different desirable properties, while being optimal (or close-to-optimal) in a non-parametric setting of bounded-support distributions.



**Weaknesses:**

1. The plots in the appendix are not very clear. The text along the vertical lines is overlapping and unreadable. It would be good to spread-out the fugures and probably figures in a vector form for clearer display.

2. In view of the recent results on fragility of optimized bandit algorithm, I believe that the MAB results should be studied and stated beyond the expected regret. See for instance "Fan, Lin, and Peter W. Glynn. "The fragility of optimized bandit algorithms." arXiv preprint arXiv:2109.13595 (2021)."


**Questions:**

1. In line 21, the Reg(T) is referred to as pseudo-regret. How is this different from the usual expected regret? Why "pseudo" in the regret?

2. How would the analysis change (or challenges in extending) if instead of Bernoulli-kl in the exponent, one uses KL (the lower bound opt.)? Could the current results and analysis be already extended to that algorithm? I believe that is then the MED algorithm? It would then suggest a natural way to extend the algorithm and analysis for exponential families and more general distributions, like the heavy-tailed ones considered in Agrawal, Juneja, Koolen, 2021. A discussion along these lines would be interesting to see.

3. How does MS compare with KL-MS numerically?

4. How is the performance of the algorithm affected if the bounds [0,1] are not exactly known. For example, if the samples are from some misspecified setting, i.e., distribution with support in  [-0.5, 0.5] but the algorithms is guaranteed only [0,1]-supported distributions?

---

> ### Author Rebuttal · Authors · 2023-08-10
>
> We thank the reviewer for all responses and for taking the time to review our work and provide valuable feedback thoroughly.
>
> *(1) The plots in the appendix could be clearer.*
>
> We appreciate the reviewer pointing out the problem. We will make necessary modifications to the plots to present them more clearly. E.g., for the plots in the appendix H, we will remove the (overlapping) vertical text and retain the legend to clarify the plot.
>
>
> *(2) What is the definition of pseudo-regret?*
>
> For the definition of pseudo-regret, we follow the definition of [1, Eq. (1.4)]. This is the same as the 'expected regret' notion in the Fan & Glynn reference you gave (and probably what you meant by 'expected regret').
>
> Note that another notion of 'expected regret' is in the literature [1, Eq. (1.2)]. We will add a remark in the final version, clarifying this terminology overload issue in the literature.
>
> *(3) The MAB results should be studied and stated beyond the ‘expected regret’ (in the sense of Fan and Glynn, 2021).*
>
> Thanks for the reference. We agree that studying the tail property of the pseudo-regret (in the sense of Fan and Glynn, 2021) of MS is interesting and leave it for future work.
>
> *(4) How would the analysis change (or challenges in extending) if instead of Bernoulli-KL in the exponent, one uses KL (the lower bound opt.)?*
>
> By the KL (the lower bound opt.), we think that you meant KL( empirical distribution of arm a, maximum empirical reward) in the sense of the KL defined in our line 119. If so, indeed this would be the MED algorithm. We don’t yet know how to adapt our analysis to provide a new analysis of MED; although we agree that this is an interesting question for further investigation.
>
> *(5) How does MS compare with KL-MS numerically?*
>
> We added two plots in the global response to show an comparison between MS and KL-MS in the [0,1]-bounded reward stochastic bandits environments. KL-MS has lower regrets overall, due to its exploitation of the [0,1]-bounded reward structure.
>
> *(6) Does KL-MS work in a misspecified model setting, say the support of the reward set is [-0.5,0.5]?*
>
> In this case, KL-MS may not work correctly: according to the KL-MS algorithm (Eq. (2)), KL is undefined if $\hat{\mu}_{t-1,a}$ is negative.
>
> [1] S. Bubeck and N. Cesa-Bianchi. Regret analysis of stochastic and nonstochastic multi-armed bandit problems, 2012

---

> > ### Comment · Reviewer_GStc · 2023-08-11
> > **Response to the rebuttal**
> >
> > Thank you for your response. I acknowledge reading the entire thread of reviews and corresponding rebuttals. For now, I don't have further questions.

---

### Author Rebuttal · Authors · 2023-08-10

We thank all reviewers for taking the time to review our work and provide valuable feedback thoroughly. Here we address two common points shared by reviewers.

**The comparison between algorithms in terms of regret.**

We chose the reward setting following the experimental setup of Thompson sampling literature [1], where we consider two 2-arm bandit environments with expected rewards being [0.2, 0.25] and [0.80, 0.90] respectively, with 2000 time simulations to estimate the regret of KL-MS, MS and Thompson Sampling. Our result (see attached PDF file) shows that in terms of performance: (1) KL-MS is better than MS by exploiting the variance information of all arms; (2) KL-MS performs worse than Bernoulli Thompson Sampling; we suspect that this is due to Thompson sampling exploiting more aggressively in such relatively-easy environments.

**The proof idea and the novelty in the analysis (R2, R4).**

Since the focus of our paper is on establishing both asymptotic and finite-time regret guarantees for the KL-MS algorithm (specifically, asymptotic optimality in Bernoulli setting, sub-UCB property, $\sqrt{ \dot\mu_1 \ln K }$ minimax ratio), we need to give a finite-sample bound on $\mathbf{E}[N_{T, a}]$, the expected number of pulls to suboptimal arm a. To this end, we divide it into four parts (divisions similar in spirit is standard in the analysis of bandit algorithms, e.g. [1, Eq. (2)]), and bound each part respectively. The four parts are:

* u, a burn-in term,
* F1, which corresponds to the "steady state" when the empirical means of arm a and the optimal arm are both estimated accurately;
* F2, which corresponds to the case when the empirical mean of arm a is abnormally high;
* F3, which corresponds to the case when the empirical mean of the optimal arm is abnormally low.

As we mention in lines 272-275, bounding F1 and F2 are relatively straightforward, similar to the MS analysis [3]. Our main technical challenge lies in the analysis of F3. For that term, we had a detailed 'roadmap of analysis' in Appendix D.3.1 that explains our intuition and the main techniques used.
Of these, we wish to highlight two techniques from our paper that can be of independent interest:

* a careful double-integral argument that simplifies prior works that bounds the expectation of a certain function of the empirical reward using a tail probability bound on the empirical reward, which already establishes Bernoulli asymptotic optimality, sub-UCB property, and $\sqrt{\dot\mu_1 \ln T}$ minimax ratio (lines 481-487). This effectively mimics a “peeling argument” over uncountably infinite layers. We can avoid unnecessary analysis by using the double trick instead of deploying a fine-tuning peeling device as in [2], which may become frustrating in the Bernoulli case.

* To further establish a $\sqrt{\dot\mu_1 \ln K}$ minimax ratio, we conduct refined analysis on F3 by splitting the cases based on whether the value of $N_{t,1}$ exceeds H, a new choice of threshold that ensures adaptivity to $\dot\mu_1$ (lines 498-512 and Remark 7).

We apologize that the important Appendix D.3.1 (which carries our intuition and summarizes the key techniques) was not linked from the original main submission; we will add that link from the main paper in the final version.

[1] Kaufmann, E., Korda, N., & Munos, R. (2012, October). Thompson sampling: An asymptotically optimal finite-time analysis. In International conference on algorithmic learning theory (pp. 199-213). Berlin, Heidelberg: Springer Berlin Heidelberg.

[2] Agrawal, S., & Goyal, N. (2017). Near-optimal regret bounds for thompson sampling. Journal of the ACM (JACM), 64(5), 1-24.

[3] Bian, J., & Jun, K. S. (2022, May). Maillard sampling: Boltzmann exploration done optimally. In International Conference on Artificial Intelligence and Statistics (pp. 54-72). PML

---

### Decision · Program_Chairs · 2023-09-21

**Decision:**

Accept (poster)

**Comment:**

The reviewers came to consensus that this paper has strong contribution in the context of randomized bandit algorithms. On the other hand, many presentation issues are raised such as the experiments, positioning of the algorithms and relation with existing work. I determined to recommend acceptance but I also agree with the opinions that the paper needs through revision on the above presentation issues. I strongly expect that the authors seriously take these comments and improve the paper in the final version.